# Tucker-FNO: Tensor Tucker-Fourier Neural Operator and its Universal Approximation Theory

**Guancheng Zhou**[1,2,*] **Zelin Zeng**[1,*] **Yisi Luo**[1,†] **Qi Xie**[1], **Deyu Meng**[1,3,†]
[1]Xi'an Jiaotong University, [2]Shanghai Innovation Institute,
[3]Macau University of Science and Technology

## Abstract

Fourier neural operator (FNO) has demonstrated substantial potential in learning mappings between function spaces, such as numerical partial differential equations (PDEs). However, FNO may suffer from inefficiencies when applied to large-scale, high-dimensional function spaces due to the computational overhead associated with high-dimensional Fourier and convolution operators. In this work, we introduce the Tucker-FNO, an efficient neural operator that decomposes the high-dimensional FNO into a series of 1-dimensional FNOs through Tucker decomposition, thereby significantly reducing computational complexity while maintaining expressiveness. Especially, by using the theoretical tools of functional decomposition in Sobolev space, we rigorously establish the universal approximation theorem of Tucker-FNO. Experiments on high-dimensional numerical PDEs such as Navier-Stokes, Plasticity, and Burger's equations show that Tucker-FNO achieves substantial improvement in execution time and performance over FNO. Moreover, by virtue of the compact Tucker decomposition, Tucker-FNO generalizes seamlessly to high-dimensional visual signals by learning mappings from the positional encoding space to the signal's implicit neural representations (INRs). Under this operator INR framework, Tucker-FNO gains consistent improvements on continuous signal restoration over traditional INR methods in terms of efficiency and accuracy. The code is available at https://github.com/GuanchengZhou/Tucker-FNO.

## 1 Introduction

Neural operator (NO) methods have demonstrated great potential in diverse fields of science (Li et al., 2020) and engineering (Pal et al., 2024). Compared with traditional neural network methods, which learn the mapping from signal spaces, NO methods (Li et al., 2021; 2023a; Li & Ye, 2025) design neural networks to learn mappings between function spaces. By discretely sampling a finite number of observations, the function space can summarize information in tensor form, enabling the implementation of NO methods in a discretized version. Equipped with the universal approximation theorem (Kovachki et al., 2021), NO methods can be applied to a variety of tasks, including fast solution of partial differential equations (PDEs) (Li et al., 2021) and signal restoration (Pal et al., 2024; Liu & Tang, 2025).

As representative examples, PDEs describe the phenomena in physics, engineering, and other fields from a mathematical perspective. Traditional numerical solvers (e.g., finite element methods and finite difference methods) face the challenge of balancing solving speed and accuracy. In contrast, to obtain the PDE solution for a new condition, NO only needs to run the neural networks once, which greatly accelerates the solving speed. The universal approximation theorem of neural operators (Li et al., 2021) further guarantees the performance of NO methods in learning PDE solutions.

Another example is the implicit neural representation (INR) using NOs. Specifically, (Pal et al., 2024) has shown the ability of NO in signal restoration by leveraging NO-based INR, uncovering

---

[*]Equal contribution.
[†]Corresponding authors.

the potential of NO for general signal processing. The INR methods parameterize data through learning the mappings from the coordinates to the value of the signal, which can be considered as a coordinates-value transform. NO-based INR methods learn the mapping between two function spaces, which are related to the coordinate space and signal space. By learning data representations in such a function space transformation, the NO-based INR (Pal et al., 2024) can be effectively trained and evaluated, and can uniquely control the spatial interpolation behavior explicitly.

Traditional NOs based on data-domain integral operators may lack the efficiency comparable to that of traditional neural networks. Fourier neural operators (FNOs) (Li et al., 2021) address this by learning mappings in Fourier space, showing promise for efficient function mappings. However, the Fourier transform is still computationally expensive in high dimensions, with complexity reaching $O(d_v n^3 \log n^3)$ for 3-dimensional PDEs on an $n \times n \times n$ grid and $d_v$-dimensional latent space. This cost makes FNO relatively inefficient for high-dimensional PDEs. Furthermore, both conventional NOs and FNOs suffer from the curse of dimensionality for representing high-dimensional visual data, which may limit their efficiency in related applications using INR (Pal et al., 2024).

Low-rank decomposition can effectively reduce model dimensions and enhance computational and parameter efficiency (Kolda & Bader, 2009a; Luo et al., 2024). Motivated by the compactness of low-rank decomposition, we propose Tucker-FNO, a novel NO that leverages Tucker decomposition to enhance operator learning. Especially, we extend the recent functional tensor decomposition (Luo et al., 2024) to the operator learning level for the first time, and decompose a high-dimensional FNO into a series of 1-dimensional FNOs in the Tucker format. This splits the high-dimensional Fourier transform into multiple 1-dimensional transforms, reducing computational complexity from $O(n^3 \log n^3)$ to $O(3d_v n \log n)$ for 3-dimensional PDEs. Furthermore, using the Stone-Weierstrass and universal approximation theorems, we formally derive a tensor product decomposition of high-dimensional functions and prove the universal approximation capability of Tucker-FNO—the first such result for a tensor-decomposed NO. The contributions of this work are as follows:

- We propose a novel Tucker-FNO method. Different from traditional FNO methods, our methods utilize Tucker tensor decomposition to decompose the high-dimensional FNO into multiple 1-dimensional FNOs, which accelerates the FNO through operator decomposition.

- By utilizing theoretical tools of functional decomposition, we establish the universal approximation theorem for the decomposition-based FNO, which provides a theoretical guarantee for the representation ability of Tucker-FNO.

- Extensive experiments demonstrate the superior performance and efficiency of our methods for numerical PDE solutions. We also extend the Tucker-FNO to high-dimensional signal restoration tasks. For signal restoration tasks, our method shows performance that exceeds that of traditional INRs and similar NO-based INR.

## 2 RELATED WORK

### 2.1 NEURAL OPERATORS

Operator learning constructs mapping from parameter function spaces to solution spaces, and is represented discretely on grids via finite observations in practice (Lu et al., 2021; Raissi et al., 2019; Mishra & Molinaro, 2023; Kovachki et al., 2023). While conventional neural operators are mesh-dependent and require retraining for different resolutions, Li et al. (2020) introduced a resolution-invariant graph neural operator (GNO), later improved via fast Fourier transforms (FFT) to construct the Fourier neural operator (Li et al., 2021). Subsequent variants like Geo-FNO (Li et al., 2023a) handle arbitrary geometries, F-FNO (Tran et al., 2023) uses separable spectral layers to reduce parameters and enable deeper architectures, and T-FNO (Kossaifi et al., 2024) decompose the weights in FNO to reduce the number of parameters. While these methods improve model performance and reduce parameter number, they incur additional computational overhead. Recently, D-FNO (Li & Ye, 2025) observes that FFT calculations dominate the overall computational complexity in FNO and introduces decomposition to mitigate this cost in FNO. However, such decomposition may compromise the representational capacity of FNO (e.g., its universal approximation property), potentially leading to performance degradation. Our proposed Tucker-FNO applies tensor Tucker decomposition to decompose the high-dimensional FNO, replacing high-dimensional FFT with multiple 1-dimensional FFTs while preserving expressiveness (in most cases, enhancing performance).

Especially, we theoretically demonstrate that Tucker-FNO still possesses the universal approximation theorem.

## 2.2 Implicit Neural Representation

Implicit neural representations have garnered great success in continuously representing signals across various tasks and modalities, such as signal restoration (Chen & Wang, 2022; Shi et al., 2024; Liu et al., 2024; Saragadam et al., 2022; 2023; Luo et al., 2024), generation (Skorokhodov et al., 2021; Chen & Zhang, 2019), and object detection (Zheng et al., 2024). NeRF (Mildenhall et al., 2020) and subsequent works (Feng et al., 2024; Takikawa et al., 2021; Yariv et al., 2021) leveraged INRs to predict RGB values and density for any given coordinates and view directions within a 3-dimensional scene. To mitigate the computational overhead associated with traditional INR methods, incorporating tensor decomposition-based approaches (Luo et al., 2024) can yield superior and more robust results. Recent research (Pal et al., 2024) studies the potential of NO in signal restoration tasks through combining NO and INR. Specifically, OINR (Pal et al., 2024) learns a mapping from the positional encoding space to the signal's INR space, which improves the representation ability and can effectively handle downstream signal processing tasks such as denoising.

## 3 Proposed Methods

### 3.1 Notation and Preliminary

Scalars, vectors, matrices, tensors, sets and operators are denoted by fonts $x$, $\mathbf{x}$, $\boldsymbol{X}$, $\mathcal{X}$, $\mathbf{A}$, and $\mathscr{G}$, respectively. The $i$-th element of a vector $\mathbf{x}$ is denoted by $\mathbf{x}|_{(i)}$, and it is similar for matrices and tensors, i.e., $\boldsymbol{X}|_{(i_1,i_2)}$ and $\mathcal{X}|_{(i_1,i_2,\cdots,i_N)}$. The unfolding operator of a tensor $\mathcal{X} \in \mathbb{R}^{n_1 \times n_2 \times \cdots \times n_N}$ along the $d$-th mode ($d = 1, 2, \cdots, N$) is defined as $\mathtt{unfold}_d(\cdot) : \mathbb{R}^{n_1 \times \cdots \times n_N} \to \mathbb{R}^{n_d \times \Pi_{j \neq d} n_j}$, which returns the unfolding matrix along the mode $d$, and the unfolding matrix is denoted by $\boldsymbol{X}^{(d)} := \mathtt{unfold}_d(\mathcal{X})$. $\mathtt{fold}_d(\cdot)$ denotes the inverse operator of $\mathtt{unfold}_d(\cdot)$. The mode-$d$ ($d = 1, 2, \cdots, N$) tensor-matrix product is defined as $\mathcal{X} \times_d \boldsymbol{A} := \mathtt{fold}_d(\boldsymbol{A}\boldsymbol{X}^{(d)})$, which returns a tensor. $\mathbb{T} \subset \mathbb{R}$ is the periodic torus, identified with $[0, 2\pi]$. The Tucker rank of a tensor $\mathcal{X} \in \mathbb{R}^{n_1 \times n_2 \times \cdots \times n_N}$ is a vector defined as $rank_T(\mathcal{X}) := [rank(\boldsymbol{X}^{(1)}), \cdots, rank(\boldsymbol{X}^{(N)})]$ (Kolda & Bader, 2009b).

**Lemma 1** (Tensor Tucker decomposition (Kolda & Bader, 2009b)). *Let $\mathcal{X} \in \mathbb{R}^{n_1 \times n_2 \times \cdots \times n_N}$ be a tensor. If the Tucker rank of $\mathcal{X}$ is $[r_1, \cdots, r_N]$, then there exist $N$ factor matrices $\boldsymbol{U}_1 \in \mathbb{R}^{n_1 \times r_1}, \boldsymbol{U}_2 \in \mathbb{R}^{n_2 \times r_2}, \cdots, \boldsymbol{U}_N \in \mathbb{R}^{n_N \times r_N}$ and a core tensor $\mathcal{C} \in \mathbb{R}^{r_1 \times r_2 \times \cdots \times r_N}$ such that*

$$\mathcal{X} = \mathcal{C} \times_1 \boldsymbol{U}_1 \times_2 \boldsymbol{U}_2 \times_3 \cdots \times_N \boldsymbol{U}_N, \tag{1}$$

*which is called the tensor Tucker decomposition and can be written in an equivalent form as*

$$\mathcal{X}|_{(i_1,\cdots,i_N)} = \mathcal{C} \times_1 \boldsymbol{U}_1|_{(i_1,:)} \times_2 \cdots \times_N \boldsymbol{U}_N|_{(i_N,:)}. \tag{2}$$

### 3.2 The Setting of FNO

The operator is a mapping from two infinite-dimensional function spaces, i.e., $\mathscr{G} : \mathbf{A} \to \mathbf{U}$, where $\mathbf{A} \subset H^s(\mathbb{T}^d; \mathbb{R}^{d_a})$ and $\mathbf{U} \subset H^s(\mathbb{T}^d; \mathbb{R}^{d_u})$ are two function spaces. The representation of these function spaces specifically depends on the particular problem. In general, the solution of a PDE problem is a function $u$, which satisfies a constraint of the PDE form with condition $a$, and there exists an operator to learn that maps from condition $u$ to the target function $u$. Specifically, we consider a PDE problem as

$$\begin{cases} (\mathscr{L}_a u)(\mathbf{x}) = 0, & \mathbf{x} \in \mathbf{D}, \\ u(\mathbf{x}) = 0, & \mathbf{x} \in \partial\mathbf{D}, \end{cases} \tag{3}$$

where $u \in H^s(\mathbb{T}^d, \mathbb{R}^{d_u})$ is the target solution function, $\mathscr{L}_a$ is a differential operator which depends on the condition function $a \in H^s(\mathbb{T}; \mathbb{R}^{d_a})$, $\mathbf{D} \subset \mathbb{T}^d$ denotes the physical domain, and $\partial\mathbf{D} \subset \mathbf{D}$ denotes the boundary condition domain. Following the previous work (Kovachki et al., 2021), we assume that $d_a = d_u = 1$. Otherwise, we can re-formulate the target function $a : \mathbb{T}^d \to \mathbb{R}^{d_a}$ and $u : \mathbb{T}^d \to \mathbb{R}^{d_u}$ as $\hat{u}(\mathbf{x}, r) = u(\mathbf{x})|_{(r)}$, which satisfies $d_a = d_u = 1$. The goal of our operator

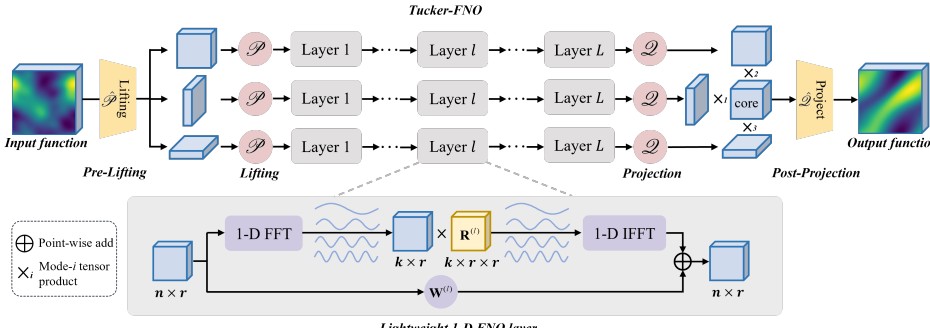

Figure 1: The architecture of the proposed Tucker-FNO. Given a sample from the condition function, the pre-lifting module extracts the input into a series of factor inputs. Then, the inputs are processed by the factor 1-dimensional FNOs, and the outputs are aggregated by Tucker decomposition. Note that the last post-projection does not influence the universal approximation.

learning is to learn an operator $\mathscr{G} : a \mapsto u$ that can efficiently approximate the PDE solution in equation 3.

NO methods (Kovachki et al., 2023; Li et al., 2021) construct a map iteratively from $a$ to $u$:

$$u = \mathscr{G}(a) = (\mathscr{Q} \circ \mathscr{L}^L \circ \cdots \circ \mathscr{L}^1 \circ \mathscr{P})(a), \tag{4}$$

where $\circ$ denotes the function composition, $L$ is the number of iterations, $\mathscr{P}$ is the lifting operator that maps the input to the $d_v$-dimensional latent representation, $\mathscr{L}^{(l)}$ is $l$-th non-linear operator layer, and $\mathscr{Q}$ is the projection operator that maps the $d_v$-dimensional latent representation to the $d_u$-dimensional output. For FNO, the non-linear operator layer $\mathscr{L}^{(l)}$ are defined as

$$\mathscr{L}^{(l)}(v)(\mathbf{x}) := \sigma\left(v(\mathbf{x})\boldsymbol{W}^{(l)} + \mathbf{b}^{(l)}(\mathbf{x}) + \mathscr{F}^{-1}(\sum_{\mathbf{k}} \mathscr{F}(v)(\mathbf{k}) \cdot \boldsymbol{R}^{(l)}(\mathbf{k}))(\mathbf{x})\right), \tag{5}$$

where $\mathscr{F}$ and $\mathscr{F}^{-1}$ denote the Fourier transform and the inverse Fourier transform (see Lemma 3 in Appendix B), $\boldsymbol{W}^{(l)} \in \mathbb{R}^{d_v \times d_v}$, $\mathbf{b}^{(l)}$ define a point-wise affine mapping, $\mathbf{k} \in \mathbb{Z}^d$ denotes the frequency, and $\boldsymbol{R}^{(l)} : \mathbb{Z}^d \to \mathbb{C}^{d_v \times d_v}$ defines the coefficients of a non-local linear mapping via the Fourier transform. For an efficient implementation, FNO keeps only the top-$\mathbf{k_{max}}$ Fourier frequency for each dimension. Specifically, we only consider the frequency in $\{\mathbf{k} \in \mathbb{Z}^d : \mathbf{k}|_{(i)} \le \mathbf{k_{max}}|_{(i)}, i = 1, 2, \cdots, d\}$.

To learn operators numerically, NO-based methods assume access only to point-wise evaluations to work with $a$ and $u$ numerically, and utilize a grid discrete sample $\mathbf{I}$ and $\mathbf{O}$ on functions $a$ and $u$ to represent the function information, i.e.,

$$\mathbf{I} := \{a(\mathbf{x}) : \mathbf{x} \in \mathbf{S}\}, \quad \mathbf{O} := \{u(\mathbf{x}) : \mathbf{x} \in \mathbf{S}\},$$
$$\mathbf{S} := \{(j_1\Delta_1, j_2\Delta_2, \cdots, j_d\Delta_d) \in \mathbb{T}^d : \Delta_k = 2\pi/n_k, j_k \in \{0, 1, \cdots, n_k\}, k = 1, 2, \cdots, d\}. \tag{6}$$

The sample set $\mathbf{S}$ is a $\Pi_{i=1}^d(n_i + 1)$-point discretization of the domain $\mathbb{T}^d$. Therefore, we have observation $\mathbf{I}$ and $\mathbf{O}$ for a finite collection of input-output pairs. Therefore, the operator learning task can be formulated discretely as learning an operator $\mathscr{G}$ satisfying $\mathbf{O} = \mathscr{G}(\mathbf{I})$. The Fourier transform is realized through FFT, which is a discrete Fourier transform. The discrete Fourier transform is computed efficiently via FFT. For a 3-dimensional PDE estimation problem of sample size $n \times n \times n$, the complexity of the Fourier transform is $O(d_v n^3 \log(n^3))$, which is relatively large w.r.t. $n$.

## 3.3 THE SETTING OF TUCKER-FNO

Inspired by the tensor Tucker decomposition in Lemma 1, a continuous function can be approximated by a functional tensor Tucker decomposition.

**Theorem 1** (Universal approximation theorem of functional tensor Tucker decomposition). *Let $a : \Omega \to \mathbb{R}$ be a continuous function defined on a compact subset $\Omega = [s_1, t_1] \times \cdots \times [s_d, t_d] \subset \mathbb{R}^d$. For*

*any $\epsilon > 0$, there exists a core tensor $\mathcal{C} \in \mathbb{R}^{r_1 \times \cdots \times r_d}$, and continuous functions $g_i : [s_i, t_i] \to \mathbb{R}^{r_i}$, such that*

$$\sup_{\mathbf{x} \in \Omega} |f(\mathbf{x}) - \mathcal{C} \times_1 g_1(\mathbf{x}|_{(1)}) \times_2 \cdots \times_d g_d(\mathbf{x}|_{(d)})| < \epsilon, \tag{7}$$

*where $r_i > 0$ is the Tucker rank.*

*Sketch of proof.* The detailed proof is proposed in Appendix C. From the Stone-Weierstrass theorem, the function can be approximated by sufficiently high-order polynomials, and the high-order polynomials can construct the Tucker functional tensor decomposition.

The above theorem proves that an arbitrary continuous function can be approximated by the functional tensor Tucker decomposition to arbitrary precision. Therefore, we can estimate the target function $u$ by

$$u(\mathbf{x}) \approx \mathcal{C} \times_1 g_1(\mathbf{x}|_{(1)}) \times_2 \cdots \times_d g_d(\mathbf{x}|_{(d)}), \tag{8}$$

and the operator learning task transforms from estimating operators on $u$ to estimating operators on $g_i$. Motivated by this insight, we propose the *Tucker-Fourier Neural Operator* (Tucker-FNO), which decomposes the high-dimensional FNO into a series of 1-dimensional FNOs via tensor Tucker decomposition. Specifically, we define the Tucker-FNO $\mathcal{N} : a \mapsto u$ as:

$$\mathcal{N}(a)(\mathbf{x}) := \mathcal{C} \times_1 \mathcal{N}_1(a)(\mathbf{x}|_{(1)}) \times_2 \cdots \times_d \mathcal{N}_d(a)(\mathbf{x}|_{(d)}), \tag{9}$$

where $\mathcal{C} \in \mathbb{R}^{r_1 \times \cdots \times r_d}$, $(r_1, \cdots, r_d) \in \mathbb{N}_+^d$ is the rank of Tucker-FNO, and $\mathcal{N}_i$ is the 1-dimensional FNO described in equation 5.

Since we decompose the $d$-dimensional FNO into $d$ 1-dimensional FNOs, we correspondingly construct a pre-lifting module, which transforms the $d$-dimensional representative input tensor $\mathcal{I}|_{(i_1,i_2,\cdots,i_d)} := a\left(\frac{i_1}{n_1}2\pi, \cdots, \frac{i_d}{n_d}2\pi\right) \in \mathbb{R}^{n_1 \times \cdots \times n_d}$ into a series of 1-dimensional representative factor matrices $\boldsymbol{I}_i$. Motivated by D-FNO (Li & Ye, 2025), the 1-dimensional representative factor matrix $\boldsymbol{I}_i$ can be extracted linearly through a pre-lifting operator $\hat{\mathscr{P}}$, expressed as:

$$\hat{\mathscr{P}}(\mathcal{I}) := (\boldsymbol{I}_1, \cdots, \boldsymbol{I}_d), \text{ where } \boldsymbol{I}_i := \texttt{unfold}_i(\mathcal{I})\boldsymbol{W}_i, \tag{10}$$

where $\boldsymbol{W}_i \in \mathbb{R}^{\Pi_{j \neq i} n_j \times d_v}$ is the pre-lifting weight, $d_v$ is the latent dimension, and $\boldsymbol{I}_i \in \mathbb{R}^{n_i \times d_v}$ is the $i$-th factor matrix. Therefore, each representative factor matrix $\boldsymbol{I}_i$ contains partial information of the function $a$. To enhance the representability of Tucker-FNO, we empoly a post-projection $\hat{\mathscr{Q}}$ after the Tucker-FNO, where $\hat{\mathscr{Q}}$ utilizes the structure of a multi-layer perceptron (MLP) for the discrete setting. In general, we set its input dimension as $d_v$. Under discrete numerical computing, the Tucker-FNO equation 9 can be expressed as

$$\mathcal{O} = \hat{\mathscr{Q}}(\mathcal{C} \times_1 \mathcal{N}_1(\boldsymbol{I}_1) \times_2 \cdots \times_d \mathcal{N}_d(\boldsymbol{I}_d)), \tag{11}$$

where each $\mathcal{N}_i$ is a 1-dimensional FNO. In this way, we effectively decompose the $d$-dimensional FNO into a series of 1-dimensional FNOs by Tucker decomposition.

### 3.4 THE UNIVERSAL APPROXIMATION THEOREM FOR TUCKER-FNO

The classical FNO (Kovachki et al., 2021) enjoys the universal approximation theorem for any continuous operator learning, as stated below.

**Lemma 2** (Universal approximation theorem of FNO (Kovachki et al., 2021)). *Let $\mathscr{U} : H^s(\mathbb{T}^d; \mathbb{R}^{d_a}) \to H^{s'}(\mathbb{T}^d; \mathbb{R}^{d_u})$ be a continuous operator. Let $\mathbf{F} \subset H^s(\mathbb{T}^d; \mathbb{R}^{d_a})$ be a compact subset. Then for any $\epsilon > 0$, there exists an FNO $\mathcal{N} : H^s(\mathbb{T}^d; \mathbb{R}^{d_a}) \to H^{s'}(\mathbb{T}^d; \mathbb{R}^{d_u})$ of the form equation 5, which is continuous as an operator $H^s \to H^{s'}$, such that $\sup_{f \in \mathbf{F}} \|\mathscr{U}(f) - \mathcal{N}(f)\|_{H^s} \leq \epsilon$.*

Similarly, we aim to demonstrate that Tucker-FNO also possesses universal approximation capabilities. That is, we show that Tucker-FNO can approximate a broad class of nonlinear operators in continuous function spaces. The overview of our proof is presented in Appendix A. Especially, we first establish the universal approximation property for the operator Tucker decomposition. Specifically, according to Theorem 1, a multivariate function $a$ can be decomposed into a series of 1-dimensional factor functions using functional tensor Tucker decomposition. This decomposition can be further extended to the operator level, as stated below.

**Theorem 2** (Universal approximation theorem of operator Tucker decomposition). *Let* $\mathbf{H} \subseteq H^s(\mathbb{T}^d; \mathbb{R})$ *be a compact set of continuous functions with* $s > d/2$. *Assume that all* $h \in \mathbf{H}$ *are analytic on* $\mathbb{T}^d$. *Then, for every* $h \in \mathbf{H}$ *and any* $\epsilon > 0$, *there exists ranks* $(r_1, \cdots, r_d) \in \mathbb{N}_+^d$, *continuous operators* $\mathscr{U}_i : \mathbf{H} \to H^s(\mathbb{T}; \mathbb{R}^{r_i})$, *and a core tensor* $\mathcal{C} \in \mathbb{R}^{r_1 \times \cdots \times r_d}$, *such that*

$$\sup_{h \in \mathbf{H}} \|h(\cdot) - \mathcal{C} \times_1 \mathscr{U}_1 h(\cdot_1) \times_2 \cdots \times_d \mathscr{U}_d h(\cdot_d)\|_{H^s} \le \epsilon. \tag{12}$$

*Sketch of proof.* The detailed proof is placed in Appendix E. By Theorem 1 and the finite-dimensional function space constructed by the Fourier basis, the function in the finite-dimensional function space can be approximated by the Tucker decomposition form (Lemma 10 in Appendix D). Therefore, the infinite-dimensional function space can be further approximated by the finite-dimensional function space. Through the above approximation and Lemma 10, the proof is done.

In Theorem 2, the factor operator $\mathscr{U}_i$ can be further approximated by a 1-dimensional FNO using Lemma 2, and thus the universal approximation theorem of Tucker-FNO equation 9 is established.

**Theorem 3** (Universal approximation theorem of Tucker-FNO). *Let* $\mathscr{G} : H^s(\mathbb{T}^d; \mathbb{R}) \to H^{s'}(\mathbb{T}^d; \mathbb{R})$ *be a continuous operator. Suppose* $\mathbf{F} \subseteq H^s(\mathbb{T}^d; \mathbb{R})$ *be a compact set and* $\mathbf{H} := \mathscr{G}(\mathbf{F})$. *Assume that* $s, s' > d/2$ *and that all* $h \in \mathbf{H}$ *are analytic on* $\mathbb{T}^d$. *Then for all* $\epsilon > 0$, *there exists rank* $(r_1, \cdots, r_d) \in \mathbb{N}_+^d$, *Tucker-FNO* $\mathscr{N} : \mathbf{F} \to \mathbf{H}$ *of the form equation 9, and a core tensor* $\mathcal{C} \in \mathbb{R}^{r_1 \times \cdots \times r_d}$, *such that*

$$\sup_{f \in \mathbf{F}} \|\mathscr{G}(f) - \mathscr{N}(f)\|_{H^s} \le \epsilon. \tag{13}$$

The detailed proof of Theorem 3 is proposed in Appendix F. Thus, we demonstrate that Tucker-FNOs are universal approximators, capable of approximating any continuous operator to a desired accuracy. It is important to note that, owing to the universal approximation properties of MLP (Rosenblatt, 1958), the inclusion of the post-projection $\hat{\mathscr{Q}}$ in equation 11 does not influence the universal approximation theorem of Tucker-FNO.

## 3.5 TUCKER-FNO FOR SIGNAL RESTORATION

Following OINR (Pal et al., 2024), which demonstrates the potential of NO in signal processing through NO-based INR, we further introduce Tucker-FNO to high-dimensional signal restoration tasks. Specifically, the natural signal is continuous and can be considered as a continuous function in signal space $H^s(\mathbb{T}^d; \mathbb{R})$. Traditional INRs learn a map from the coordinate positional encoding to the signal space, and operator-based INRs learn an operator $\mathscr{G}$ from the input function space $\mathbf{F}$ to signal function space $\mathbf{H}$, where $\mathbf{F}, \mathbf{H} \in H^s(\mathbf{\Omega}, \mathbb{R})$ and $\mathbf{\Omega}$ is the domain of definition (e.g., a 2-dimensional plane for images, and a 3-dimensional cube for volumes). Following OINR, we define our input function in terms of sinusoidal positional encodings (Mildenhall et al., 2020) which involve sinusoids across many frequencies. Specifically, for a 2-dimensional plane for images, we define the input space $\mathbf{F}$ as

$$\mathbf{F} := \{f_S(x, y) : S > 0\}, \tag{14}$$

where $f_S(x, y) := [\sin(2^l \pi x), \cos(2^l \pi x), \sin(2^l \pi y), \cos(2^l \pi y), \cdots], l \in \{0, 1, \cdots, S-1\}$, where $S$ denotes the encoding length. Therefore, the operator-based INR can be defined as:

$$h(\mathbf{x}) = \mathscr{G}[f](\mathbf{x}), \tag{15}$$

where $f \in \mathbf{F}$ is the input function, $h \in \mathbf{H}$ is the signal function and $\mathscr{G}$ is the operator mapping from $\mathbf{F}$ to $\mathbf{H}$. For Tucker-FNO, the $d$-dimensional sinusoidal positional encodings can be reformulated as

$$f_S(\mathbf{x}|_{(1)}, \mathbf{x}|_{(2)}, \cdots, \mathbf{x}|_{(d)}) = [f_S^*(\mathbf{x}|_{(1)}), f_S^*(\mathbf{x}|_{(2)}), \cdots, f_S^*(\mathbf{x}|_{(d)})], \tag{16}$$

where $f_S^*(x) := [\sin(2^l \pi x), \cos(2^l \pi x), \cdots], l \in \{0, 1, \cdots, S-1\}$. Then, the sinusoidal positional encoding can be decomposed across each dimension, which is similar to Tucker decomposition. Therefore, thanks to the decomposition property, Tucker-FNO can remove the pre-lifting operator and be simplified to

$$\mathscr{G}[f_S] = \hat{\mathscr{Q}} \circ (\mathcal{C} \times_1 \mathscr{U}_1(f_1^*) \times_2 \cdots \mathscr{U}_d(f_d^*)). \tag{17}$$

Similar to Theorem 3, the Tucker-FNO here enjoys universal approximation ability to learn any high-dimensional signal structures (e.g., image signals).

Table 1: The comparisons of computational complexity and parameter number for FFT-related calculations. $d_v$ is the latent dimension, $k$ is the frequency truncation threshold, $n$ is the input size, $\hat{r}$ is the maximum of decomposition rank.

| Components | FNO | T-FNO | Tucker-FNO |
|---|---|---|---|
| FFT | $O(d_v dn^d \log(n))$ | $O(d_v dn^d \log(n))$ | $O(d_v dn \log(n))$ |
| Multiplying Weights | $O(d_v^2 k^d)$ | $O(\hat{r} d_v^2 k^d)$ | $O(d_v^2 dk)$ |
| Inverse FFT | $O(d_v dn^d \log(n))$ | $O(d_v dn^d \log(n))$ | $O(d_v dn \log(n))$ |
| Params | $O(d_v^2 k^d)$ | $O(\hat{r}^{d+2} + \hat{r}(d_v + k))$ | $O(d_v^2 dk)$ |

## 3.6 Computational Complexity Analysis

By decomposing the $d$-dimensional FNO into $d$ 1-dimensional FNOs, Tucker-FNO can efficiently process the high-dimensional data. Although recent tensorized FNOs (Kossaifi et al., 2024; Tran et al., 2023) also incorporate the tensor decomposition to improve FNO by decomposing the model weight to reduce the model parameter, it leads to higher computational complexity. As noted in a recent study (Li & Ye, 2025), FFT calculations dominate the overall computational complexity of FNO, indicating that the key to accelerating FNO lies in reducing the complexity of FFT-related calculations. In FNO, the FFT-related calculations includes FFT, multiplying weights, and inverse FFT. In the following analysis, we evaluate the computational complexity of FNO based on these components, using a $d$-dimensional grid of $n \times n \times \cdots \times n$ as the example to calculate the computational complexity. The complexity comparisons among FNO, T-FNO (Kossaifi et al. (2024), a recent tensorized FNO) , and Tucker-FNO are as follows:

- **FNO.** The computational complexity for FFT and inverse FFT is $O(d_v n^d \log(n^d)) = O(d_v dn^d \log(n))$, where $d_v$ is the latent dimension of FNO, and the computational complexity for multiplying weights is $O(d_v^2 k^d)$, where $k$ is the frequency truncation threshold.

- **T-FNO.** T-FNO decomposes the model weight $\mathcal{R} \in \mathbb{R}^{k \times k \times \cdots \times k \times d_v \times d_v}$, where $k$ is the frequency truncation threshold and $d_v$ is the latent dimension. Specifically, the weight $\mathcal{R}$ can be decomposed to $\mathcal{R} = \mathcal{C} \times_1 \boldsymbol{U}_1 \times_2 \cdots \times_{d+2} \boldsymbol{U}_{d+2}$, where $\mathcal{C} \in \mathbb{R}^{r_1 \times r_2 \times \cdots \times r_{d+2}}$ is the core tensor, $(r_1, \cdots, r_{d+2})$ is the rank, and $\boldsymbol{U}_i \in \mathbb{R}^{k \times r_i}$ (for $i \leq d$) or $\boldsymbol{U}_i \in \mathbb{R}^{d_v \times r_i}$ (for $i > d$) are the factor matrices. This weight decomposition does not influence the computational complexity, i.e., the computational complexity for FFT and inverse FFT is still $O(d_v dn^d log(n))$. For multiplying weights, the computational complexity for multiplying weights increase to $O(\hat{r} d_v^2 k^d + d_v^2 k^d) = O(\hat{r} d_v^2 k^d)$, where $\hat{r} = \max\{r_1, \cdots, r_{d+2}\}$ is the maximum of rank, and $O(\hat{r} d_v^2 k^d)$ is the computational complexity for the decomposition of $\mathcal{R}$. Due to the computation for the the decomposition of $R$, T-FNO incurs increased computational complexity.

- **Tucker-FNO.** By decomposing the $d$-dimensional FNO into a series of 1-dimensional FNO, Tucker-FNO processes 1-dimensional FFT and inverse FFT for $d$ times, i.e., the computational complexity for FFT and inverse FFT is reduced to $O(d_v dn \log(n))$. Similarly, the computational complexity for multiplying weights is reduced to $O(d_v^2 dk)$, where $k$ is the frequency truncation threshold.

For clarity, the computational complexities of FNO, T-FNO, and Tucker-FNO are summarized in Table 1. Notably, Tucker-FNO significantly reduces the computational burden associated with FFT-related operations. Furthermore, in Tucker-FNO, decomposition is performed after all FFT-related computations, with a complexity of $O(\hat{r} d_v n^d)$, where $\hat{r}$ is the maximum of rank. While this complexity may appear to grow rapidly, it only becomes dominant when the rank is extremely large, which is validated in Appendix I. Beyond computational efficiency, Tucker-FNO also reduces the number of model parameters, as shown by the weight scales reported in Table 1.

**Remark.** *In summary, although both T-FNO and Tucker-FNO employ tensor decomposition, their implementations and motivations differ. T-FNO applies decomposition to the network parameters to reduce the parameter number, introducing an parameterization scheme for FNO. In contrast, Tucker-FNO applies decomposition directly to the operator, aiming to reduce computational complexity. Thus, T-FNO focuses on parameter efficiency, whereas Tucker-FNO proposes a novel FNO-based paradigm centered on computational efficiency.*

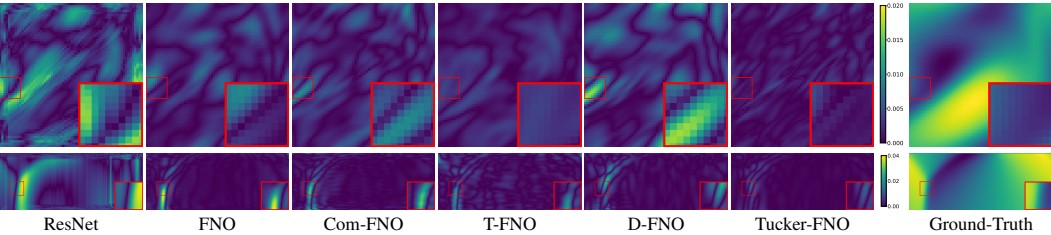

Figure 2: Error maps comparisons of different NO methods for Navier-Stokes (the first row) and Burger's equation (the second row).

# 4 EXPERIMENTS

## 4.1 EXPERIMENT SETTINGS

**Task.** To validate the effectiveness and efficiency of Tucker-FNO, we evaluate the performance on the NO-based PDE approximation task and INR-based signal restoration task. For PDE approximation tasks, we compare the proposed Tucker-FNO with multiple neural network architectures and NO-based approximation methods on the Navier-Stokes equation, the Plasticity equation, and the Burger equation. The details of PDE datasets are presented in Appendix G. For signal restoration tasks, we utilize the NO-based INR architectures (Pal et al., 2024) and compare the proposed Tucker-FNO-based INR in equation equation 17 with multiple INR signal restoration methods and other NO-based INRs on the image signal denoising, inpainting, and 3-dimensional volume representation tasks for different signals, e.g., multi-spectral images (MSIs), multi-dimensional images, and 3-dimensional volume. For image denoising, the testing data are MSIs from the benchmark CAVE dataset* (Yasuma et al., 2010), including *Cups*, *Balloons*, *Cloth*, *Toys*, *Clay*, and *Fruits*. We consider three noisy cases containing Gaussian noise with different noise deviations of 0.1, 0.15, and 0.2. For image inpainting, testing data consists of two videos (*Foreman* and *Carphone*†). We consider random missing with observed sampling rates (SRs) 0.1, 0.15, and 0.2 for image inpainting. For the 3-dimensional volume representation task, the occupancy volume is sampled in a $512 \times 512 \times 512$ voxel grid following OINR (Pal et al., 2024), where each voxel within the volume is assigned a value of 1 inside an object and 0 outside an object.

**Benchmark.** For PDE approximation tasks, we compare our methods with ResNet18 (He et al., 2016), FNO (Li et al., 2021), Com-FNO (Li et al., 2024), T-FNO (Kossaifi et al., 2024), D-FNO (Li & Ye, 2025). For signal restoration tasks, we compare our methods with sinusoidal representation networks (SIREN) (Sitzmann et al., 2020), multiplicative filter networks (MFN) (Fathony et al., 2020), low-rank tensor function representation (LRTFR) (Luo et al., 2024), OINR (Pal et al., 2024), and OINR-FNO, which replaces NO with FNO in OINR.

**Metric.** For the PDE approximation task, following F-FNO Tran et al. (2023), the results are quantitatively evaluated by normalized root mean square error (NMSE) and vorticity correlation error (VCE, $1 - vorticity\ correlation$). For signal restoration tasks, the results are quantitatively evaluated by peak-signal-to-noise ratio (PSNR), structural similarity (SSIM), and NMSE.

Table 2: Performance (NMSE ↓, VCE ↓) on Navier-Stokes, Burger, and Plasticity equations among different methods. The **best** and second-best values are highlighted.

| Config. | Navier-Stokes $T = 20$ | | $T = 30$ | | Plasticity | | Burger | |
|---|---|---|---|---|---|---|---|---|
| Metric | NMSE | VCE | NMSE | VCE | NMSE | VCE | NMSE | VCE |
| ResNet | 0.0911 | 0.0053 | 0.2582 | 0.0351 | 0.1015 | 0.0587 | 0.0941 | 0.0651 |
| FNO | 0.0034 | 1.0311e-5 | 0.0072 | 3.9776e-5 | 0.0080 | **1.9514e-4** | 0.0087 | 4.4465e-5 |
| Com-FNO | 0.0036 | 1.1265e-5 | 0.0071 | 3.8146e-5 | 0.0084 | 2.0897e-4 | **0.0063** | 2.4543e-5 |
| T-FNO | 0.0035 | 8.8214e-6 | 0.0065 | 3.0354e-5 | 0.0079 | 2.0076e-4 | 0.0075 | 2.9672e-5 |
| D-FNO | 0.0063 | 3.0994e-5 | 0.0132 | 9.1426e-5 | 0.0097 | 2.2150e-4 | 0.0071 | 2.8610e-5 |
| Tucker-FNO | **0.0028** | **8.2254e-6** | **0.0061** | **2.9742e-5** | **0.0073** | 2.0010e-4 | 0.0070 | **2.0050e-5** |

## 4.2 TUCKER-FNO IN PDE APPROXIMATION TASK

**Setup.** The details of the settings of the PDEs are presented in Appendix G. Here, we perform training for 500 epochs using a batch size of 20 using the Adam optimizer (Kingma, 2014) with a

---

*https://www.cs.columbia.edu/CAVE/databases/multispectral/

†http://trace.eas.asu.edu/yuv/

Table 4: The average quantitative results (PSNR ↑ (dB), SSIM ↑, MSE ↓) by different methods for multi-spectral image inpainting. The **best** and second-best values are highlighted.

| Sampling rate | | 0.1 | | | 0.15 | | | 0.2 | | |
|---|---|---|---|---|---|---|---|---|---|---|
| Data | Metric | PSNR | SSIM | NMSE | PSNR | SSIM | NMSE | PSNR | SSIM | NMSE |
| Videos *Foreman* *Carphone* (144×176×100) | Observed | 5.20 | 0.023 | 0.949 | 5.45 | 0.035 | 0.922 | 5.71 | 0.047 | 0.895 |
| | SIREN | 23.85 | 0.806 | 0.112 | 24.11 | 0.815 | 0.109 | 24.61 | 0.821 | 0.105 |
| | MFN | 22.47 | 0.778 | 0.132 | 24.16 | 0.801 | 0.108 | 26.20 | 0.828 | 0.083 |
| | LRTFR | 26.96 | 0.813 | 0.071 | 28.75 | 0.870 | 0.064 | 30.22 | 0.903 | 0.056 |
| | OINR | 23.32 | 0.738 | 0.108 | 24.04 | 0.791 | 0.095 | 25.17 | 0.840 | 0.083 |
| | OINR-FNO | 24.53 | 0.803 | 0.080 | 25.02 | 0.837 | 0.076 | 26.27 | 0.857 | 0.072 |
| | Tucker-FNO | **28.09** | **0.860** | **0.074** | **29.57** | **0.901** | **0.059** | **31.17** | **0.939** | **0.047** |

| PSNR 3.61 | PSNR 23.26 | PSNR 21.58 | PSNR 26.24 | PSNR 21.56 | PSNR 24.06 | PSNR 27.54 | PSNR Inf |
|---|---|---|---|---|---|---|---|

| PSNR 7.05 | PSNR 25.27 | PSNR 25.86 | PSNR 28.11 | PSNR 25.8 | PSNR 25.38 | PSNR 30.17 | PSNR Inf |
| Observed | SIREN | MFN | LRTFR | OINR | OINR-FNO | Tucker-FNO | Original |

Figure 3: The results of multi-dimensional image inpainting by different methods on video *Foreman* (SR=0.1) and *Carphone* (SR=0.15).

learning rate of 0.001. The latent dimension $d_v$ is set 32, and the Tucker rank of Tucker-FNO is set to the same as the latent dimension. The Tucker rank rate for T-FNO is $0.4$, which indicates that the Tucker rank of each dimension is $0.4$ of the size of the dimension. The Fourier frequency truncation threshold is $\mathbf{k_{max}} = (12, 12, \cdots)$ for each dimension. The depth of NO-based methods is 4. And the post-projection $\hat{\mathscr{Q}}$ is a 2-layer MLP.

***Result summary.*** The results of PDE approximation task are shown in Table 2 and Fig. 2. Tucker-FNO outperforms the baselines FNO, T-FNO and D-FNO, achieving the best performance. From the Fig. 2, the comparisons of the error maps on Navier-Stokes and Burger's equations and the results on the Plasticity equation demonstrate that Tucker-FNO performs better on regions with significant numerical variation. More qualitative results are demonstrated in Appendix H. Additionally, the proposed Tucker-FNO demonstrates outperforming advantages in efficiency. From Table 3, we can observe that the proposed Tucker-FNO is more efficient than FNO, T-FNO and D-FNO in terms of the parameter number of model parameters and the execution time. In $256 \times 256$ data, Tucker-FNO reduces the parameter by 50% and the execution time by 48.6% compared to FNO. T-FNO reduces the parameters but increases computational complexity. While D-FNO reduces the computation complexity, it compromises performance as shown in Table 2. In contrast, Tucker-FNO excels in both performance and efficiency.

Table 3: Comparisons (Params ↓, Time/iter ↓ (seconds)) of parameter number and running time in $256 \times 256$ data.

| Method | Params | Time/iter |
|---|---|---|
| FNO | 593.7 K | 0.0685 |
| T-FNO | 232.6 K | 0.0746 |
| D-FNO | 307.8 K | 0.0382 |
| Tucker-FNO | 293.5 K | 0.0352 |

## 4.3 TUCKER-FNO IN SIGNAL RESTORATION

***Setup.***
The rank of Tucker-FNO is set $(r_1, r_2, r_3) = (n_1, n_2, \lfloor n_3/2 \rfloor)$, where $n_d$s ($d = 1, 2, 3$) denote the sizes of the observed data. We parameterize each 1-dimensional factor mapping as a DNN, which consists of one lifting layer with sine activation, two FNO layers with a latent dimension of $d_v = 32$ and a final linear projection across all experiments. The factor network takes sinusoidal positional encodings described in equation 16 with $S = 10$. The post-projection is a 2-layer MLP. We use the Adam

Table 5: Comparisons (Params ↓, Time/iter ↓ (seconds)) of parameter number and running time in the signal restoration task with $1024 \times 1024 \times 31$ data.

| Method | Params | Time/iter |
|---|---|---|
| OINR-FNO | 1.06 M | 0.2322 |
| Tucker-FNO | 553 K | 0.0963 |

Table 6: Ablation experiments of latent dimensions and frequency truncation threshold for PDE solving problems in Plasticity equations. The Tucker rank is set as $d_v$.

| $k$ | 8 | | 12 | | 16 | |
|-----|------|-----|------|-----|------|-----|
| $d_v$ | NMSE | VCE | NMSE | VCE | NMSE | VCE |
| 16 | 0.0098 | 2.07e-4 | 0.0092 | 2.06e-4 | 0.0092 | 2.06e-5 |
| 32 | 0.0074 | 2.18e-4 | 0.0074 | 2.00e-4 | 0.0073 | 1.98e-4 |
| 64 | 0.0068 | 2.45e-4 | 0.0064 | 2.04e-4 | 0.0062 | 2.00e-4 |

Table 7: Ablation experiments of Tucker-FNO rank and the latent dimensions for PDE solving problems in Plasticity equations.

| Rank | 16 | | 32 | | 64 | |
|------|------|-----|------|-----|------|-----|
| $d_v$ | NMSE | VCE | NMSE | VCE | NMSE | VCE |
| 16 | 0.0092 | 2.06e-4 | 0.0087 | 1.84e-4 | 0.0086 | 1.97e-5 |
| 32 | 0.0079 | 2.03e-4 | 0.0074 | 2.00e-4 | 0.0071 | 1.78e-4 |
| 64 | 0.0064 | 1.87e-4 | 0.0064 | 2.00e-4 | 0.0063 | 2.04e-4 |

optimizer with a learning rate of 0.001 and no weight decay. To determine the optimal frequency truncation threshold $\mathbf{k_{max}} = (k, k, k)$ which controls the number of retained Fourier frequencies in each spatial dimension, we consider a range of candidate values for $k \in \{6, 12, 18, 24, 30, 36\}$ to balance model expressivity and computational efficiency.

***Result summary.*** The quantitative results of multi-dimensional image inpainting are summarized in Table 4. Our proposed Tucker-FNO achieves better performances and surpasses the baselines on video restoration with a considerable margin. As illustrated in Fig. 3, Tucker-FNO holds superior edge preservation and smoothness recovery for different videos. The results of image denoising and 3-dimensional volume representation are demonstrated in Appendix H. Similar to the PDE approximation task, Tucker-FNO also demonstrates advantages in efficiency. From Table 5, the proposed Tucker-FNO achieves a significant execution time reduction and parameter reduction in signal restoration as compared with OINR (Pal et al., 2024). The results of Tucker-FNO for these high-dimensional visual data validate its superior efficiency and applicability.

## 4.4 ABLATION EXPERIMENTS

Table 6 demonstrates the influence of latent dimension $d_v$ and the frequency truncation threshold $k$, where the rank is the same as $d_v$. Table 7 further examines the influence of the rank and $d_v$ while keeping $k$ fixed. From these results, Tucker-FNO's $d_v$, rank, and $k$ exhibit a positive correlation with the Tucker-FNO's performance, and the rank and $k$ demonstrate relatively greater robustness.

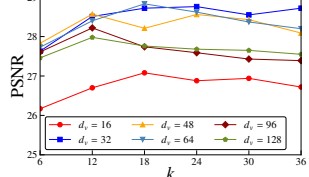

Figure 4: The PSNR results of Tucker-FNO w.r.t. different $d_v$ and $k$ of the network for *Foreman* (SR = 0.15) inpainting.

In the *Carphone* video reconstruction task, our method performs robustly across various $k$ and latent dimension $d_v$ in Fig. 4. Performance initially increases then decreases with these hyperparameters, peaking at $d_v = 32$ and $k = 24$. This pattern arises because excessive parameters lead to over-fitting, while moderate width implicitly captures signal low-rankness, facilitating faster and better convergence.

## 5 CONCLUSION

We proposed an efficient neural operator based on Tucker decomposition, termed Tucker-FNO, which decomposes the high-dimensional FNO into multiple 1-dimensional FNOs, thereby largely accelerating the classical FNO while maintaining expressiveness. Utilizing theoretical tools of functional decomposition, we establish the universal approximation theorem for Tucker-FNO, which provides a theoretical guarantee for the representation ability of Tucker-FNO. Extensive experiments demonstrate that Tucker-FNO achieves substantially improved performance in terms of both effectiveness and efficiency for the PDE approximation task and the signal restoration task. Moreover, Tucker-FNO has the potential in future research to handle higher-dimensional PDE approximation problems. The detailed optimization dynamics and implicit regularization of Tucker-FNO also warrant further exploration.

ACKNOWLEDGMENTS

This work was supported in part by the National Key R&D Program of China (2025YFA1016400), the Fundamental and Interdisciplinary Disciplines Breakthrough Plan of the Ministry of Education of China (JYB2025XDXM101), the Tianyuan Fund for Mathematics of the National Natural Science Foundation of China (12426105), and the National Natural Science Foundation of China (124B2029 and 62476214).

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

## A THE SKETCH OF PROOF FOR UNIVERSAL APPROXIMATION THEOREM OF TUCKER-FNO

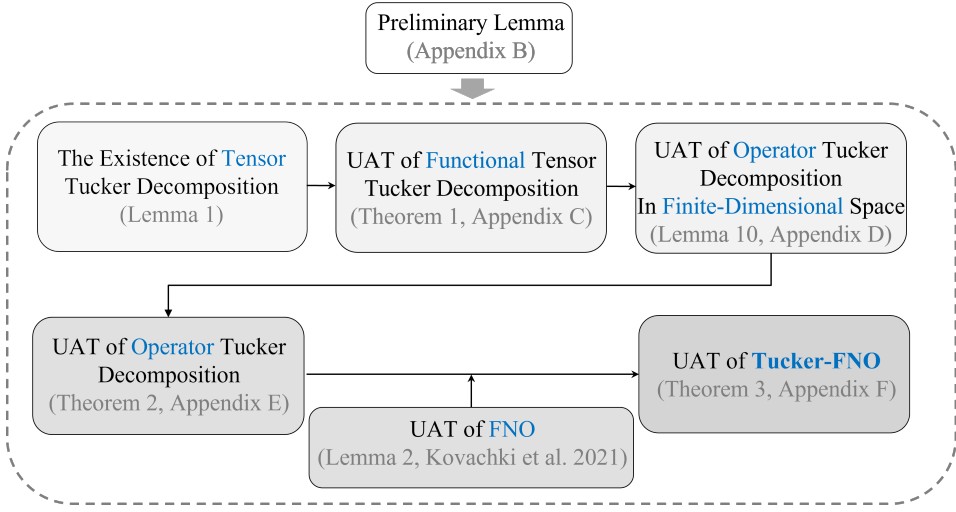

Figure 5: The sketch of proof for the universal approximation theorem of Tucker-FNO. For simplicity, the universal approximation theorem is abbreviated as UAT.

Our proof follows a general approach: starting from *the tensor level* (Lemma 1), we extend to *the function level* (Theorem 1), then to *operators in finite-dimensional function spaces* (Lemma 10), and finally demonstrate *universal approximation for operators in the general function space* (Theorem 2).

Specifically, in Appendix B, we demonstrate some preliminary auxiliary lemmas for the following proof. Then, in Appendix C, through Stone-Weierstrass theorem, we propose the universal approximation theorem of functional tensor Tucker decomposition. Therefore, in Appendix D, we further propose the universal approximation theorem of operator Tucker decomposition in finite-dimensional function space. In Appendix E, through the universal approximation theorem of FNO in (Kovachki et al., 2021), we finally propose the universal approximation theorem of Tucker-FNO.

## B   PRELIMINARY LEMMA

**Lemma 3** (Fourier transform and inverse Fourier transform). *For any such function $v \in L^2(\mathbb{T}^d)$, there exists Fourier Transformer as*

$$\mathscr{F}(v)(\mathbf{k}) := \frac{1}{(2\pi)^d} \int_{\mathbb{T}^d} v(x) e^{-i\mathbf{k}\cdot x} \mathrm{d}\mathbf{x}. \tag{18}$$

*For $\mathbf{k} \in \mathbb{Z}^d$, the $\mathbf{k}$-th Fourier coefficient of $v$ is denoted by $\hat{v}_{\mathbf{k}} = \mathscr{F}(v)(\mathbf{k})$.*

*The inverse Fourier Transformer is defined as:*

$$\mathscr{F}^{-1}(\hat{v})(\mathbf{x}) := \sum_{\mathbf{k}\in\mathbb{Z}^d} \hat{v}_k e^{i\mathbf{k}\cdot\mathbf{x}} \tag{19}$$

**Definition 1** (The $H^s$ norm). *Let $h \in \mathbf{H} \subseteq H^s(\mathbb{T}^d; \mathbb{R})$, where $\mathbb{T}^d = [0, 2\pi]^d$, its Fourier series expansion is given by:*

$$h(\mathbf{x}) = \sum_{\mathbf{k}\in\mathbb{Z}^d} c_{\mathbf{k}} e^{i\mathbf{k}\cdot\mathbf{x}}, \ c_{\mathbf{k}} = \frac{1}{(2\pi)^d} \int_{\mathbb{T}^d} h(\mathbf{x}) e^{-i\mathbf{k}\cdot\mathbf{x}} \mathrm{d}\mathbf{x}, \tag{20}$$

*where $\mathbf{k} = (\mathbf{k}|_{(1)}, \ldots, \mathbf{k}|_{(d)}) \in \mathbb{Z}^d \setminus \{0\}$. The $H^s$-norm of $h$ is*

$$||h||_{H^s}^2 = \sum_{\mathbf{k}\in\mathbb{Z}^d} (1 + |\mathbf{k}|^2)^s |c_{\mathbf{k}}|^2. \tag{21}$$

To prove that the combination of the Fourier basis can approximate any function in $H^s(\mathbb{T}^d, \mathbb{R})$, we propose Lemma 4-7, and then propose the approximation lemma of the Fourier basis in Lemma 8.

**Lemma 4.** *For any $d \geq 1$, $s \geq 0$ and $\mathbf{k} \in \mathbb{R}^d \setminus \{0\}$, we have*

$$(1 + |\mathbf{k}|^2)^s \leq \binom{s}{\lfloor s/2 \rfloor} \sum_{|\beta|\leq s} |\mathbf{k}|^{2|\beta|}. \tag{22}$$

*Proof.* According to the binomial theorem, we can expand $(1 + |\mathbf{k}|^2)^s$ as follows:

$$(1 + |\mathbf{k}|^2)^s = \sum_{m=0}^{s} \binom{s}{m} |\mathbf{k}|^{2m}. \tag{23}$$

The binomial coefficient $\binom{s}{m}$ achieves its maximum value when $m = \lceil s/2 \rceil$ or $m = \lfloor s/2 \rfloor$, thus we have

$$(1 + |\mathbf{k}|^2)^s \leq \sum_{m=0}^{s} \binom{s}{\lfloor s/2 \rfloor} |\mathbf{k}|^{2m}. \tag{24}$$

Since $\binom{m+d-1}{d-1} \geq 1$ for all $m \geq 0$, we have

$$\begin{aligned}
\sum_{m=0}^{s} \binom{s}{\lfloor s/2 \rfloor} |\mathbf{k}|^{2m} &\leq \sum_{m=0}^{s} \binom{s}{\lfloor s/2 \rfloor} \binom{m+d-1}{d-1} |\mathbf{k}|^{2m} \\
&= \binom{s}{\lfloor s/2 \rfloor} \sum_{m=0}^{s} \Big(\sum_{|\beta|=m} 1\Big) |\mathbf{k}|^{2m} \\
&= \binom{s}{\lfloor s/2 \rfloor} \sum_{|\beta|\leq s} |\mathbf{k}|^{2|\beta|}.
\end{aligned} \tag{25}$$

Combining equation 24 and equation 25, the proof is completed. $\qquad\square$

**Lemma 5.** *For any $d \geq 1$, $s \geq 0$ and $\mathbf{k} \in \mathbb{R}^d \setminus \{0\}$, we have*

$$\sum_{|\beta| \leq s} |\mathbf{k}|^{2|\beta|} \leq (d+1)^s |\mathbf{k}|^{2s}. \tag{26}$$

*Proof.* The direct computation is as follows

$$
\begin{aligned}
\sum_{|\beta| \leq s} |\mathbf{k}|^{2|\beta|} &= \sum_{m=0}^{s} \Big( \sum_{|\beta|=m} 1 \Big) |\mathbf{k}|^{2m} \\
&= \sum_{m=0}^{s} \binom{m+d-1}{d-1} |\mathbf{k}|^{2m} \\
&\leq \sum_{m=0}^{s} \binom{m+d-1}{d-1} |\mathbf{k}|^{2s} \\
&= \binom{s+d}{d} |\mathbf{k}|^{2s} \\
&= \prod_{i=1}^{s} \left( 1 + \frac{d}{i} \right) |\mathbf{k}|^{2s} \leq (d+1)^s |\mathbf{k}|^{2s}.
\end{aligned}
\tag{27}
$$

Therefore, the proof is completed. $\square$

**Lemma 6.** *Let $h \in \mathbf{H} \subseteq H^s(\mathbb{T}^d; \mathbb{R})$, where $\mathbb{T}^d = [0, 2\pi]^d$. Its Fourier series expansion is given by:*

$$h(\mathbf{x}) = \sum_{\mathbf{k} \in \mathbb{Z}^d} c_{\mathbf{k}} e^{i\mathbf{k} \cdot \mathbf{x}}, \ c_{\mathbf{k}} = \frac{1}{(2\pi)^d} \int_{\mathbb{T}^d} h(\mathbf{x}) e^{-i\mathbf{k} \cdot \mathbf{x}} \, d\mathbf{x}, \tag{28}$$

*where $\mathbf{k} = (\mathbf{k}|_{(1)}, \ldots, \mathbf{k}|_{(d)})$ and $|\mathbf{k}| = \sqrt{\mathbf{k}^2|_{(1)} + \cdots + \mathbf{k}^2|_{(d)}}$. We aim to prove that its Fourier coefficients $c_{\mathbf{k}}$ satisfy:*

$$|c_{\mathbf{k}}| \leq C_m |\mathbf{k}|^{-m}, \ \forall m \in \mathbb{N}, \ \mathbf{k} \in \mathbb{Z}^d \setminus \{0\}, \tag{29}$$

*where $C_m$ is a constant depending only on the $m$-th derivatives of $h$.*

*Proof.* Choose $j$ such that $\left| \mathbf{k}|_{(j)} \right| = \max_{1 \leq i \leq d} \left| \mathbf{k}|_{(i)} \right|$, then $\left| \mathbf{k}|_{(j)} \right| \geq |\mathbf{k}|/\sqrt{d}$. Perform $m$-th integration by parts along the $j$-th direction:

$$c_{\mathbf{k}} = \frac{1}{(i\mathbf{k}|_{(j)})^m} \cdot \frac{1}{(2\pi)^d} \int_{\mathbb{T}^d} \frac{\partial^m h(\mathbf{x})}{\partial \mathbf{x}|_{(j)}^m} e^{-i\mathbf{k} \cdot \mathbf{x}} \, d\mathbf{x}, \tag{30}$$

where $\mathbf{x}_i$ denote $\mathbf{x}|_{(i)}$ here for simplicity. Estimate and take absolute values:

$$|c_{\mathbf{k}}| \leq \frac{1}{\left| \mathbf{k}|_{(j)} \right|^m} \cdot \frac{1}{(2\pi)^d} \int_{\mathbb{T}^d} \left| \frac{\partial^m h(\mathbf{x})}{\partial \mathbf{x}|_{(j)}^m} \right| d\mathbf{x} \leq \frac{\|\partial_j^m h\|_{L^\infty}}{\left| \mathbf{k}|_{(j)} \right|^m}. \tag{31}$$

Using $\left| \mathbf{k}|_{(j)} \right| \geq |\mathbf{k}|/\sqrt{d}$, we can conclude $|c_{\mathbf{k}}| \leq C_m/|\mathbf{k}|^m$ with $C_m = (\sqrt{d})^m \|\partial_j^m h\|_{L^\infty}$. Therefore, the proof is completed. $\square$

**Lemma 7.** *For any $d \geq 1$, $s \geq 0$ and $\mathbf{x} \in \mathbb{R}^d \setminus \{0\}$, if $m - s \geq d/2$, we have*

$$\int_{\|\mathbf{x}\|_\infty > n} |\mathbf{x}|^{-2(m-s)} d\mathbf{x} \leq C_{m,s,d} n^{-2(m-s-d/2)}. \tag{32}$$

*where $C_{m,s,d} = \frac{2^d \cdot d}{2(m-s)-1}$.*

*Proof.* Since $\|\mathbf{x}\|_\infty = \max_{i=1,\cdots,d} \left| \mathbf{x}|_{(i)} \right|$, thus in $d$-dimensional space, we can rewrite $\|\mathbf{x}\|_\infty > n$ as follows

$$\|\mathbf{x}\|_\infty > n = \bigcup_{i=1}^{d} \{\mathbf{x} : \left| \mathbf{x}|_{(i)} \right| > n, \left| \mathbf{x}|_{(j)} \right| \leq n \text{ for } j \neq i\}. \tag{33}$$

By symmetry, the contribution from each region where $\left|\mathbf{x}\right|_{(i)} > n$ is identical, specifically

$$\int_{\|\mathbf{x}\|_\infty > n} |\mathbf{x}|^{-2(m-s)}\mathrm{d}\mathbf{x} = d\int_{\left|\mathbf{x}\right|_{(1)} > n}\int_{-n}^{n}\cdots\int_{-n}^{n}|\mathbf{x}|^{-2(m-s)}. \tag{34}$$

It is obvious that $|\mathbf{x}| > \left|\mathbf{x}\right|_{(1)}$, thus the integral can be bounded and directly calculated as follows:

$$\begin{aligned}
&d\int_{|\mathbf{x}_1| > n}\int_{-n}^{n}\cdots\int_{-n}^{n}|\mathbf{x}|^{-2(m-s)}\mathrm{d}\mathbf{x}_d\cdots\mathrm{d}\mathbf{x}_2\mathrm{d}\mathbf{x}_1\\
\leq&d\int_{|\mathbf{x}_1| > n}\int_{-n}^{n}\cdots\int_{-n}^{n}|\mathbf{x}_1|^{-2(m-s)}\mathrm{d}\mathbf{x}_d\cdots\mathrm{d}\mathbf{x}_2\mathrm{d}\mathbf{x}_1\\
=&d(2n)^{d-1}\int_{|\mathbf{x}_1| > n}|\mathbf{x}_1|^{-2(m-s)}\mathrm{d}\mathbf{x}_1\\
=&2d(2n)^{d-1}\int_n^\infty \mathbf{x}_1^{-2(m-s)}\mathrm{d}\mathbf{x}_1\\
=&2d(2n)^{d-1}\frac{n^{1-2(m-s)}}{2(m-s)-1}\\
=&\frac{2^d\cdot d}{2(m-s)-1}n^{-2(m-s-d/2)},
\end{aligned} \tag{35}$$

where $\mathbf{x}_i$ denotes $\mathbf{x}|_{(i)}$ for simplicity. The proof is completed by combining equation 34 and equation 35. $\qquad\square$

**Lemma 8** (The approximation lemma of the Fourier basis). *Let $h \in \mathbf{H} \subseteq H^s(\mathbb{T}^d; \mathbb{R})$, where $\mathbb{T}^d = [0, 2\pi]^d$. Its Fourier series expansion is given by:*

$$h(\mathbf{x}) = \sum_{\mathbf{k}\in\mathbb{Z}^d} c_\mathbf{k}e^{i\mathbf{k}\cdot\mathbf{x}},\ c_\mathbf{k} = \frac{1}{(2\pi)^d}\int_{\mathbb{T}^d}h(\mathbf{x})e^{-i\mathbf{k}\cdot\mathbf{x}}\,\mathrm{d}\mathbf{x}, \tag{36}$$

*Let the truncated Fourier series be*

$$P_n(h)(\mathbf{x}) = \sum_{\|\mathbf{k}\|_\infty \leq n} c_k e^{i\mathbf{k}\cdot\mathbf{x}}, \tag{37}$$

*and define the remainder as $R_n(h)(\mathbf{x}) := h(\mathbf{x}) - P_n(h)(\mathbf{x})$. Thus we can conclude that*

$$\lim_{n\to+\infty}\|R_n(h)\|_{H^s}^2 = 0. \tag{38}$$

*Proof.* We have

$$\begin{aligned}
\|R_n(h)\|_{H^s}^2 &= \sum_{\|\mathbf{k}\|_\infty > n}(1+|\mathbf{k}|^2)^s|c_\mathbf{k}|^2\\
&\leq \binom{s}{\lfloor s/2\rfloor}\sum_{\|\mathbf{k}\|_\infty > n}\left(\sum_{|\beta|\leq s}|\mathbf{k}|^{2|\beta|}\right)|c_\mathbf{k}|^2\\
&\leq \binom{s}{\lfloor s/2\rfloor}(d+1)^s\sum_{\|\mathbf{k}\|_\infty > n}|\mathbf{k}|^{2s}|c_\mathbf{k}|^2\\
&= \binom{s}{\lfloor s/2\rfloor}C_m^2(d+1)^s\sum_{\|\mathbf{k}\|_\infty > n}|\mathbf{k}|^{-2(m-s)},
\end{aligned} \tag{39}$$

where the first inequality follows from Lemma 4, the second inequality follows from Lemma 5 and the last inequality follows from Lemma 6.

The series $\sum_{k\in\mathbb{Z}^d}|k|^{-2(m-s)}$ converges if $m > s + d/2$. According to Lemma 7, we further estimate:

$$\sum_{\|\mathbf{k}\|_\infty > n}|\mathbf{k}|^{-2(m-s)} \leq \int_{\|\mathbf{x}\|_\infty > n}|\mathbf{x}|^{-2(m-s)} \leq C_{m,s,d}n^{-2(m-s-d/2)}. \tag{40}$$

Thus we have

$$\|R_n(h)\|_{H^s} \le M_{m,s,d} n^{-(m-s-d/2)}, \tag{41}$$

where $M_{m,s,d} = \sqrt{\binom{s}{\lfloor s/2 \rfloor}} \sqrt{C_{m,s,d}} C_m (d+1)^{s/2}$.

For any $\epsilon > 0$, exists $N = s + d/2 + \lceil \log_2(M_{m,s,d}/\epsilon) \rceil$, such that

$$\|R_n\|_{H^s} \le \epsilon. \tag{42}$$

Therefore, the proof is completed. $\qquad\square$

**Lemma 9** (The relation between $L_\infty$ norm and the $H^s$ norm)**.** *Let $s > 0$, and a compact set $\mathcal{F} \subset H^s(\Omega; \mathbb{R})$ where $\Omega \in \mathbb{R}^d$ is a compact subset. Here, $\forall f \in \mathbb{F}$ $f$ is a analytic function on $\Omega$. There exists $A_s > 0$, such that*

$$\|f\|_{H^s} \le A_s \sup_{\mathbf{x} \in \Omega} |f(\mathbf{x})|. \tag{43}$$

*Proof.* From $f$ being analytic in compact set $\Omega$ and the Cauchy inequality, there exist constant $C, M > 0$ that are independent of $f$, such that

$$\|D^\alpha f(\mathbf{x})\|_\infty \le C M^{|\alpha|} |\alpha|!. \tag{44}$$

Because $\Omega$ is a compact set, we can define $A := \int_\Omega 1 \mathrm{d}x > 0$. Then, we have

$$\|f\|_{\ell_2} \le \sqrt{A} \|f\|_\infty \le \sqrt{A} C M^{|\alpha|} |\alpha|!. \tag{45}$$

For $\|f\|_{H^s}$, we have

$$\|f\|_{H^s}^2 = \sum_{|\alpha| \le s} \|D^\alpha f\|_{\ell_2}^2 \le A \sum_{|\alpha| \le s} (C M^{|\alpha|} |\alpha|!)^2. \tag{46}$$

Because $\mathcal{F}$ is a compact set and $f$ is analytic, there exists a constant $A_s > 0$, such that

$$\sum_{|\alpha| \le s} (C M^{|\alpha|} |\alpha|!)^2 \le A_s^2 \|f\|_\infty. \tag{47}$$

Therefore, we have

$$\|f\|_{H^s} \le \sqrt{A} A_s \|f\|_\infty. \tag{48}$$

The proof is completed. $\qquad\square$

## C  PROOF OF UNIVERSAL APPROXIMATION THEOREM OF FUNCTIONAL TUCKER DECOMPOSITION (THEOREM 1)

***Proof of Theorem 1.*** Due to $\Omega$ is a compact set, the Heine-Cantor theorem is valid. So, $f$ being continuous on $\Omega$ implies uniform continuity, i.e., for any $\epsilon > 0$, there exists $\delta > 0$, such that

$$\|\mathbf{x} - \mathbf{x}'\| < \delta \Rightarrow \|f(\mathbf{x}) - f(\mathbf{x}')\| < \epsilon. \tag{49}$$

Therefore, from the Stone-Weierstrass Theorem, for any $\epsilon > 0$, there exists polynomial $P(\mathbf{x}) = \sum_{i_1=0}^{r_1-1} \cdots \sum_{i_d=0}^{r_d-1} a_{i_1,\cdots,i_d} \Pi_{j=1}^d \mathbf{x}^{i_j}|_{(j)}$, such that

$$\sup_{\mathbf{x} \in \Omega} |f(\mathbf{x}) - P(\mathbf{x})| < \epsilon. \tag{50}$$

Define index sets $\mathcal{I}_i := \{0, 1, \cdots, r_i - 1\}$, core tensor $\mathcal{C} \in \mathbb{R}^{r_1 \times \cdots \times r_d}$, and basis functions $g_i(x) := [x^0, x^1, \cdots, x^{r_i-1}]^T$. Therefore, $P$ can be rewritten as

$$P(\mathbf{x}) = \mathcal{C} \times_1 g_1(\mathbf{x}_1) \times_2 \cdots \times_d g_d(\mathbf{x}_d). \tag{51}$$

Then, from equation 50, we have

$$\sup_{\mathbf{x} \in \Omega} |f(\mathbf{x}) - \mathcal{C} \times_1 g_1(\mathbf{x}|_{(1)}) \times_2 \cdots \times_d g_d(\mathbf{x}|_{(d)})| < \epsilon. \tag{52}$$

The proof is completed.

# D    PROOF OF UNIVERSAL APPROXIMATION THEOREM OF OPERATOR TUCKER DECOMPOSITION IN FINITE-DIMENSIONAL FUNCTION SPACE (FOR THEOREM 2)

**Definition 2.** *Let $d$ be the number of variables and $n$ the maximum Fourier degree. The basis functions are $\phi_\alpha(\mathbf{x}) = e^{i\mathbf{k}\cdot\mathbf{x}}$ for all multi-indices $\mathbf{k} = (\mathbf{k}|_{(1)}, \mathbf{k}|_{(2)}, \cdots, \mathbf{k}|_{(d)}) \in \mathbb{Z}^d$ with $\|\mathbf{k}\|_\infty \leq n$, then we can define*

$$\mathbf{A}_n = span\{\phi_{\mathbf{k}}(\mathbf{x}) | \|\mathbf{k}\|_\infty \leq n\}. \tag{53}$$

*For any coordinate vector $\mathbf{a} := (\mathbf{a}|_{(\mathbf{k})})_{\|\mathbf{k}\|_\infty \leq n} \in \mathbb{R}^N$, the mapping $\mathscr{H} : \mathbb{R}^N \to \mathbf{A}_n$ (where $N = \sharp\{\phi_{\mathbf{k}}(\mathbf{x}) | \|\mathbf{k}\|_\infty \leq n\} = (2n+1)^d$) is defined as*

$$\mathscr{H}(\mathbf{a})(\mathbf{x}) = \sum_{\|\mathbf{k}\|_\infty \leq n} \mathbf{a}|_{(\mathbf{k})} \phi_{\mathbf{k}}(\mathbf{x}), \tag{54}$$

*where the basis is ordered by ascending total degree, and lexicographically within the same degree. Each component $\mathbf{a}|_{(\mathbf{k})}$ corresponds uniquely to the basis $\phi_{\mathbf{k}}(\mathbf{x})$.*

**Lemma 10** (Universal approximation theorem of Tucker operator decomposition finite-dimensional function space). *Let $\mathbf{H} \subseteq H^s(\mathbb{T}^d; \mathbb{R})$ be a compact subset with $s > \frac{d}{2}$ and define $\mathbf{A}_n = span\{\phi_{\mathbf{k}}(\mathbf{x}) | \|\mathbf{k}\|_\infty \leq n\}$. Assume that all $h \in \mathbf{H}$ are analytic on $\mathbb{T}^d$. Then for all $h \in \mathbf{H}$, $\epsilon > 0$, there exists ranks $(r_1, \cdots, r_d) \in \mathbb{N}^d$, continuous operators $\mathscr{U}_i : \mathbf{A}_n \cap \mathbf{H} \to H^s(\mathbb{T}^d; \mathbb{R}^{r_i})$ and a core tensor $\mathcal{C} \in \mathbb{R}^{r_1 \times r_2 \times \cdots \times r_d}$, such that*

$$\sup_{\mathbf{h} \in \mathbf{H}} \|h(\cdot) - \mathcal{C} \times_1 \mathscr{U}_1 h(\cdot_1) \times_2 \cdots \times_d \mathscr{U}_d h(\cdot_d)\|_{H^s} < \epsilon. \tag{55}$$

*Proof.* Define the basis function of $\mathbf{A}_n$ is $\{\phi_1, \cdots, \phi_N\}$, i.e., $\mathbf{A}_n = span\{\phi_1, \cdots, \phi_N\}$. From Theorem 1, for $\epsilon > 0$, there exist tucker rank $(r_1, \cdots, r_d)$, the core tensor $\mathcal{C}^{(i)} \in \mathbb{R}^{r_1 \times \cdots \times r_d}$, such that

$$|\phi_i(\mathbf{x}) - \mathcal{C}^{(i)} \times_1 g_1^{(i)}(\mathbf{x}|_{(1)}) \times_2 \cdots \times_d g_d^{(i)}(\mathbf{x}|_{(d)})| \leq \frac{\epsilon}{N} \tag{56}$$

For $h \in \mathbf{A}_n$, we have $h = \sum_{i=1}^N a_i^{(h)} \phi_i$, where $\phi_i(\mathbf{x}) := e^{-i\mathbf{k}_i \cdot \mathbf{x}}$, and $a_i^{(h)} := < h, \phi_i >$ is continuous for $h$.

Therefore, define

$$
\begin{aligned}
\mathcal{C} &:= \text{diag}\{\mathcal{C}^{(1)}, \cdots, \mathcal{C}^{(N)}\}, \\
\mathscr{U}(h)(\mathbf{x}_i) &:= [\sqrt[d]{a_1^{(h)}} g_1^{(i)}(\mathbf{x}|_{(1)}), \cdots, \sqrt[d]{a_d^{(h)}} g_d^{(i)}(\mathbf{x}|_{(d)})]^T,
\end{aligned} \tag{57}
$$

we have

$$
\begin{aligned}
&|h(\mathbf{x}) - \mathcal{C} \times_1 \mathscr{U}_1(h)(\mathbf{x}|_{(1)}) \times_2 \cdots \times_d \mathscr{U}_d(\mathbf{x}|_{(d)})| \\
=&|\sum_{i=1}^N a_i^{(h)} \phi_i(\mathbf{x}) - \sum_{i=1}^N a_i^{(h)} \mathcal{C}^{(i)} \times_1 g_1^{(i)}(\mathbf{x}|_{(1)}) \times_2 \cdots \times_d g_d^{(i)}(\mathbf{x}|_{(d)})| \\
=&|\sum_{i=1}^N a_i^{(h)} (\phi_i(\mathbf{x}) - \mathcal{C}^{(i)} \times_1 g_1^{(i)}(\mathbf{x}|_{(1)}) \times_2 \cdots \times_d g_d^{(i)}(\mathbf{x}|_{(d)}))| \\
\leq& \sum_{i=1}^N a_i^{(h)} |\phi_i(\mathbf{x}) - \mathcal{C}^{(i)} \times_1 g_1^{(i)}(\mathbf{x}_{(1)}) \times_2 \cdots \times_d g_d^{(i)}(\mathbf{x}|_{(d)})| \\
\leq& \sum_{i=1}^N a_i^{(h)} \frac{\epsilon}{N}.
\end{aligned} \tag{58}
$$

For $\mathbf{H}$ is a compact set and $a_i^{(h)} = < h, \phi_i >$, there exist constant $A > 0$, such that $a_i^{(h)} \leq A$. Therefore, from equation 58, we have

$$|h(\mathbf{x}) - \mathcal{C} \times_1 \mathscr{U}_1(h)(\mathbf{x}|_{(1)}) \times_2 \cdots \times_d \mathscr{U}_d(\mathbf{x}|_{(d)})| \leq A\epsilon, \tag{59}$$

i.e., $\|h(\cdot) - \mathcal{C} \times_1 \mathscr{U}_1(h)(\cdot_1) \times_2 \cdots \times_d \mathscr{U}_d(\cdot_d)\|_\infty \leq A\epsilon$.

From Lemma 9, there exists $A_s$, such that $\|f\|_{H^s} < A_s\|f\|_\infty$. Then, we have

$$
\begin{aligned}
&\sup_{\mathbf{h}\in\mathcal{H}} \|h(\cdot) - \mathcal{C} \times_1 \mathscr{U}_1 h(\cdot_1) \times_2 \cdots \times_d \mathscr{U}_d h\|_{H^s} \\
&\leq A_s \sup_{\mathbf{h}\in\mathcal{H}} \|h(\cdot) - \mathcal{C} \times_1 \mathscr{U}_1 h(\cdot_1) \times_2 \cdots \times_d \mathscr{U}_d h\|_\infty \\
&\leq A_s A\epsilon.
\end{aligned}
\tag{60}
$$

Therefore, the proof is completed. $\qquad\square$

## E  PROOF OF UNIVERSAL APPROXIMATION THEOREM OF FUNCTIONAL TUCKER DECOMPOSITION (THEOREM 2)

***Proof of Theorem 2.*** From Lemma. 8, for all $\epsilon > 0$, there exists $n$ such that

$$
\|h - P_n(h)\|_{H^s} \leq \frac{\epsilon}{2}, \ \forall h \in \mathbf{H},
\tag{61}
$$

where $P_n$ is define in equation 37.

From Lemma. 10, there exist continuous operators $\tilde{\mathscr{U}}_i : \mathbf{A}_n \to H^s(\mathbb{T}^d; \mathbb{R}^{r_i})$ and core tensor $\mathcal{C} \in \mathbb{R}^{r_1 \times \cdots \times r_d}$, such that

$$
\sup_{h_n \in \mathbf{A}_n} \|h_n(\mathbf{x}) - \mathcal{C} \times_1 \tilde{\mathscr{U}}_1 h_n(\mathbf{x}|_{(1)}) \times_2 \cdots \times_d \tilde{\mathscr{U}}_d h_n(\mathbf{x}|_{(d)})\|_{H^s} \leq \frac{\epsilon}{2}.
\tag{62}
$$

We define $\mathcal{U}_i := \tilde{\mathscr{U}}_i \circ P_n$. For $P_n(h)(\mathbf{x}) = \sum_{\|k\|_\infty \leq n} c_k e^{i\mathbf{k}\cdot\mathbf{x}}$ and $c_\mathbf{k} = \frac{1}{(2\pi)^d} \int_{\mathbb{T}^d} h(\mathbf{x}) e^{-i\mathbf{k}\cdot\mathbf{x}} \, d\mathbf{x}$, we have $c_\mathbf{k}$ is continuous for $h$ and $P_n$ is also continuous for $h$, i.e., $P_n$ is also a continuous operator. Therefore, From the continuity of $\tilde{\mathscr{U}}_i$ and $P_n$, $\mathcal{U}_i = \tilde{\mathscr{U}}_i \circ P_n$ is also a continuous operator.

For all $h \in \mathbf{H}$, we have

$$
\begin{aligned}
&\|h(\cdot) - \mathcal{C} \times_1 \mathscr{U}_1 h(\cdot_1) \times_2 \cdots \times_d \mathscr{U}_d h(\cdot_d)\|_{H^s} \\
&= \|h(\cdot) - \mathcal{C} \times_1 \tilde{\mathscr{U}}_1(P_n h(\cdot_1)) \times_2 \cdots \times_d \tilde{\mathscr{U}}_d(P_n h(\cdot_d))\|_{H^s} \\
&\leq \|h - P_n(h)\|_{H^s} + \|P_n(h)(\cdot) - \mathcal{C} \times_1 \tilde{\mathscr{U}}_1(P_n h(\cdot_1)) \times_2 \cdots \times_d \tilde{\mathscr{U}}_d(P_n h(\cdot_d))\|_{H^s} \\
&\leq \epsilon,
\end{aligned}
\tag{63}
$$

i.e.,

$$
\sup_{h\in\mathbf{H}} \|h(\cdot) - \mathcal{C} \times_1 \mathscr{U}_1 h(\cdot_1) \times_2 \cdots \times_d \mathscr{U}_d h(\cdot_d)\|_{H^s} \leq \epsilon.
\tag{64}
$$

Therefore, we have continuous operators $\mathscr{U}_i$ satisfying the lemma.

## F  PROOF OF UNIVERSAL APPROXIMATION THEOREM OF TUCKER-FNO (THEOREM 3)

***Proof of Theorem 3.*** For $\mathscr{G}$ is continuous and $\mathbf{F}$ is a compact subset, $\mathbf{H}$ is also a compact subset on $H^{s'}(\mathbb{T}^d; \mathbb{R})$. Therefore, from Theorem 2 and the constraint of FS-rank, for all $\epsilon > 0$, there exist $\hat{\mathscr{U}}_i : \mathbf{H} \to H^{s'}(\mathbb{T}; \mathbb{R})$ and the core tensor $\mathcal{C} \in \mathbb{R}^{r_1 \times \cdots \times r_d}$ satisfying

$$
\|h(\cdot) - \mathcal{C} \times_1 \hat{\mathscr{U}}_1 h(\cdot_1) \times_2 \cdots \times_d \hat{\mathscr{U}}_d h(\cdot_d)\|_{H^s} \leq \epsilon_1, \forall h \in \mathbf{H}.
\tag{65}
$$

Therefore, we define $\mathscr{U}_i := \hat{\mathscr{U}}_i \circ \mathscr{G}$, such that

$$
\begin{aligned}
&\|\mathscr{G}f(\cdot) - \mathcal{C} \times_1 \mathscr{U}_1 f(\cdot_1) \times_2 \cdots \times_d \mathscr{U}_d f(\cdot_d)\|_{H^s} \\
&= \|\mathscr{G}f(\cdot) - \mathcal{C} \times_1 \hat{\mathscr{U}}_1 \mathscr{G}f(\cdot_1) \times_2 \cdots \times_d \hat{\mathscr{U}}_d \mathscr{G}f(\cdot_d)\|_{H^s} \\
&\leq \frac{\epsilon}{2}.
\end{aligned}
\tag{66}
$$

Because $\mathscr{U}_i$ are continuous operator define in the compact set $\mathbf{F}$, there exist $C$, such that

$$
\sup_{f\in\mathbf{F}} \|\mathscr{U}_i(f)\|_{H^s} \leq C.
\tag{67}
$$

From Lemma 2, there exist continuous FNOs $\mathscr{N}_i : \mathbf{F} \to H^{s'}(\mathbb{T}^d; \mathbb{R}^{r_i})$, such that

$$\|\mathscr{U}_i(f) - \mathscr{N}_i(f)\|_{H^s} \leq \epsilon_2, \quad \forall \epsilon_2 > 0. \tag{68}$$

From equation 67, we have

$$\sup_{f \in \mathbf{F}} \|\mathscr{N}_i(f)\|_{H^s} \leq C + \epsilon_2. \tag{69}$$

Therefore, we have

$$
\begin{aligned}
&\|\mathcal{C} \times_1 \mathscr{U}_1 f \times_2 \cdots \times_d \mathscr{U}_d f - \mathcal{C} \times_1 \mathscr{N}_1 f \times_2 \cdots \times_d \mathscr{N}_d f\|_{H^s} \\
\leq\ & \|\mathcal{C} \times_1 \mathscr{U}_1 f \times_2 \cdots \times_d \mathscr{N}_d f - \mathcal{C} \times_1 \mathscr{U}_1 f \times_2 \cdots \times_{d-1} \mathscr{U}_{d-1} f \times_d \mathscr{N}_d f\|_{H^s} \\
&+ \|\mathcal{C} \times_1 \mathscr{U}_1 f \times_2 \cdots \times_d \mathscr{N}_d f - \mathcal{C} \times_1 \mathscr{U}_1 f \times_2 \cdots \times_{d-1} \mathscr{N}_{d-1} f \times_d \mathscr{N}_d f\|_{H^s} \\
&\cdots \\
&+ \|\mathcal{C} \times_1 \mathscr{U}_1 f \times_2 \mathscr{U}_2 f \times_2 \cdots \times_d \mathscr{N}_d f - \mathcal{C} \times_1 \mathscr{N}_1 f \times_2 \cdots \times_d \mathscr{N}_d f\|_{H^s} \\
=\ & \|\mathcal{C} \times_1 \mathscr{U}_1 f \times_2 \cdots \times_d (\mathscr{U}_d - \mathscr{N}_d) f\|_{H^s} \\
&+ \|\mathcal{C} \times_1 \mathscr{U}_1 f \times_2 \cdots \times_{d-1} (\mathscr{U}_{d-1} - \mathscr{N}_{d-1}) f \times_d \mathscr{N}_d f\|_{H^s} \\
&\cdots \\
&+ \|\mathcal{C} \times_1 (\mathscr{U}_1 - \mathscr{N}_1) f \times_2 \mathscr{U}_2 f \times_2 \cdots \times_d \mathscr{N}_d f\|_{H^s}
\end{aligned} \tag{70}
$$

By defining

$$\mathscr{T}_i f := \mathcal{C} \times_1 \mathscr{U}_1 f \times_2 \cdots \times_{i-1} \mathscr{U}_{i-1} f \times_{i+1} \mathscr{N}_{i+1} f \times_{i+2} \cdots \times_d \mathscr{N}_d f, \tag{71}$$

we have

$$
\begin{aligned}
&\|\mathcal{C} \times_1 \mathscr{U}_1 f \times_2 \cdots \times_d \mathscr{U}_d f - \mathcal{C} \times_1 \mathscr{N}_1 f \times_2 \cdots \times_d \mathscr{N}_d f\|_{H^s} \\
\leq\ & \|\mathscr{T}_d f \times (\mathscr{U}_d - \mathscr{N}_d) f\|_{H^s} + \|\mathscr{T}_{d-1} f \times (\mathscr{U}_{d-1} - \mathscr{N}_{d-1}) f\|_{H^s} \cdots + \|\mathscr{T}_1 f \times (\mathscr{U}_1 - \mathscr{N}_1) f\|_{H^s},
\end{aligned} \tag{72}
$$

From Sobolev Embedding Theorem and Kato-Ponce Inequality, for $f_1, f_2 \in H^s(\mathbb{T}^d; \mathbb{R}^r)$, there exists $C_{f_1, f_2} > 0$, such that,

$$\|f_1 \times f_2\|_{H^s} \leq C_{f_1, f_2} \|f_1\|_{H^s} \|f_2\|_{H^s}. \tag{73}$$

Therefore, there exists $\hat{C} > 0$, such that

$$\|\mathscr{T}_i f \times (\mathscr{U}_i f - \mathscr{N}_i f)\|_{H^s} \leq \hat{C} \|\mathscr{T}_i f\|_{H^s} \|\mathscr{U}_i f - \mathscr{N}_i f\|_{H^s}. \tag{74}$$

For each $\mathscr{T}_i$, we have

$$
\begin{aligned}
&\|\mathscr{T}_i f\|_{H^s} \\
=& \|\mathcal{C} \times_1 \mathscr{U}_1 f \times_2 \cdots \times_{i-1} \mathscr{U}_{i-1} f \times_{i+1} \mathscr{N}_{i+1} f \times_{i+2} \cdots \times_d \mathscr{N}_d f\|_{H^s} \\
\leq& \tilde{C} \|\mathscr{U} f\|_{H^s} \cdots \|\mathscr{U}_{i-1} f\|_{H^s} \|\mathscr{U}_{i+1} f\|_{H^s} \cdots \|\mathscr{N}_d f\|_{H^s} \\
\leq& \tilde{C} C^{i-1} (C + \epsilon_2)^{d-i} \\
\leq& \tilde{C} (C + \epsilon_2)^{d-1},
\end{aligned} \tag{75}
$$

where $\tilde{C} > 0$ is existed from the analysis from equation 73.

Therefore, we have

$$
\begin{aligned}
&\|\mathcal{C} \times_1 \mathscr{U}_1 f \times_2 \cdots \times_d \mathscr{U}_d f - \mathcal{C} \times_1 \mathscr{N}_1 f \times_2 \cdots \times_d \mathscr{N}_d f\|_{H^s} \\
\leq& \hat{C}^d \Pi_{i=1}^d \|\mathscr{T}_i f\|_{H^s} \|\mathscr{U}_i f - \mathscr{N}_i f\|_{H^s} \\
\leq& \hat{C}^d \tilde{C}^d (C + \epsilon_2)^{d(d-1)} \epsilon_2^d.
\end{aligned} \tag{76}
$$

Obviously, when $\epsilon_2 \to 0$, we have $\hat{C}^d \tilde{C}^d (C + \epsilon_2)^{d(d-1)} \epsilon_2^d \to 0$. Therefore, for all $\epsilon > 0$, there exist continuous FNOs $\mathscr{N}_i : \mathbf{F} \to H^{s'}(\mathbb{T}^d; \mathbb{R}^{r_i})$, such that

$$\|\mathcal{C} \times_1 \mathscr{U}_1 f \times_2 \cdots \times_d \mathscr{U}_d f - \mathcal{C} \times_1 \mathscr{N}_1 f \times_2 \cdots \times_d \mathscr{N}_d f\|_{H^s} \leq \frac{\epsilon}{2}. \tag{77}$$

Then, from equation 66 and equation 77, we have

$$
\begin{aligned}
&\|\mathscr{G}f(\cdot) - \mathcal{C} \times_1 \mathscr{N}_1 f(\cdot_1) \times_2 \cdots \times_d \mathscr{N}_d f(\cdot_d)\|_{H^s} \\
&\leq \|\mathscr{G}f(\cdot) - \mathcal{C} \times_1 \mathscr{U}_1 f(\cdot_1) \times_2 \cdots \times_d \mathscr{U}_d f(\cdot_d)\|_{H^s} \\
&\quad + \|\mathcal{C} \times_1 \mathscr{U}_1 f(\cdot_1) \times_2 \cdots \times_d \mathscr{U}_d f(\cdot_d) - \mathcal{C} \times_1 \mathscr{N}_1 f(\cdot_1) \times_2 \cdots \times_d \mathscr{N}_d f(\cdot_d)\|_{H^s} \\
&\leq \frac{\epsilon}{2} + \frac{\epsilon}{2} = \epsilon
\end{aligned}
\tag{78}
$$

Therefore, we have Tucker-FNO

$$
\mathscr{N}(f)(\mathbf{x}) := \mathcal{C} \times_1 \mathscr{N}_1 f(\mathbf{x}|_{(1)}) \times_2 \cdots \times_d \mathscr{N}_d f(\mathbf{x}|_{(d)}),
\tag{79}
$$

such that,

$$
\|\mathscr{G}(f) - \mathscr{N}(f)\| \leq \epsilon.
\tag{80}
$$

The proof is completed.

## G  THE SETTINGS OF PDE

For PDE approximation task, we evaluate Tucker-FNO in three PDEs, e.g., Navier-Stokes, Plasticity and Burger's equations.

The Navier-Stokes equation for a viscous, incompressible fluid in vorticity form on the unit torus:

$$
\begin{aligned}
\frac{\partial w(\mathbf{x}, t)}{\partial t} + u(\mathbf{x}, t) \cdot \nabla w(\mathbf{x}, t) &= \mu \Delta w(\mathbf{x}, t) + f(\mathbf{x}), & \mathbf{x} \in (0,1)^2, t \in [0, T], \\
\nabla \cdot u(\mathbf{x}, t) &= 0, & \mathbf{x} \in (0,1)^2, t \in [0, T], \\
w(\mathbf{x}, 0) &= w_0(\mathbf{x}), & \mathbf{x} \in (0,1)^2,
\end{aligned}
\tag{81}
$$

where $u$ is the velocity field, $w = \nabla \times u$ is the vorticity, $w_0$ is the initial vorticity and $\mu$ is the viscosity coefficient. $f$ is the force term, which is fixed as $f(x) = 0.1(\sin(2\pi(x_1 + x_2)) + \cos(2\pi(x_1 + x_2)))$. The initial vorticity $w_0$ follows the Gaussian random field with a covariance decay of 2.5 and a scale parameter of 7. The viscosity coefficient is set as $1e-5$. Following FNO (Li et al., 2021), we are interested in learning the operator mapping from the vorticity up to time 10 to the vorticity up to some later time $T > 10$, i.e., $\mathscr{G} : w|_{(0,1)^2 \times [0,10]} \mapsto w|_{(0,1)^2 \times (10,T]}$, and we fix the sample resolution as $64 \times 64$. The data are from (Li et al., 2021), and we utilize 1000 samples for training and 200 samples for testing.

The Plasticity equation is a non-linear PDE for the Plastic forging problem, where a block of material is impacted by a frictionless, rigid die at time $t = 0$. The governing equation is the same as the previous example given by

$$
\begin{aligned}
\rho^s \frac{\partial^2 u}{\partial t^2} + \nabla \cdot \sigma &= 0 \\
\sigma &= C : (\epsilon - \epsilon_p) \\
\dot{\epsilon_p} &= \lambda \nabla_\sigma f(\sigma) \\
f(\sigma) &= \sqrt{\frac{3}{2}} |\sigma - \frac{1}{3} tr(\sigma) \cdot I|_F - \sigma_Y,
\end{aligned}
\tag{82}
$$

where the PDE detail setting is the same as Geo-FNO(Li et al., 2021). To validate the efficiency and performance of Tucker-FNO, we extend the 1-dimensional input to 2-dimensions by duplicating the input to align it with the output, which is different from (Li et al., 2021) Our dataset is also from (Li et al., 2021), and we utilize 900 samples for training and 80 samples for testing.

The Burger's equation is a non-linear PDE to model the 1-dimensional flow of viscous fluid, which is formed as

$$
\begin{aligned}
\frac{\partial u(x, t)}{\partial t} + u(x, t) \frac{\partial u(x, t)}{\partial x} &= \mu \frac{\partial^2 u(x, t)}{\partial x^2}, & x \in (0, 1), t \in (0, 1], \\
u(x, 0) &= u_0(x), & x \in (0, 1),
\end{aligned}
\tag{83}
$$

where $u_0$ is the initial condition and $\mu = 5e-3$ is the viscosity coefficient. Following FNO (Li et al., 2021), we are interested in the operator mapping from the initial condition $u_0$ to the target solution $u$, i.e., $\mathscr{G} : u_0 \mapsto u$. The dataset is from (Li et al., 2021), and we utilize 100 samples for training, and 20 samples for testing.

## H  MORE NUMERICAL EXPERIMENTS

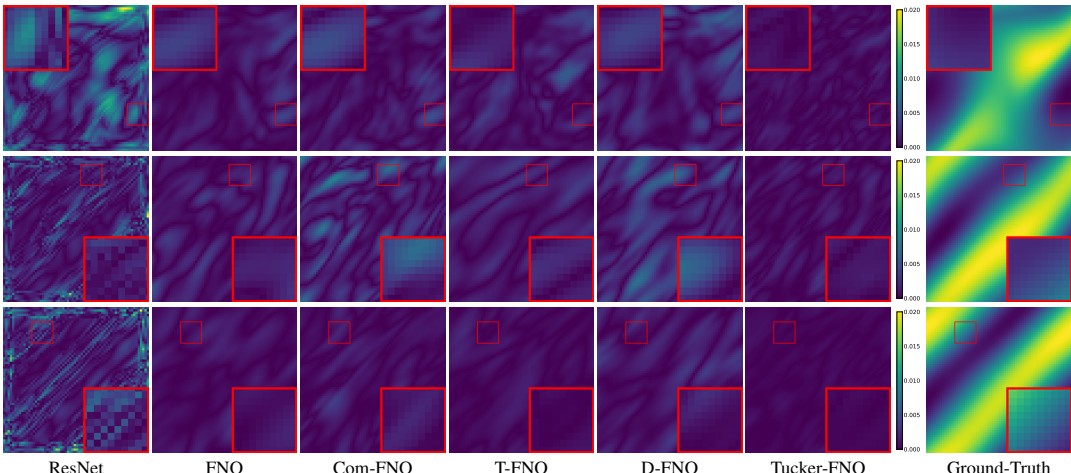

Figure 6: Error map comparisons of different NO methods for the Navier-Stokes equation.

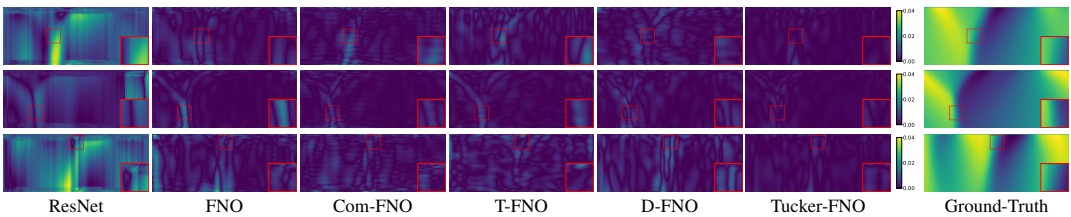

Figure 7: Error map comparisons of different NO methods for the Burger's equation.

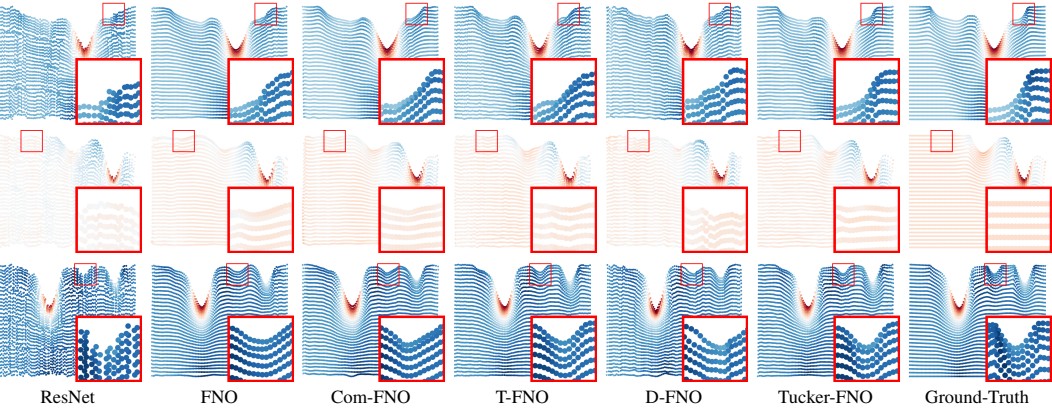

Figure 8: Qualitative comparisons of different NO methods for the Plasticity equation.

***More results of PDE approximation task.*** Here, we demonstrate more qualitative results of the Navier-Stokes, Plasticity, and Burger's equation in Fig.6-8. These qualitative results further demonstrate the superiority of our Tucker-FNO in regions containing high-frequency signals.

***More results of large-scale PDE approximation task.*** To investigate the influence of decomposition on FNO for large-scale PDE problems with complex geometry, we replace the FNO in GINO (Li et al., 2023b) with our Tucker-FNO, and evaluate it in Car-CFD task (Li et al., 2023b), which is a 3-dimensional Reynold-Averaged Navier-Stokes PDE problem with velocity $20m/s$. The dataset takes 611 weight-tight shapes out of the 889 instances, and divides the 611 instances into 500 for training and 111 for validation. For Car-CFD task, the goal is to estimate the full pressure field given

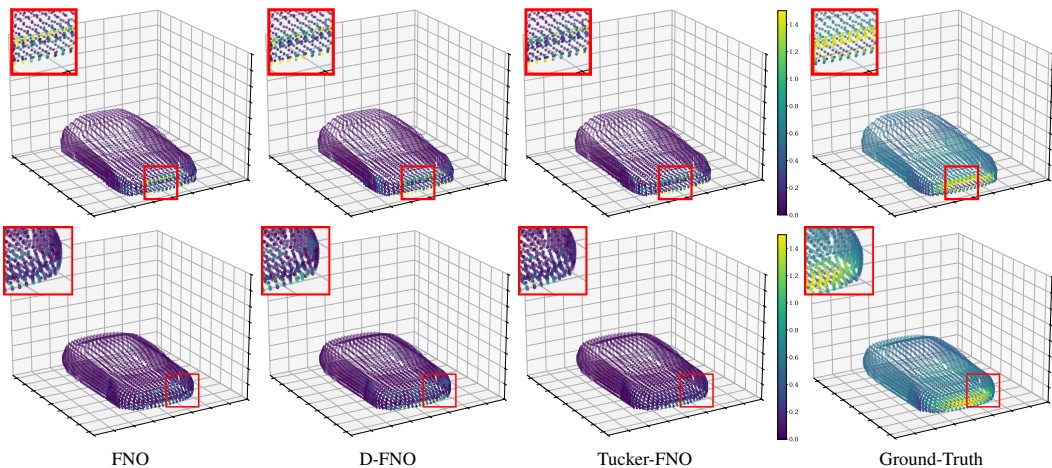

| FNO | D-FNO | Tucker-FNO | Ground-Truth |

Figure 9: Error map comparisons of different NO methods for the Car-CFD. The first row demonstrates the results with $\mathbf{k_{max}} = (12, 12, 12)$, and the second row demonstrates the results with $\mathbf{k_{max}} = (16, 16, 16)$.

Table 8: The average quantitative results (PSNR ↑ (dB), SSIM ↑, NMSE ↓) by different methods for multi-spectral image denoising. The **best** and second-best values are highlighted.

| Noise level | | 0.1 | | | 0.15 | | | 0.2 | | |
|---|---|---|---|---|---|---|---|---|---|---|
| Data | Metric | PSNR | SSIM | NMSE | PSNR | SSIM | NMSE | PSNR | SSIM | NMSE |
| MSIs *Ballons* *Fruits* *Toys* (256×256×31) | Observed | 20.00 | 0.313 | 0.922 | 16.48 | 0.202 | 0.940 | 13.98 | 0.140 | 0.953 |
| | SIREN | 26.23 | 0.821 | 0.173 | 25.92 | 0.795 | 0.180 | 25.82 | 0.730 | 0.185 |
| | MFN | 28.43 | 0.833 | 0.151 | 27.23 | 0.794 | 0.163 | 26.14 | 0.779 | 0.170 |
| | LRTFR | 30.60 | 0.896 | 0.139 | 29.51 | 0.866 | 0.144 | 28.63 | 0.839 | 0.150 |
| | OINR | 30.48 | 0.880 | 0.140 | 29.14 | 0.861 | 0.150 | 28.27 | 0.821 | 0.155 |
| | OINR-FNO | 31.19 | 0.909 | 0.122 | 29.40 | 0.880 | 0.134 | 28.94 | 0.838 | 0.142 |
| | Tucker-FNO | **33.41** | **0.945** | **0.092** | **31.63** | **0.911** | **0.113** | **30.30** | **0.875** | **0.130** |
| MSIs *Cups* *Clay* *Cloth* (256×256×31) | Observed | 20.01 | 0.231 | 0.932 | 16.48 | 0.152 | 0.950 | 13.98 | 0.110 | 0.967 |
| | SIREN | 28.55 | 0.862 | 0.155 | 27.83 | 0.834 | 0.165 | 27.04 | 0.795 | 0.173 |
| | MFN | 30.41 | 0.877 | 0.138 | 27.63 | 0.824 | 0.165 | 25.91 | 0.776 | 0.184 |
| | LRTFR | 32.30 | 0.939 | 0.108 | 31.27 | 0.911 | 0.117 | 30.24 | 0.864 | 0.130 |
| | OINR | 33.18 | 0.930 | 0.097 | 31.65 | 0.909 | 0.112 | 30.25 | 0.869 | 0.130 |
| | OINR-FNO | 33.80 | 0.944 | 0.088 | 32.28 | 0.917 | 0.101 | 30.73 | 0.875 | 0.122 |
| | Tucker-FNO | **35.89** | **0.957** | **0.061** | **33.79** | **0.927** | **0.089** | **31.52** | **0.900** | **0.112** |

the shape of the vehicle as input, and we input the meshgrid in the surface, which stores 3586 mesh points for each sample. For the GINO framework, it maps the non-grid data into a grid latent space using an encoder GNO, and then FNO performs learning. The latent grid space is then mapped back to the original non-grid space through a decoder GNO. Here, we perform training using Adam optimizer (Kingma, 2014) with a learning rate 0.001. Following the configuration of GINO, the latent dimension $d_v$ is 32, and the rank of Tucker-FNO is set to the latent dimension. The Fourier frequency truncation threshold is $\mathbf{k_{max}} = (12, 12, 12)$ or $\mathbf{k_{max}} = (16, 16, 16)$. The depth of NO-based methods is 4. And the post-projection $\hat{\mathscr{Q}}$ is a 2-layer MLP. The preliminary results under 10 training epochs are shown in Table 9 and the qualitative results are demonstrated in the Figure 9. These results indicate that Tucker-FNO remains effective and efficient for the large-scale 3-D task.

***More results of image denoising.*** The results of MSI noise removal are shown in Table 8 and Fig. 10. It can be observed that Tucker-FNO consistently outperforms all baselines across different noise cases and datasets. According to Fig. 10, Tucker-FNO could obtain better visual results than other competing methods. Specifically, Tucker-FNO could attenuate noise well and also preserve the details of the MSI better.

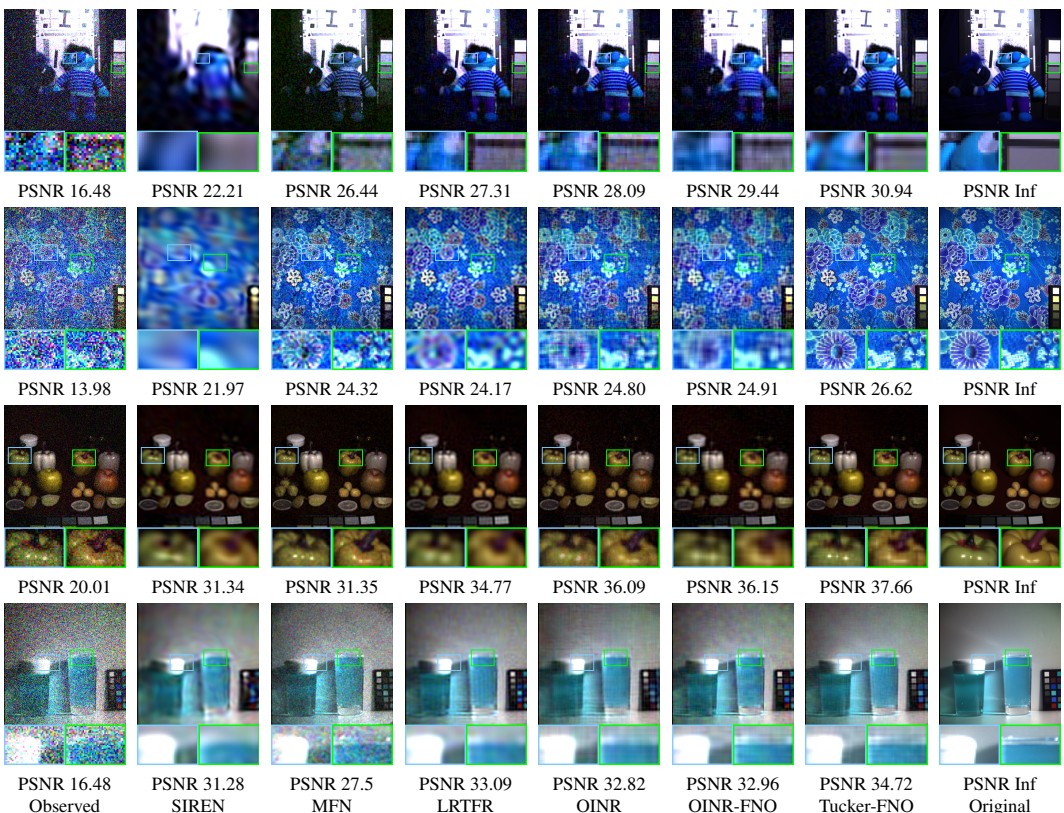

| | | | | | | |
|---|---|---|---|---|---|---|
| PSNR 16.48 | PSNR 22.21 | PSNR 26.44 | PSNR 27.31 | PSNR 28.09 | PSNR 29.44 | PSNR 30.94 | PSNR Inf |
| PSNR 13.98 | PSNR 21.97 | PSNR 24.32 | PSNR 24.17 | PSNR 24.80 | PSNR 24.91 | PSNR 26.62 | PSNR Inf |
| PSNR 20.01 | PSNR 31.34 | PSNR 31.35 | PSNR 34.77 | PSNR 36.09 | PSNR 36.15 | PSNR 37.66 | PSNR Inf |
| PSNR 16.48 Observed | PSNR 31.28 SIREN | PSNR 27.5 MFN | PSNR 33.09 LRTFR | PSNR 32.82 OINR | PSNR 32.96 OINR-FNO | PSNR 34.72 Tucker-FNO | PSNR Inf Original |

Figure 10: The qualitative results of MSI denoising by different methods on *Toys* (0.15), *Cloth* (0.2), *Fruits* (0.1) and *Cups* (0.15).

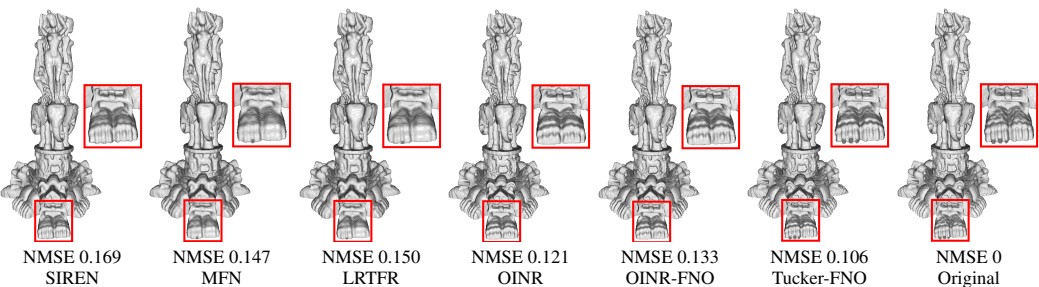

| | | | | | | |
|---|---|---|---|---|---|---|
| NMSE 0.169 SIREN | NMSE 0.147 MFN | NMSE 0.150 LRTFR | NMSE 0.121 OINR | NMSE 0.133 OINR-FNO | NMSE 0.106 Tucker-FNO | NMSE 0 Original |

Figure 11: The qualitative results of 3-dimensional volume representation by different methods.

***More results of 3D volume representation.*** INRs are commonly used as a continuous representation of 3D volumes or surfaces. Fig. 11 shows that Tucker-FNO performs well in NMSE in this case. Tucker-FNO exhibits superior recovery performance and demonstrates a distinct advantage in high-frequency signals (as indicated by the red box in Fig. 11).

***More results of video inpainting.*** To further test the scalability of our method on a long and high-resolution video from Zhang et al. (2024). The video dataset has a relatively large spatial resolution of $720 \times 960$ and contains 600 frames in total. In the experiments, we extract frames of varying lengths to evaluate the video inpainting performance (sampling rate = 0.1) of our method across different data scales. The results, presented in Table 10, show that Tucker-FNO holds advantages in terms of both effectiveness and efficiency than the OINR baseline across varying frame lengths. These findings reveal the scalability of Tucker-FNO on relatively large-scale data.

Table 9: Performance (Test Error ↓, Training Time ↓ (seconds/epoch), Evaluation Time ↓ (seconds/epoch), FNO Running Time ↓ (seconds/sample)) on Car-CFD. The test error is the denormalized L2 error here.

| Method | Fourier Mode | Test Error | Training Time | Evaluation Time | FNO Running Time |
|---|---|---|---|---|---|
| GINO | 12 | 0.2047 | 1832.13 | 327.18 | 0.0078 |
| GINO-D-FNO | 12 | 0.2187 | 1801.25 | 309.24 | 0.0065 |
| GINO-Tucker-FNO | 12 | 0.2031 | 1792.43 | 300.77 | 0.0062 |
| GINO | 16 | 0.1382 | 1858.23 | 337.28 | 0.0081 |
| GINO-D-FNO | 16 | 0.1432 | 1832.54 | 329.73 | 0.0070 |
| GINO-Tucker-FNO | 16 | 0.1297 | 1826.12 | 326.65 | 0.0064 |

Table 10: Comparisons (PSNR ↑ (dB), Time/iter ↓ (seconds)) of OINR and Tucker-FNO on a long and high-resolution video.

| Frames | Methods | PSNR | Time/iter |
|---|---|---|---|
| 100 | OINR | 39.6516 | 0.0231 |
| | Tucker-FNO | 46.5796 | 0.0173 |
| 300 | OINR | 0.9024 | 0.0409 |
| | Tucker-FNO | 42.0626 | 0.0280 |
| 600 | OINR | 35.4867 | 0.0702 |
| | Tucker-FNO | 39.7912 | 0.0342 |

## I   MORE COMPARISONS OF EFFICIENCY

Table 11: Comparisons (Params ↓, MACs ↓, Time/iter ↓ (milliseconds)) of parameter number, computation complexity, and execution time in the PDE approximation task.

| Size | $64 \times 64$ | | | $128 \times 128$ | | | $256 \times 256$ | | |
|---|---|---|---|---|---|---|---|---|---|
| Method | Params | MACs | Time/iter | Params | MACs | Time/iter | Params | MACs | Time/iter |
| FNO | 593.7 K | 4.01 M | 5.62 | 593.7 K | 55.6 M | 26.19 | 593.7 K | 217.66 M | 68.50 |
| D-FNO | 234.1 K | 3.95 M | 2.16 | 258.7 K | 23.75 M | 15.24 | 307.8 K | 98.77 M | 38.30 |
| Tucker-FNO | 256.6 K | 2.44 M | 3.15 | 268.9 K | 16.45 M | 14.14 | 293.5 K | 80.23 M | 35.23 |

We conduct more experiments to validate the efficiency of Tucker-FNO on the computation cost here. Table 11-12 demonstrates the parameters of the model, multiply accumulate operations (MACs), and execution time across different scales of data and various tasks. In Table 11, the results demonstrate that Tucker-FNO can significantly reduce both the number of parameters and the computational complexity of the algorithm in the PDE approximation task. As the input scale increases, the advantage of Tucker-FNO grows accordingly. In Table 12, Tucker-FNO presents a more significant improvement in the signal restoration task than the PDE approximation task.

Here, we further test the running time of the lifting, FNO processing, decomposition and post-projection separately in FNO and Tucker-FNO in Tablr 14, where $d_v = 16$ and $k = 12$. We set a 2-d PDE problem where the size of input is $128 \times 128$ with batch size 8. To highlight the efficiency difference, we test the running time on the CPU. The results demonstrate that the bottleneck of the decomposition module occurs when the rank is extremely large ($> 1024$), which far exceeds the rank settings ($\sim 32$) of our method. Consequently, for Tucker-FNO, FFT calculations still dominate the overall computational complexity in most scenarios, only in extreme scenarios (rank $> 1024$) might the complexity of decomposition become the dominant factor.

## J   MORE ABLATION EXPERIMENTS

Table 12: Comparisons (Params ↓, MACs ↓, Time/iter ↓ (seconds)) of parameter number and running time in the signal restoration task.

| Size | $512 \times 512 \times 31$ | | | $768 \times 768 \times 31$ | | | $1024 \times 1024 \times 31$ | | |
|---|---|---|---|---|---|---|---|---|---|
| Method | Params | MACs | Time/iter | Params | MACs | Time/iter | Params | MACs | Time/iter |
| OINR-FNO | 1.06 M | 3.338 B | 0.0525 | 1.06 M | 7.512 B | 0.1292 | 1.06 M | 13.354 B | 0.2322 |
| Tucker-FNO | 486 K | 0.538 B | 0.0424 | 519 K | 1.211 B | 0.0638 | 553 K | 2.152 B | 0.0963 |

Table 13: Ablation experiments (NMSE ↓, VCE ↓) of Tucker-FNO rank for PDE solving problems in Burger's equations.

| $k$ | 8 | | 12 | | 16 | |
|---|---|---|---|---|---|---|
| $d_v$ | NMSE | VCE | NMSE | VCE | NMSE | VCE |
| 16 | 0.0124 | 8.3804e-5 | 0.0071 | 3.1113e-5 | 0.0069 | 2.7835e-5 |
| 32 | 0.0086 | 4.1782e-5 | 0.0070 | 2.0050e-5 | 0.0063 | 1.1622e-5 |
| 64 | 0.0082 | 3.8027e-5 | 0.0060 | 1.0669e-5 | 0.0055 | 9.4175e-5 |

Here, we conduct more experiments on the width $d_v$ and Fourier frequency truncation $\mathbf{k_{max}} = (k, k, \cdots)$. Table 13 further demonstrates the ablation results in Burger's equation. Similar to the Plasticity equation in Table 6, performance improves with both $k$ and $d_v$. Moreover, Fig. 12 demonstrate the influence of $k$ and $d_v$ in the inpainting and denoising tasks. The best performance occurs at $d_v = 16 \sim 32$ and $k = 18 \sim 24$, which further validates the effectiveness of the low-rank structure in Tucker-FNO.

Table 15: Performance (NMSE ↓, VCE ↓) on Plasticity equation.

| Method | NMSE | VCE |
|---|---|---|
| FNO | 0.0080 | 1.9514e-4 |
| D-FNO | 0.0097 | 2.2150e-4 |
| Tucker-FNO | 0.0073 | 2.0010e-4 |
| CP-FNO | 0.0082 | 2.0021e-4 |

Here, we compare the different tensor decomposition in the Plasticity equation. By applying the methodological paradigm to Canonical Polyadic (CP) decomposition (Kolda & Bader, 2009a), we derive a variant of Tucker-FNO, termed CP-FNO. Table 15 demonstrates comparisons between Tucker-FNO and CP-FNO. Although the efficiency of CP decomposition is higher than that of Tucker decomposition, it has a relatively inferior accuracy. This can be attributed to the stronger representation ability of the Tucker decomposition.

Table 14: Comparisons (Time ↓ (seconds), Params ↓) of running time and parameter number in the PDE approximation task.

| Methods | | | Time | | | | Params |
| Type | Rank | Lifting | FNO Processing | Decomposition | Projection | Total Time | |
|---|---|---|---|---|---|---|---|
| FNO | - | 0.0010 | 0.4057 | - | 0.0935 | 0.5002 | 593281 |
| Tucker-FNO | 16 | 0.0008 | 0.1761 | 0.0005 | 0.0423 | 0.2197 | 70593 |
| Tucker-FNO | 64 | 0.0011 | 0.2753 | 0.0011 | 0.0597 | 0.3372 | 170529 |
| Tucker-FNO | 256 | 0.0009 | 0.2736 | 0.0088 | 0.0691 | 0.3551 | 1307553 |
| Tucker-FNO | 1024 | 0.0010 | 0.3114 | 0.1412 | 0.0648 | 0.5184 | 17652129 |

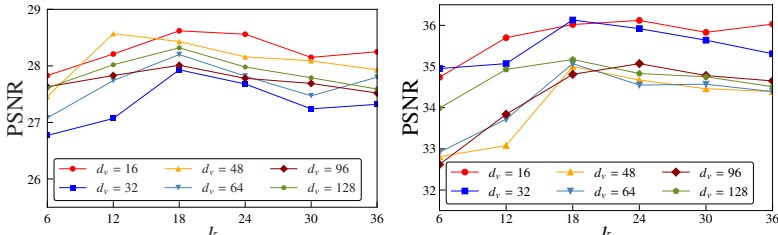

Figure 12: The PSNR results of Tucker-FNO w.r.t. different parameters $k$ and $d_v$ for *Carphone* inpainting with sampling rate 0.15 (left), and *Clay* denoising with noise deviation of 0.15 (right).

