# OpenReview forum: "Tucker-FNO: Tensor Tucker-Fourier Neural Operator and its Universal Approximation Theory"
_ICLR.cc/2026/Conference — ICLR 2026 Poster_

### Official Review · Reviewer_EBp7 · 2025-10-29

**Soundness:** 3
**Presentation:** 2
**Contribution:** 3
**Rating:** 4
**Confidence:** 4

**Summary:**

This work present the Tucker-FNO, an approach modifies the FNO to support the processing of high-dimensional inputs. Using a Tucker style functional decomposition, high-dimensional inputs may be processed into several low-dimensional inputs and processed in parallel by 1-dimensional FNOs. The authors provide a universal approximation result experiments on PDEs, image in-painting, and image denoising.

**Strengths:**

- Performing tensor factorization on the operator/function space as opposed to the parameter space is an interesting idea.
- The authors provide a functional Tucker decomposition UAT, which is, as far as I am aware, novel and could be useful in other applications.
- A broad set of baselines beyond PDEs is investigated, where Tucker-FNO shows relatively good performance.
- Several ablation studies are also present, which show notable improvements in speed and parameter efficiency in comparison to the FNO and alternative approaches.

**Weaknesses:**

1. Prior work and novelty. The manuscript does not adequately situate itself amongst existing literature, particularly the Multi-Grid Tensorized FNO [https://arxiv.org/abs/2310.00120]. This work also relies on Tucker-factorization. Although it acts on the weights of the FNO as opposed to the inputs, the works share substantial similarities in their goal to process large inputs and maintain parameter efficiency. Given this similarity, this work must be cited and compared both conceptually and empirically to clarify the novelty.
2. Limited 3D validation. While this work repeatedly claims applicability to 3D (or general high-D) PDEs, there are only experiments on 2D or 2D+time PDEs. The problems investigated are also relatively simple, and model performance has become saturated in the larger literature. Experiments on more difficult 3D or 3D+time baselines are required to support claims of scaling. Baselines include GINO [2309.00583] and GenCFD [2409.18359].
3. Implementation clarity and code. I feel many of the details of how the approach actually works in practice are obfuscated by rigorous theoretical derivations. The code provided in the supplementary material, likewise, is incomplete and does not resolve key implementation questions.

Minor typos: line 174 "construct an mapping", line 240 "multi-layer perception" (I believe perceptron is meant here), line 247 big O notation of O(log(n^3)) should be simplified to O(log(n)).

**Questions:**

1. Is it correct that the input does not undergo an explicit Tucker factorization, but rather it is processed in a way that assumes such a factorization exists? And then the inverse of the Tucker factorization is performed at the end, multiplying the factorized inputs by the core tensor?
1b. If my understanding above is correct, is this process factored in to the complexity calculation? How does this process scale in the case that the Tucker rank is large, even if it is chosen as a hyperparameter?
2. Is there any theoretical justification for Tucker factorization of the operator/input as opposed to the weights? Or can these be shown to be equivalent?
3. How does the approach perform on a PDE describing a flow in 3 spatial dimensions? Please provide an experiment with baselines.

---

> ### Author Response · Authors · 2025-11-20
> **Response to Reviewer EBp7 Part 1**
>
> We sincerely thank the reviewer for the valuable and constructive comments! We have carefully made more discussions and clarifications according to your comments.
>
> For easy reference, a list of responses to the reviewers' comments has been compiled, and the corresponding changes have been highlighted in **blue** in the main text for easy reference of our revision. Should you need further information, please let us know. We look forward to hearing from you soon.

---

> ### Author Response · Authors · 2025-11-20
> **Response to Reviewer EBp7 Part 2**
>
> >**W1: Prior work and novelty. The manuscript does not adequately situate itself amongst existing literature, particularly the Multi-Grid Tensorized FNO [https://arxiv.org/abs/2310.00120]. This work also relies on Tucker-factorization. Although it acts on the weights of the FNO as opposed to the inputs, the works share substantial similarities in their goal to process large inputs and maintain parameter efficiency. Given this similarity, this work must be cited and compared both conceptually and empirically to clarify the novelty.**
>
> R1: Thanks!
> We have provided a detailed comparison between tensorized FNO and Tucker-FNO in the Section 3.6 of the revised manuscript. Here, we offer a brief summary.
> Although both recent tensorized FNOs and our Tucker-FNO employ Tucker decomposition methods, the **implementation** and **motivation** are significantly different:
>
> 1. **Tensorized FNOs:** Tensorized FNOs utilize decomposition to **reduce the model parameters**, where decomposition is applied to **the model parameter**.
> Specifically, the tensorized FNOs decompose the model parameters, i.e., $\mathcal{R}=\mathcal{C}\times\_1\mathcal{U}\_1\times\_2\cdots\times\_d\mathcal{U}\_d\times\_{d+1}\mathcal{U}\_{d+1}\times\_{d+2} \mathcal{U}\_{d+2}$, where $\mathcal{R}\in\mathbb{R}^{k\times\cdots\times k\times d_v\times d_v}$ is the parameter tensor, $\mathcal{C}\in\mathbb{R}^{r_1\times\cdots\times r_{d+2}}$ is the core tensor, and $\mathcal{U}_i\in\mathbb{R}^{k\times r_i}$ (for $i\leq d$) or $\mathcal{U}_i\in \mathbb{R}^{d_v\times r_i}$ (for $i>d$) is the factor matrices.
> Here, for simplicity, we analyze only a single-layer FNO.
> The parameter number of tensorized FNO is $O(\hat{r}^{d+2} +\hat{r}(d_v+k))$.
> Its computational complexity for FFT and inverse FFT still equals to the traditional FNO, i.e., $O(d_vd n^d\log(n))$ for $d$-dimensional PDE problems.
> Moreover, due to the decomposition, its computation complexity for multiplying weights increases from $O(d_v^2k^d)$ to $O(\hat{r}d_v^2k^d)$.
>
> 2. **Tucker-FNO:** Our Tucker-FNO utilizes decomposition to **reduce the computational complexity**, where decomposition is applied to **the operator**. By decomposing the $d$-dimensional FNO into a series of $1$-dimensional FNOs, the computational complexity for FFT and inverse FFT is reduced to $O(d_vdn\log(n))$, and that for multiplying weights is reduced to $O(d_v^2dk)$.
>
> In summary, tensorized FNO is a new parameterization scheme for FNO's network parameters, with its underlying principles remaining similar to traditional FNO, and its motivation is to design a parameterization scheme with reduced the parameter number for FNO.
> Our Tucker-FNO, on the other hand, is a novel paradigm under the FNO framework. Especially, our motivation is to design a new neural operator paradigm with reduced computational complexity.
>
> ||Tensorized FNO |Tucker-FNO|
> |-|-|-|
> |FFT Computational Complexity| $O(d_vn^d\log(n))$ **(equal to FNO)** | $O(d_vn\log(n))$ **(less than FNO)**|
> |Multiplying Weights| $O(\hat{r}d_v^2k^d)$ **(higher than FNO)** | $O(d_v^2dk)$ **(less than FNO)**|
> |Parameter Number| $O(\hat{r}^{d+2}+\hat{r}(k+d_v))$ | $O(d_v^2dk)$ |
> |Decomposition|Model Weight|Operator|
> |Contribution| A parameterization scheme for FNO | A new computationally efficient FNO with UAT |
>
> Furthermore, we have compared the T-FNO [1] in the tables below, which is a classical tensorized FNO method. Here, T-FNO’s rank rate is set as $0.4$, which indicates that the Tucker rank of each dimension is $0.4$ of the size of the dimension, and other hyperparameters are the same as FNO.
> The results are demonstrated in the tables below.
> The first table below (i.e., Table 2 in the revised manuscript) demonstrates the effectiveness comparisons, and the second table below (i.e., Table 3 in the revised manuscript) demonstrates the efficiency comparisons.
> From the results, Tucker-FNO still holds advantages, and Tucker-FNO can both reduce computational complexity and the model parameters by decomposing the operator.
>
> | |N-S (T=20)|N-S (T=20)|N-S (T=30)|N-S (T=30)|Plasticity|Plasticity|Burger|Burger|
> |-|-|-|-|-|-|-|-|-|
> |Methods|NMSE|VCE| NMSE|VCE| NMSE|VCE| NMSE|VCE|
> |ResNet|0.0911|0.0053|0.2582|0.0351|0.1015|0.0587|0.0941|0.0651
> |FNO|0.0034|1.0311e-5|0.0072|3.9776e-5|0.0080|1.9514e-4|0.0087|4.4465e-5
> |Com-FNO|0.0036|1.1265e-5|0.0071|3.8146e-5|0.0084|2.0897e-4|0.0063|2.4543e-5
> |T-FNO|0.0035|8.8214e-6|0.0065|3.0354e-5|0.0079|2.0076e-4|0.0075|2.9672e-5
> |D-FNO|0.0063|3.0994e-5|0.0132|9.1426e-5|0.0097|2.2150e-4|0.0071|2.8610e-5
> |Tucker-FNO|0.0028|8.2254e-6|0.0061|2.9742e-5|0.0073|2.0010e-4|0.0070|2.0050e-5
>
> |Methods|Params|Time/iter|
> |-|-|-|
> |FNO|593.7 K|0.0685|
> |T-FNO|232.6 K|0.0746|
> |Tucker-FNO|293.5 K|0.0352|
>
> [1] Kossaifi, J., Kovachki, N. B., Azizzadenesheli, K., & Anandkumar, A. Multi-Grid Tensorized Fourier Neural Operator for High-Resolution PDEs. Transactions on Machine Learning Research.

---

> ### Author Response · Authors · 2025-11-20
> **Response to Reviewer EBp7 Part 3**
>
> >**W2: Limited 3D validation. While this work repeatedly claims applicability to 3D (or general high-D) PDEs, there are only experiments on 2D or 2D+time PDEs. The problems investigated are also relatively simple, and model performance has become saturated in the larger literature. Experiments on more difficult 3D or 3D+time baselines are required to support claims of scaling. Baselines include GINO [2309.00583] and GenCFD [2409.18359].**
>
> R2: Thanks!
> According to your suggestion, we have evaluated our Tucker-FNO in a complex setting which include geometry.
> Here, we replace the FNO with our Tucker-FNO in GINO [2], and evaluate it in a large-scale computational fluid dynamics dataset from [2], which is a 3-D PDE problem.
> The input shapes of the dataset are from the ShapeNet Car category [3].
> The preliminary results under 10 training epochs are shown in the table below (i.e., Table 9 in the revised manuscript), and the visualizations are demonstrated in the Figure 8 of the revised manuscript.
> The results demonstrate that our Tucker-FNO remains effective for **large-scale 3-D PDEs**, which include complex geometry.
>
> |Methods|Test Error|
> |-|-|
> |GINO |0.2047|
> |GINO-D-FNO|0.2187|
> |GINO-Tucker-FNO|0.2031|
>
> [2] Li, Z., Kovachki, N., Choy, C., Li, B., Kossaifi, J., Otta, S., Nabian, M., Stadler, M., Hundt, C., Azizzadenesheli, K., Anandkumar, A. (2023) Geometry-Informed Neural Operator for Large-Scale 3D PDEs. NeurIPS 2023.
>
> [3] Angel X Chang, Thomas Funkhouser, Leonidas Guibas, Pat Hanrahan, Qixing Huang, Zimo Li, Silvio Savarese, Manolis Savva, Shuran Song, Hao Su, et al. Shapenet: An information-rich 3d model repository. arXiv preprint arXiv:1512.03012, 2015.

---

> > ### Comment · Reviewer_EBp7 · 2025-11-23
> >
> > I would like to thank the authors for taking the time to investigate the many avenues I suggested during the review. This has clarified many of my concerns and questions. I have also read through comments and responses from the other reviewers and the related discussions.
> >
> > However, I feel that there are still some open questions with regards to the 3D PDE experiment on general geometry. This is the dataset from GINO correct? As they report, there are $O(10^5)$ mesh points on the surface and $O(10^7)$ in space. Is the entire mesh used? If there is downsampling, how many input points are used? What does the test error represent -- is this a relative L2 error, and for what variable? pressure and velocity are mentioned as examples, but this is not explicitly stated in the table or elsewhere in the appendix, as far as I can find. Since efficiency, not improved test error, is the main claim of the proposed work, I would also like to request training/evaluation times. The Fourier mode limit also seems relatively low -- is this coming from the original hyperparameters of GINO or was there some hyperparameter sweep done to select 12?
> >
> > Currently, I am not entirely convinced in the Tucker-FNO's effectiveness for high-dimensional and large scale problems. If the authors can provide more comprehensive details on this experiment that demonstrate it is indeed in the high-dimensional and large-scale class of problems, I would be happy to raise my score. Once again, I would like to thank the authors to their commitment in this rebuttal and discussion phase, and I look forward to their response.

---

> > > ### Author Response · Authors · 2025-11-25
> > > **Response to Reviwer EBp7 Part 9**
> > >
> > > >**What does the test error represent -- is this a relative L2 error, and for what variable? pressure and velocity are mentioned as examples, but this is not explicitly stated in the table or elsewhere in the appendix, as far as I can find.**
> > >
> > > Thanks! Yes, the test error is defined follow the de-normalized L2 following GINO, and we have included a description of the test error in the caption of Table 9 in the revised manuscript. Moreover, as suggested, we further included the detailed description of the Car-CFD task in Appendix H of the revised manuscript, and the brief description is as follow.
> > >
> > > Car-CFD Dataset is the processed version of the dataset introduced in [6], and is introduced in GINO [2]. It encodes a triangular mesh over the surface of a 3D model car and provides the air pressure at each centroid and vertex of the mesh when the car is placed in a simulated wind tunnel with a recorded inlet velocity. For GINO, the inputs are a signed distance function evaluated over a regular 3D grid of query points, as well as the inlet velocity. Outputs are pressure values at each centroid of the triangle. In summary, the velocity is a parameter for the PDE, which is fixed at 20m/s, and the pressure is the output of NO (e.g., GINO, GINO-DFNO, and GINO-Tucker-FNO).
> > >
> > > [2] Li, Z., Kovachki, N., Choy, C., Li, B., Kossaifi, J., Otta, S., Nabian, M., Stadler, M., Hundt, C., Azizzadenesheli, K., Anandkumar, A. (2023) Geometry-Informed Neural Operator for Large-Scale 3D PDEs. NeurIPS 2023.
> > >
> > > [6] Umetani, N. and Bickel, B. (2018). "Learning three-dimensional flow for interactive aerodynamic design". ACM Transactions on Graphics, 2018. https://dl.acm.org/doi/10.1145/3197517.3201325.

---

> > > ### Author Response · Authors · 2025-11-25
> > > **Response to Reviwer EBp7 Part 11**
> > >
> > > >**Currently, I am not entirely convinced in the Tucker-FNO's effectiveness for high-dimensional and large scale problems. If the authors can provide more comprehensive details on this experiment that demonstrate it is indeed in the high-dimensional and large-scale class of problems, I would be happy to raise my score. Once again, I would like to thank the authors to their commitment in this rebuttal and discussion phase, and I look forward to their response.**
> > >
> > > Thanks! As suggested, we have provided a more detailed explanation to demonstrate that Car-CFD is a large-scale and high-dimensional PDE task, along with additional experiments to further validate the effectiveness of Tucker-FNO on Car-CFD. And we hope that the additional experiments (Table 9 in the revised manuscript) would help to validate the efficiency and effectiveness of Tucker-FNO.
> > >
> > > Moreover, we further demonstrate the computational efficiency of Tucker-FNO in a 10-D toy setting. Here, we compare the single-layer FNO and Tucker-FNO, where $d_v=4$ and $k=2$, in a 10-D PDE approximation problem in terms of efficiency. The shape of input is $4\times 4\times\cdots\times 4$. The comparison of parameter number and running time is demonstrated in the table below. It can be seen that Tucker-FNO still holds certain efficiency advantages in the high-dimensional (10-D) example.
> > >
> > > |Methods|Params|Time (seconds)|
> > > |-|-|-|
> > > |FNO|26M|0.0093|
> > > |Tucker-FNO|22M|0.0061|
> > >
> > > Currently, most research on neural operators focuses on 3-D or lower-dimensional PDEs, while studies addressing the solution of very high-dimensional PDEs remain relatively scarce. In the future, we can further explore the use of Tucker-FNO to improve the efficiency of solving genuinely high-dimensional PDEs (>3-D) by combining with specialized designs such as reduced basis architectures [7].
> > >
> > > Should you require any further information, please do not hesitate to let us know. We would greatly appreciate your feedback.
> > >
> > > [7] Luo, D., O’Leary-Roseberry, T., Chen, P., & Ghattas, O. (2025). Efficient PDE-constrained optimization under high-dimensional uncertainty using derivative-informed neural operators. SIAM Journal on Scientific Computing, 47(4), C899-C931.

---

> > > > ### Comment · Reviewer_EBp7 · 2025-11-27
> > > >
> > > > Thank you for the new tables. Given the results from Tables 9 and 10 demonstrate the potential to scale to problems at quite large scales, I feel that my remaining concerns have been adequately addressed. I will update my assessment accordingly.

---

> > > > > ### Author Response · Authors · 2025-11-27
> > > > > **Response to Reviwer EBp7**
> > > > >
> > > > > We thank the reviewer again for the constructive comments and suggestions!

---

> > > ### Author Response · Authors · 2025-11-25
> > > **Response to Reviwer EBp7 Part 12**
> > >
> > > Should you require any further information, please do not hesitate to let us know. We would greatly appreciate your feedback.
> > >
> > > >**Reference**
> > >
> > > [1] Kossaifi, J., Kovachki, N. B., Azizzadenesheli, K., & Anandkumar, A. Multi-Grid Tensorized Fourier Neural Operator for High-Resolution PDEs. Transactions on Machine Learning Research.
> > >
> > > [2] Li, Z., Kovachki, N., Choy, C., Li, B., Kossaifi, J., Otta, S., Nabian, M., Stadler, M., Hundt, C., Azizzadenesheli, K., Anandkumar, A. (2023) Geometry-Informed Neural Operator for Large-Scale 3D PDEs. NeurIPS 2023.
> > >
> > > [3] Angel X Chang, Thomas Funkhouser, Leonidas Guibas, Pat Hanrahan, Qixing Huang, Zimo Li, Silvio Savarese, Manolis Savva, Shuran Song, Hao Su, et al. Shapenet: An information-rich 3d model repository. arXiv preprint arXiv:1512.03012, 2015.
> > >
> > > [4] Kangjie Li and Wenjing Ye. D-fno: A decomposed fourier neural operator for large-scale parametric partial differential equations. Computer Methods in Applied Mechanics and Engineering, 436: 117732, 2025
> > >
> > > [5] Olek C Zienkiewicz, Robert Leroy Taylor, and Perumal Nithiarasu. The finite element method for fluid dynamics. Butterworth-Heinemann, 2013.
> > >
> > > [6] Umetani, N. and Bickel, B. (2018). "Learning three-dimensional flow for interactive aerodynamic design". ACM Transactions on Graphics, 2018. https://dl.acm.org/doi/10.1145/3197517.3201325.
> > >
> > > [7]  Luo, D., O’Leary-Roseberry, T., Chen, P., & Ghattas, O. (2025). Efficient PDE-constrained optimization under high-dimensional uncertainty using derivative-informed neural operators. SIAM Journal on Scientific Computing, 47(4), C899-C931.

---

> ### Author Response · Authors · 2025-11-20
> **Response to Reviewer EBp7 Part 4**
>
> >**W3: Implementation clarity and code. I feel many of the details of how the approach actually works in practice are obfuscated by rigorous theoretical derivations. The code provided in the supplementary material, likewise, is incomplete and does not resolve key implementation questions.**
>
> R3: Thanks! We have provided comprehensive code implementations (including signal restoration and PDE approximation tasks) in our revised supplementary material to enhance reproducibility. I hope this may addresses your concerns.
>
> >**W4: Minor typos: line 174 "construct an mapping", line 240 "multi-layer perception" (I believe perceptron is meant here), line 247 big O notation of O(log(n^3)) should be simplified to O(log(n)).**
>
> R4: Thank you for your valuable suggestions! We have revised these typos in the revised manuscript.

---

> ### Author Response · Authors · 2025-11-20
> **Response to Reviewer EBp7 Part 5**
>
> >**W5: Is it correct that the input does not undergo an explicit Tucker factorization, but rather it is processed in a way that assumes such a factorization exists? And then the inverse of the Tucker factorization is performed at the end, multiplying the factorized inputs by the core tensor? 1b. If my understanding above is correct, is this process factored in to the complexity calculation? How does this process scale in the case that the Tucker rank is large, even if it is chosen as a hyperparameter?**
>
> R5: Thanks!
> Yes, your understanding is totally correct.
> And, we don't consider the computational complexity of the decomposition, as it is not the primary computational burden in FNO.
> Since the recent study [4] demonstrated that FFT calculations dominate the computational complexity in FNO, we ignore the other computational complexity.
>
> The decomposition's computational complexity is $O(\hat{r}d_vn^d)$, where $\hat{r}=max\\{r_1,\cdots,r_d\\}$ is the rank and $n$ is the size of input.
> In the table below (i.e., Table 13 in the revised manuscript), we demonstrate the running time of the lifting, FNO processing, decomposition and post-projection in FNO and Tucker-FNO, where $d_v=16$ and $k=12$. We set a 2-D PDE problem where the size of input is $128\times 128$ with batch size $8$.
> To highlight the differences, we test the running time on the CPU.
> The results demonstrate that the bottleneck of the decomposition module occurs when the rank is extremely large (>1024), which far exceeds the rank settings (~32) of our method.
> Consequently, for Tucker-FNO, FFT calculations still dominate the overall computational complexity in most scenarios, only in extreme scenarios (rank>1024) might the complexity of decomposition become the dominant factor.
>
> |Methods|Lifting|FNO Processing|Decomposition|Projection|Total Time|Param|
> |-|-|-|-|-|-|-|
> |FNO|0.0010|0.4057|-|0.0935|0.5002|593281|
> |Tucker-FNO (rank=16)|0.0008|0.1761|0.0005|0.0423|0.2197|70593|
> |Tucker-FNO (rank=64)|0.0011|0.2753|0.0011|0.0597|0.3372|170529|
> |Tucker-FNO (rank=256)|0.0009|0.2736|0.0088|0.0691|0.3551|1307553|
> |Tucker-FNO (rank=1024)|0.0010|0.3114|0.1412|0.0648|0.5184|17652129|
>
> [4] Kangjie Li and Wenjing Ye. D-fno: A decomposed fourier neural operator for large-scale parametric partial differential equations. Computer Methods in Applied Mechanics and Engineering, 436: 117732, 2025

---

> ### Author Response · Authors · 2025-11-20
> **Response to Reviewer EBp7 Part 6**
>
> >**W6: Is there any theoretical justification for Tucker factorization of the operator/input as opposed to the weights? Or can these be shown to be equivalent?**
>
> R6: Thanks!
> Yes, we have provided a detailed theoretical comparison between tensorized FNOs (which decompose the weights) and Tucker-FNO (which decomposes the operator/input) in Section 3.6 of the revised manuscript. A brief summary is also included in our response to your comment W1. As discussed, Tucker factorization applied to the operator/input is theoretically distinct from that applied to the model weights. Thanks!
>
> >**W7: How does the approach perform on a PDE describing a flow in 3 spatial dimensions? Please provide an experiment with baselines.**
>
> R7: Thanks! We have further demonstrate the results in computational the fluid dynamics problem in the response to your comment W2, which is a PDE describing in 3 spatial dimensions. Thanks.

---

> ### Author Response · Authors · 2025-11-20
> **Response to Reviewer EBp7 Part 7**
>
> Should you require any further information, please do not hesitate to let us know. We would greatly appreciate your feedback.
>
> >**Reference**
>
> [1] Kossaifi, J., Kovachki, N. B., Azizzadenesheli, K., & Anandkumar, A. Multi-Grid Tensorized Fourier Neural Operator for High-Resolution PDEs. Transactions on Machine Learning Research.
>
> [2] Li, Z., Kovachki, N., Choy, C., Li, B., Kossaifi, J., Otta, S., Nabian, M., Stadler, M., Hundt, C., Azizzadenesheli, K., Anandkumar, A. (2023) Geometry-Informed Neural Operator for Large-Scale 3D PDEs. NeurIPS 2023.
>
> [3] Angel X Chang, Thomas Funkhouser, Leonidas Guibas, Pat Hanrahan, Qixing Huang, Zimo Li, Silvio Savarese, Manolis Savva, Shuran Song, Hao Su, et al. Shapenet: An information-rich 3d model repository. arXiv preprint arXiv:1512.03012, 2015.
>
> [4] Kangjie Li and Wenjing Ye. D-fno: A decomposed fourier neural operator for large-scale parametric partial differential equations. Computer Methods in Applied Mechanics and Engineering, 436: 117732, 2025

---

> ### Author Response · Authors · 2025-11-25
> **Response to Reviwer EBp7 Part 8**
>
> We sincerely thank the reviewer for the comments and suggestions on large-scale and high-dimensional (generally 3-D) problems. We have included the specific settings for the Car-CFD task in Appendix H of the revised manuscript and further discussed the performance of our method to demonstrate its efficiency.
>
> >**However, I feel that there are still some open questions with regards to the 3D PDE experiment on general geometry. This is the dataset from GINO correct? As they report, there are $O(10^5)$ mesh points on the surface and $O(10^7)$ in space. Is the entire mesh used? If there is downsampling, how many input points are used?**
>
> Thanks! Yes, we utilize the Car-CFD dataset in GINO from [https://zenodo.org/records/13936501], which takes 611 weight-tight shapes out of the 889 instances, and divides the 611 instances into 500 for training and 111 for validation.
>
> The dataset in GINO that contains $O(10^5)$ mesh points on the surface and $O(10^7)$ in the spatial domain is the Ahmed-Body dataset, which is not publicly available. Therefore, we use the Car-CFD dataset. For Car-CFD task, the goal is to estimate the full pressure field given the shape of the vehicle as input, and we input the meshgrid in the surface, which stores 3586 mesh points for each sample. For the GINO framework, it maps the non-grid data into a grid latent space using an encoder GNO, and then FNO performs learning. The latent grid space is then mapped back to the original non-grid space through decoder GNO. Following the config of [GINO](https://github.com/neuraloperator/neuraloperator/blob/14c0f7320dc7c94e907a16fd276248df2d71407c/config/gino_carcfd_config.py#L12), we set the latent grid size as $(32, 32, 32)$. Therefore, the GNO performs a mapping from $(3568,3)$ to $(32,32,32,d_v)$ by the encoder GNO, and performs a mapping from $O(32,32,32,d_v)$ back to $(3568, 3)$ by the decoder GNO. Between them, the Tucker-FNO performs a mapping from $(32,32,32,d_v)$ to $(32,32,32,d_v)$. Given the size of the dataset, it is rational to say that Tucker-FNO remains effective for large-scale PDE solving. In fact, using a finite element solver [5], each simulation for a single sample of this size takes approximately 50 minutes, making it a large-scale PDE solving task [2].
>
> [2] Li, Z., Kovachki, N., Choy, C., Li, B., Kossaifi, J., Otta, S., Nabian, M., Stadler, M., Hundt, C., Azizzadenesheli, K., Anandkumar, A. (2023) Geometry-Informed Neural Operator for Large-Scale 3D PDEs. NeurIPS 2023.
>
> [5] Olek C Zienkiewicz, Robert Leroy Taylor, and Perumal Nithiarasu. The finite element method for fluid dynamics. Butterworth-Heinemann, 2013.

---

> ### Author Response · Authors · 2025-11-25
> **Response to Reviwer EBp7 Part 10**
>
> >**Since efficiency, not improved test error, is the main claim of the proposed work, I would also like to request training/evaluation times.**
>
> Thanks! As suggested, we have reported the training time (per epoch), evaluation time (per epoch), and the specific FNO module running time (per sample) in GINO and GINO-Tucker-FNO ($k=16$) in the Car-CFD task. The results are demonstrated in the table below (i.e., Table 9 in the revised manuscript). Tucker-FNO holds an advantage in terms of efficiency for the Car-CFD task.
>
> |Methods|Traning Time (seconds/epoch)|Evaluation Time (seconds/epoch)|FNO Running Time (seconds/sample)
> |-|-|-|-|
> |GINO|1858.23|337.28|0.0081|
> |GINO-Tucker-FNO|1826.12|326.65|0.0064|
>
> >**The Fourier mode limit also seems relatively low -- is this coming from the original hyperparameters of GINO or was there some hyperparameter sweep done to select 12?**
>
> Thanks! We set the Fourier mode as $12$ from our experience with other PDE tasks. Previous studies have shown that $k$ in FNO is scalable, and our earlier results indicate that Tucker-FNO is also scalable with respect to $k$. Therefore, we conduct training with $k=12$, assuming that the results are scalable.
>
> As suggested, we further evaluate GINO, GINO-DFNO, and GINO-Tucker-FNO using the default setting where Fourier mode is configured as $16$, which is mentioned in the official configuration of [GINO](https://github.com/neuraloperator/neuraloperator/blob/14c0f7320dc7c94e907a16fd276248df2d71407c/config/models.py#L159) [2]. The results are shown in the below table (i.e., Table 9 of the revised manuscript), which demonstrates that Tucker-FNO remains effective for **large-scale 3-D PDEs**, and is scalable with respect to the Fourier mode $k$.
>
> |Methods|Fourier Mode|Test Error|
> |-|-|-|
> |GINO|12|0.2047|
> |GINO-DFNO|12|0.2187|
> |GINO-Tucker-FNO|12|0.2031|
> |GINO|16| 0.1382 |
> |GINO-DFNO|16|0.1432
> |GINO-Tucker-FNO|16|0.1297|
>
> [2] Li, Z., Kovachki, N., Choy, C., Li, B., Kossaifi, J., Otta, S., Nabian, M., Stadler, M., Hundt, C., Azizzadenesheli, K., Anandkumar, A. (2023) Geometry-Informed Neural Operator for Large-Scale 3D PDEs. NeurIPS 2023.

---

### Official Review · Reviewer_rkTN · 2025-10-30

**Soundness:** 3
**Presentation:** 3
**Contribution:** 3
**Rating:** 6
**Confidence:** 1

**Summary:**

This paper aims to address the inefficiency of fourier neural operator (FNO) in solving large-scale and high-dimensional PDEs. To reduce computational complexity, this paper proposes Tucker-FNO, a neural operator that decomposes the high-dimensional FNO into a series of 1-dimensional FNOs through Tucker decomposition. It also proves the universal approximation theorem of Tucker-FNO. Experiments are done on PDEs and video reconstruction. The proposed method outperforms previous methods on both tasks.

**Strengths:**

1. The idea is simple yet effective.
2. The theoretical and practical computation decreases significantly.

**Weaknesses:**

1. Figure 4 looks weird to me. The Carphone video reconstruction gets worse when the Fourier frequency truncation k increases. It is explained in the paper that "excessive parameters hinder convergence efficiency". However, it is not the case in the PDE problem in Table 4. Why does this difference exist? What if we simply wait the model to converge?
2. Related to 1), if the model is not scalable, then decreasing the computation complexity seems not as important as claimed. A figure of scaling would be appreciated.

**Questions:**

1. Why Tucker rank is directly set to the latent dimension for PDEs? How does the performance change with Tucker rank?
2. Why not compare with F-FNO since it is already mentioned in the Related Work along with others that are compared with?

---

> ### Author Response · Authors · 2025-11-20
> **Response to Reviewer rkTN Part 1**
>
> We sincerely thank the reviewer for the valuable and constructive comments! We have carefully made more discussions and clarifications according to your comments.
>
> For easy reference, a list of responses to the reviewers' comments has been compiled, and the corresponding changes have been highlighted in **blue** in the main text for easy reference of our revision. Should you need further information, please let us know. We look forward to hearing from you soon.
>
> >**W1: figure 4 looks weird to me. The Carphone video reconstruction gets worse when the Fourier frequency truncation k increases. It is explained in the paper that "excessive parameters hinder convergence efficiency". However, it is not the case in the PDE problem in Table 4. Why does this difference exist? What if we simply wait the model to converge?**
>
> R1: Thanks! In fact, the PDE approximation task and the signal restoration task are different in terms of the task complexity and the optimization paradigm:
>
> 1. For PDE approximation task, it follows the classical **supervised** machine learning paradigm. Through a dataset, the model learns the projection from the input to the output.
>
> 2. For our signal restoration task, it follows the classical **self-supervised** machine learning paradigm. For each signal, a model is trained to map coordinate indices to signal values, following the Implicit Neural Representation (INR) framework. Notably, in the INR paradigm, training is performed on a single signal (e.g., one image or one video), without relying on large-scale datasets.
>
> || PDE Approximation Task | Signal Restoration (INR Paradigm)|
> |-|-|-|
> |Machine Learning Paradigm| Supervised | Self-Supervised|
> |Data Scale| Relatively Large | Small (A Single Signal for Each Model)|
>
> Because the INR paradigm lacks access to large-scale datasets, signal restoration under this setting is more susceptible to overfitting, as predicted by statistical learning theory [1]. As a result, the estimation error tends to increase with model complexity. This phenomenon is also discussed in [2].
> Consequently, due to the different learning paradigms of the two tasks, the signal restoration task is more prone to overfitting, resulting in the method's performance not improving with increasing model size.
>
> As suggested, we allowed the model to converge fully, and the updated results for the Foreman dataset at SR=0.15, which are shown in the table below (i.e., Figure 4 in the revised manuscript). These results have been incorporated into the revised manuscript.
> Despite the risk of overfitting, Tucker-FNO demonstrates robustness to hyperparameter variations, achieving satisfactory performance across a wide range of configurations.
>
> |k|6|12|18|24|30|36|
> |-|-|-|-|-|-|-|
> |Rank|PSNR|PSNR|PSNR|PSNR|PSNR|PSNR
> |16 | 34.74 | 35.70 | 36.02 | 36.12 | 35.83 | 36.03 |
> |32 | 34.95 | 35.07 | 36.13 | 35.92 | 35.64 | 35.31 |
> |48 | 32.81 | 33.08 | 35.00 | 34.67 | 34.46 | 34.39 |
> |64 | 32.92 | 33.72 | 35.07 | 34.55 | 34.57 | 34.39 |
> |96 | 32.62 | 33.84 | 34.81 | 35.07 | 34.78 | 34.65 |
> |128 | 33.99 | 34.93 | 35.17 | 34.83 | 34.75 | 34.51 |
>
> [1] Vapnik, V. N. (1999). An overview of statistical learning theory. IEEE transactions on neural networks, 10(5), 988-999.
>
> [2] Luo, Y., Zhao, X., Li, Z., Ng, M. K., & Meng, D. (2023). Low-rank tensor function representation for multi-dimensional data recovery. IEEE transactions on pattern analysis and machine intelligence, 46(5), 3351-3369.

---

> ### Author Response · Authors · 2025-11-20
> **Response to Reviewer rkTN Part 2**
>
> >**W2: Related to 1), if the model is not scalable, then decreasing the computation complexity seems not as important as claimed. A figure of scaling would be appreciated.**
>
> R2: Thanks!
> We discuss the differences between the PDE approximation task and the signal restoration task in the above. Here, we further elaborate on the scalability of Tucker-FNO in the context of the PDE approximation task.
>
> The scale of Tucker-FNO is determined by the latent dimension $d_v$, Tucker rank and frequency truncation threshold $k$.
> The first table below (i.e., Table 6 in the revised manuscript) demonstrates the influence of latent dimension $d_v$ and frequency truncation threshold $k$ on the performance of Tucker-FNO, where the rank is as same as the latent dimension.
> And the second table below (i.e., Table 7 in the revised manuscript) demonstrates the influence of latent dimension $d_v$ and Tucker rank on the performance of Tucker-FNO.
> Tucker-FNO’s latent dimension $d_v$, Tucker rank, and frequency truncation threshold $k$ exhibit a positive correlation with the Tucker-FNO’s performance.
> Together, the results from the below table could demonstrate the scalability of Tucker-FNO.
>
> |$k$ | 8 | 8 | 12 | 12 | 16 | 16
> |-|-|-|-|-|-|-|
> $d_v$ & Rank | NMSE | VCE | NMSE | VCE | NMSE | VCE |
> 16 | 0.0098 | 2.07e-4 | 0.0092 | 2.06e-4 | 0.0092 | 2.06e-5 |
> 32 | 0.0074 | 2.18e-4 | 0.0074 | 2.00e-4 | 0.0073 | 1.98e-4 |
> 64 | 0.0068 | 2.45e-4 | 0.0064 | 2.04e-4 | 0.0062 | 2.00e-4 |
>
>
> |Rank|16|16|32|32|64|64|
> |-|-|-|-|-|-|-|
> |$d_v$|NMSE|VCE|NMSE|VCE|NMSE|VCE|
> |16|0.0092|2.06e-4|0.0087|1.84e-4|0.0086|1.97e-5|
> |32|0.0079|2.03e-4|0.0074|2.00e-4|0.0071|1.78e-4|
> |64|0.0064|1.87e-4|0.0064|2.00e-4|0.0063|2.04e-4|

---

> ### Author Response · Authors · 2025-11-20
> **Response to Reviewer rkTN Part 3**
>
> >**W3: Why Tucker rank is directly set to the latent dimension for PDEs? How does the performance change with Tucker rank?**
>
> R3: Thanks!
> We now further demonstrate that our Tucker-FNO is not overly sensitive to the choice of rank.
> The table below (i.e., Table 7 in the revised manuscript) shows the results of NMSE with respect to rank and latent dimension $d_v$, where we set them as 2 independent hyperparameters.
> From the results, we can see that our method is relatively more robust w.r.t. decomposition rank than latent dimension $d_v$.
> And, Tucker-FNO demonstrates relatively satisfactory performances for a range of hyperparameter selection of rand and $d_v$.
>
> |Rank|16|32|64|
> |-|-|-|-|
> |$d_v$|NMSE|NMSE|NMSE|
> |16|0.0092|0.0087|0.0086|
> |32|0.0079|0.0074|0.0071|
> |64|0.0064|0.0064|0.0063|

---

> ### Author Response · Authors · 2025-11-20
> **Response to Reviewer rkTN Part 4**
>
> >**W4: Why not compare with F-FNO since it is already mentioned in the Related Work along with others that are compared with?**
>
> R4: Thanks! In fact, F-FNO and Tucker-FNO present two distinct track in terms of motivation and implementation.
> F-FNO aims to reduce the model parameter by fatctorizing the network using tensor decomposition, And, our Tucker-FNO aims to reduce the computation complexity by decomposing the operator.
> Furthermore, we have compared the T-FNO [3] in the tables below, which is a more recent tensorized FNO. Here, T-FNO’s rank rate is set as $0.4$, which indicates that the Tucker rank of each dimension is $0.4$ of the size of the dimension, and other hyperparameters are the same as FNO.
> The first table below (i.e., Table 2 in the revised manuscript) demonstrates the effectiveness comparisons, and the second table below (i.e., Table 3 in the revised manuscript) demonstrates the efficiency comparisons.
> From the results, Tucker-FNO still holds advantages, and Tucker-FNO can both reduce computational complexity and the model parameters.
>
> | |N-S (T=20)|N-S (T=20)|N-S (T=30)|N-S (T=30)|Plasticity|Plasticity|Burger|Burger|
> |-|-|-|-|-|-|-|-|-|
> |Methods|NMSE|VCE| NMSE|VCE| NMSE|VCE| NMSE|VCE|
> |ResNet|0.0911|0.0053|0.2582|0.0351|0.1015|0.0587|0.0941|0.0651
> |FNO|0.0034|1.0311e-5|0.0072|3.9776e-5|0.0080|1.9514e-4|0.0087|4.4465e-5
> |Com-FNO|0.0036|1.1265e-5|0.0071|3.8146e-5|0.0084|2.0897e-4|0.0063|2.4543e-5
> |T-FNO|0.0035|8.8214e-6|0.0065|3.0354e-5|0.0079|2.0076e-4|0.0075|2.9672e-5
> |D-FNO|0.0063|3.0994e-5|0.0132|9.1426e-5|0.0097|2.2150e-4|0.0071|2.8610e-5
> |Tucker-FNO|0.0028|8.2254e-6|0.0061|2.9742e-5|0.0073|2.0010e-4|0.0070|2.0050e-5
>
> |Methods|Params|Time/iter|
> |-|-|-|
> |FNO|593.7 K|0.0685|
> |T-FNO|232.6 K|0.0746|
> |Tucker-FNO|293.5 K|0.0352|
>
> [3] Kossaifi, J., Kovachki, N. B., Azizzadenesheli, K., & Anandkumar, A. Multi-Grid Tensorized Fourier Neural Operator for High-Resolution PDEs. Transactions on Machine Learning Research.

---

> ### Author Response · Authors · 2025-11-20
> **Response to Reviewer rkTN Part 5**
>
> Should you require any further information, please do not hesitate to let us know. We would greatly appreciate your feedback.
>
> >**Reference**
>
> [1] Vapnik, V. N. (1999). An overview of statistical learning theory. IEEE transactions on neural networks, 10(5), 988-999.
>
> [2] Luo, Y., Zhao, X., Li, Z., Ng, M. K., & Meng, D. (2023). Low-rank tensor function representation for multi-dimensional data recovery. IEEE transactions on pattern analysis and machine intelligence, 46(5), 3351-3369.
>
> [3] Kossaifi, J., Kovachki, N. B., Azizzadenesheli, K., & Anandkumar, A. Multi-Grid Tensorized Fourier Neural Operator for High-Resolution PDEs. Transactions on Machine Learning Research.

---

> ### Comment · Reviewer_rkTN · 2025-11-25
>
> Thanks for the detailed response. The rebuttal has solve most my concerns. However, I still think there's something not clear.
>
> As far as I know, the "map coordinate indices to signal values" cannot be seen as self-supervised learning in any community. Generally self-supervised learning refers to learn some new signal without knowing this signal. For example, learning image embedding, image segmentation masks, or image keypoints, only from images without knowing embeddings, masks, or keypoints. Here we know the coordinate indices AND signal values, which is very typical supervised learning. The easiest way to test scalability is to use a long and high-resolution video.

---

> > ### Author Response · Authors · 2025-11-26
> > **Response to Reviewer rkTN Part 6**
> >
> > >**As far as I know, the "map coordinate indices to signal values" cannot be seen as self-supervised learning in any community. Generally self-supervised learning refers to learn some new signal without knowing this signal. For example, learning image embedding, image segmentation masks, or image keypoints, only from images without knowing embeddings, masks, or keypoints. Here we know the coordinate indices AND signal values, which is very typical supervised learning. The easiest way to test scalability is to use a long and high-resolution video.**
> >
> > Thanks! As suggested, we further test the scalability of our method on a long and high-resolution video from [4]. The video dataset has a relatively large spatial resolution of 720×960 and contains 600 frames in total. In the experiments, we extract frames of varying lengths to evaluate the video inpainting performance (sampling rate = 0.1) of our method across different data scales. The results, presented in the table below (i.e., Table 10 in the revised manuscript), show that Tucker-FNO holds advantages in terms of both effectiveness and efficiency than the OINR baseline across varying frame lengths. These findings reveal the scalability of Tucker-FNO on relatively large-scale data.
> >
> > |Frames|Methods|PSNR (dB)| Time(seconds/iteration)|
> > |-|-|-|-|
> > |100|OINR|39.6516|0.0231|
> > |100|Tucker-FNO|46.5796|0.0173|
> > |300|OINR|39.2519|0.9024|0.0409|
> > |300|Tucker-FNO|42.0626|0.0280|
> > |600|OINR|35.4867|0.0702|
> > |600|Tucker-FNO|39.7912|0.0342|
> >
> > By the way, we totally agree that interpreting this learning problem (map coordinate indices to signal values) as supervised other than self-supervised is more appropriate. In the previous rebuttal, we interpreted this kind of learning paradigm as self-supervised because the true signal is sometimes unknown for the considered the problem. For example, for image denoising, we only have a noisy image and do not have the underlying clean image signal. Then we train the network to fit the noisy image (from coordinates to noisy signal value), and hope that the intrinsic implicit regularization brought by the neural operator itself could capture clean image signal and produce a clean image ultimately (this was more expected by the Tucker-FNO because it additionally encodes the low rankness which is beneficial for denoising). The self-supervised term was therefore occasionally used in some literature in low level-vision [5, 6, 7] under such learning paradigm, because they actually learn something new (clean signal) from only a degraded image. In the revised manuscript, we have prevented from using such self-supervised interpretation by following your suggestion. Thanks.
> >
> > [4] Zhang, Y., Wu, J., Li, W., Li, B., Ma, Z., Liu, Z., & Li, C. (2024). Video instruction tuning with synthetic data. arXiv preprint arXiv:2410.02713.
> >
> > [5] Zhang, T., Quan, Y., & Ji, H. (2024, December). Cross-scale self-supervised blind image deblurring via implicit neural representation. In The Thirty-eighth Annual Conference on Neural Information Processing Systems (Vol. 3).
> >
> > [6] Xiao, Y., Shen, Y., Liao, S., Yao, B., Cai, X., Zhang, Y., & Gao, F. (2025). Limited-view photoacoustic imaging reconstruction via high-quality self-supervised neural representation. Photoacoustics, 42, 100685.
> >
> > [7] Yang, S., Ou, Y., & Okutomi, M. (2026). PixelINR: Scan-specific self-supervised MRI reconstruction based on implicit neural representations. Biomedical Signal Processing and Control, 112, 108838.

---

> > > ### Comment · Reviewer_rkTN · 2025-11-27
> > >
> > > Thanks for the quick experiments. That solves all my concerns. I have increased my rating to 8.

---

> > > > ### Author Response · Authors · 2025-11-27
> > > > **Response to Reviewer rkTN**
> > > >
> > > > We thank the reviewer again for the constructive comments and suggestions!

---

### Official Review · Reviewer_APdc · 2025-10-30

**Soundness:** 2
**Presentation:** 3
**Contribution:** 2
**Rating:** 4
**Confidence:** 3

**Summary:**

This paper proposes Tucker-FNO, a neural operator that decomposes high-dimensional Fourier Neural Operators (FNO) into multiple 1-dimensional FNOs using Tucker decomposition. The method targets computational efficiency for partial differential equations (PDEs) and signal restoration tasks while maintaining expressiveness.

Tucker-FNO decomposes a d-dimensional FNO into d separate 1-dimensional FNOs arranged in Tucker format. A pre-lifting module extracts factor inputs from the condition function, processing them through individual 1-dimensional FNOs before aggregating outputs via Tucker decomposition. This reduces computational complexity from O(n³ log n³) to O(3 n log n) for 3D PDEs. The authors establish a universal approximation theorem using functional decomposition tools in Sobolev spaces, proving Tucker-FNO can approximate any continuous operator.

**Strengths:**

- Significant efficiency gains: Reduces parameters by and execution time compared to standard FNO on 256×256 data while improving performance​.

- The paper is well-written.

- Rigorous theoretical foundation: First tensor-decomposed neural operator with proven universal approximation capability​.

- Dual applications: Successfully handles both PDE approximation and signal restoration tasks​.

- Good empirical performance: Outperforms FNO, D-FNO, and Com-FNO on Navier-Stokes, Plasticity, and Burger's equations​.

- Better high-frequency signal handling: Shows advantages in regions with significant numerical variation​.

**Weaknesses:**

- Limited novelty in decomposition: Applies existing Tucker decomposition techniques to FNOs without fundamentally new mathematical insights. Tensorized FNOs, which use Tucker decomposition have already been implemented as part of the neuraloperator library and also published in the literature. While this in itself is not a problem, there is no mention of prior related works such as [1].

- Exprimental evaluation is mainly conducted on toy datasets and more complex examples are omitted. Baselines are also limited and lack other tensorized approaches such as [1] or the factorized FNO. This should also include comparisons to other tensor decomposition than just Tucker (e.g., tensor train, CP decomposition)

- Moreover, it is unclear whether this approach can hold in more complex settings which include geometry. It's quite likely that this approach would fall apart in such a setting, given that it breaks symmetry and even just a single test with a simple geometry would be very informative.

- Restrictive assumptions: Universal approximation theorem requires analyticity on compact sets (Theorem 3), limiting applicability to non-smooth or unbounded domains​.

- Additional hyperparameter sensitivity: Performance depends on Tucker rank selection and frequency truncation threshold, requiring careful tuning​. It's unclear how the rank should be set for a given problem.

- Scalability questions: While efficient for moderate dimensions, scaling behavior for very high-dimensional problems (d > 10) remains unexplored

[1] Kossaifi, J., Kovachki, N., Azizzadenesheli, K., & Anandkumar, A. (2023). Multi-grid tensorized fourier neural operator for high-resolution pdes. arXiv preprint arXiv:2310.00120.

**Questions:**

- I wonder why the authors chose these examples and whether they explored other tensor decompositions. It would also be interested to include such results in the theory section

---

> ### Author Response · Authors · 2025-11-20
> **Response to Reviewer APdc Part 1**
>
> We sincerely thank the reviewer for the valuable and constructive comments! We have carefully made more discussions and clarifications according to your comments.
>
> For easy reference, a list of responses to the reviewers' comments has been compiled, and the corresponding changes have been highlighted in **blue** in the main text for easy reference of our revision. Should you need further information, please let us know. We look forward to hearing from you soon.
>
> >**W1: Limited novelty in decomposition: Applies existing Tucker decomposition techniques to FNOs without fundamentally new mathematical insights. Tensorized FNOs, which use Tucker decomposition have already been implemented as part of the neuraloperator library and also published in the literature. While this in itself is not a problem, there is no mention of prior related works such as T-FNO**
>
> R1: Thanks! We have provided a detailed comparison between tensorized FNO and Tucker-FNO in the Section 3.6 of the revised manuscript. Here, we offer a brief summary.
> Although both recent tensorized FNOs and our Tucker-FNO employ Tucker decomposition methods, the **implementation** and **motivation** are significantly different:
>
> 1. **Tensorized FNOs:** Tensorized FNOs utilize decomposition to **reduce the model parameters**, where decomposition is applied to **the model parameter**.
> Specifically, the tensorized FNOs decompose the model parameters, i.e., $\mathcal{R}=\mathcal{C}\times\_1\mathcal{U}\_1\times\_2\cdots\times\_{d+2} \mathcal{U}\_{d+2}$, where $\mathcal{R}\in\mathbb{R}^{k\times\cdots\times k \times d_v\times d_v}$ is the parameter tensor, $\mathcal{C}\in\mathbb{R}^{r_1\times\cdots\times r_{d+2}}$ is the core tensor, and $\mathcal{U}_i\in\mathbb{R}^{k\times r_i}$ (for $i\leq d$) or $\mathcal{U}_i\in \mathbb{R}^{d_v\times r_i}$ (for $i>d$).
> Here, for simplicity, we analyze only a single-layer FNO.
> The parameter number of tensorized FNO is $O(\hat{r}^{d+2} +\hat{r}(d_v+k))$.
> And its computational complexity for FFT and inverse FFT still equals to the traditional FNO, i.e., $O(d_vd n^d\log(n))$ for $d$-dimensional PDE problems.
> Moreover, due to the decomposition, its computation complexity for multiplying weights increases from $O(d_v^2k^d)$ to $O(\hat{r}d_v^2k^d)$.
>
> 2. **Tucker-FNO:** Our Tucker-FNO utilizes decomposition to **reduce the computational complexity**, where decomposition is applied to **the operator**. By decomposing the $d$-dimensional FNO into a series of $1$-dimensional FNOs, the computational complexity for FFT and inverse FFT is reduced to $O(d_vdn\log(n))$, and that for multiplying weights is reduced to $O(d_v^2dk)$.
>
> In summary, tensorized FNO is a new parameterization scheme for FNO's network parameters, with its underlying principles remaining similar to traditional FNO, and its motivation is to design a parameterization scheme with reduced the parameter number for FNO.
> Our Tucker-FNO, on the other hand, is a novel paradigm under the FNO framework. Especially, our motivation is to design a new neural operator paradigm with reduced computational complexity.
>
> ||Tensorized FNO |Tucker-FNO|
> |-|-|-|
> |FFT Computational Complexity| $O(d_vn^d\log(n))$ **(equal to FNO)** | $O(d_vn\log(n))$ **(less than FNO)**|
> |Multiplying Weights| $O(\hat{r}d_v^2k^d)$ **(higher than FNO)** | $O(d_v^2dk)$ **(less than FNO)**|
> |Parameter Number| $O(\hat{r}^{d+2}+\hat{r}(k+d_v))$ | $O(d_v^2dk)$ |
> |Decomposition|Model Weight|Operator|
> |Contribution| A parameterization scheme for FNO | A new computationally efficient FNO with UAT |
>
> Furthermore, we present the efficiency comparisons between FNO, T-FNO [1] (a classical tensorized FNO) and Tucker-FNO in the table below (i.e., Table 3 in the revised manuscript).
> T-FNO’s rank rate is $0.4$, which indicates that the Tucker rank of each dimension is $0.4$ of the size of the dimension.
> The results indicate that although T-FNO can reduce model parameters, it certainly increases computational complexity. Compared with tensorized FNO, Tucker-FNO can both reduce computational complexity and the model parameters by decomposing the operator.
> |Methods|Params|Time/iter|
> |-|-|-|
> |FNO|593.7 K|0.0685|
> |T-FNO|232.6 K|0.0746|
> |Tucker-FNO|293.5 K|0.0352|
>
> [1] Kossaifi, J., Kovachki, N. B., Azizzadenesheli, K., & Anandkumar, A. Multi-Grid Tensorized Fourier Neural Operator for High-Resolution PDEs. Transactions on Machine Learning Research.

---

> ### Author Response · Authors · 2025-11-20
> **Response to Reviewer APdc Part 2**
>
> >**W2:Exprimental evaluation is mainly conducted on toy datasets and more complex examples are omitted. Baselines are also limited and lack other tensorized approaches such as MG-TFNO or the factorized FNO. This should also include comparisons to other tensor decomposition than just Tucker (e.g., tensor train, CP decomposition)**
>
> >>**W2.1: Experimental evaluation is mainly conducted on toy datasets and more complex examples are omitted.**
>
> R2.1: Thanks!
> We will evaluate Tucker-FNO on a more complex geometry example in response to your comment W3.
>
> >>**W2.2: Baselines are also limited and lack other tensorized approaches such as MG-TFNO or the factorized FNO.**
>
> Thanks! In fact, tensorized FNO and Tucker-FNO hold essential distinctions in terms of motivation and implementation.
> Instead of comparing to tensorized FNO, we have evaluated our approach against D-FNO [2], a more recent method that is more related to our methods, and achieved significant performance improvements.
>
> Following your suggestion, we have compared the T-FNO [1], which is a recent tensorized FNO method, with our Tucker-FNO.
> Here, T-FNO’s rank rate is set as $0.4$, which indicates that the Tucker rank of each dimension is $0.4$ of the size of the dimension, and other hyperparameters are the same as FNO.
> The results are demonstrated in the table below (i.e., Table 2 in the revised manuscript).
> These results demonstrate that Tucker-FNO still holds advantages compared to T-FNO.
>
> | |N-S (T=20)|N-S (T=20)|N-S (T=30)|N-S (T=30)|Plasticity|Plasticity|Burger|Burger|
> |-|-|-|-|-|-|-|-|-|
> |Methods|NMSE|VCE| NMSE|VCE| NMSE|VCE| NMSE|VCE|
> |ResNet|0.0911|0.0053|0.2582|0.0351|0.1015|0.0587|0.0941|0.0651
> |FNO|0.0034|1.0311e-5|0.0072|3.9776e-5|0.0080|1.9514e-4|0.0087|4.4465e-5
> |Com-FNO|0.0036|1.1265e-5|0.0071|3.8146e-5|0.0084|2.0897e-4|0.0063|2.4543e-5
> |T-FNO|0.0035|8.8214e-6|0.0065|3.0354e-5|0.0079|2.0076e-4|0.0075|2.9672e-5
> |D-FNO|0.0063|3.0994e-5|0.0132|9.1426e-5|0.0097|2.2150e-4|0.0071|2.8610e-5
> |Tucker-FNO|0.0028|8.2254e-6|0.0061|2.9742e-5|0.0073|2.0010e-4|0.0070|2.0050e-5
>
> >>**W2.3: This should also include comparisons to other tensor decomposition than just Tucker (e.g., tensor train, CP decomposition)**
>
> Thanks! As suggested, we have also applied our methodological paradigm to CP tensor decomposition (referred to as CP-FNO).
> The table below (i.e., Table 13 in the revised manuscript) demonstrates the results of CP-FNO in Plasticity equation.
> Although the efficiency of CP decomposition is higher than that of Tucker decomposition, it has a relatively inferior accuracy. This can be attributed to the stronger representation ability of the Tucker decomposition.
>
> |Methods | NMSE | VCE |
> |-|-|-|
> |FNO|0.0080|1.9514e-4|
> |D-FNO|0.0097|2.2150e-4|
> |Tucker-FNO|0.0073|2.0010e-4|
> |CP-FNO|0.0082|2.0021e-4|
>
> [1] Kossaifi, J., Kovachki, N. B., Azizzadenesheli, K., & Anandkumar, A. Multi-Grid Tensorized Fourier Neural Operator for High-Resolution PDEs. Transactions on Machine Learning Research.
>
> [2] Kangjie Li and Wenjing Ye. D-fno: A decomposed fourier neural operator for large-scale parametric partial differential equations. Computer Methods in Applied Mechanics and Engineering, 436: 117732, 2025.

---

> ### Author Response · Authors · 2025-11-20
> **Response to Reviewer APdc Part 3**
>
> >**W3: Moreover, it is unclear whether this approach can hold in more complex settings which include geometry. It's quite likely that this approach would fall apart in such a setting, given that it breaks symmetry and even just a single test with a simple geometry would be very informative.**
>
> R3: Thanks!
> According to your suggestion, we have evaluated our Tucker-FNO in a more complex setting which includes geometry.
> Here, we replace the FNO with our Tucker-FNO in GINO [3], and evaluate it in a large-scale computational fluid dynamics dataset from [3], which is a 3-D PDE problem.
> The input shapes of the dataset are from the ShapeNet Car category [4].
> The preliminary results under 10 training epochs are shown in the table below (i.e., Table 9 in the revised manuscript), and the visualizations are demonstrated in Figure 9 of the revised manuscript.
> The results demonstrate that our Tucker-FNO remains effective for **large-scale 3-D PDEs**, which include complex geometry.
>
> |Methods|Test Error|
> |-|-|
> |GINO |0.2047|
> |GINO-D-FNO|0.2187|
> |GINO-Tucker-FNO|0.2031|
>
> >**W4: Restrictive assumptions: Universal approximation theorem requires analyticity on compact sets (Theorem 3), limiting applicability to non-smooth or unbounded domains.**
>
> R4: Thanks! Similar to recent works w.r.t. the universal approximation theorem (UAT) for operator learning [5,6] and functional decomposition [7], our proof also depends on the mild smoothness assumption.
> However, this does not imply that our method has limited applicability in non-smooth or unbounded domains.
> Our experiments w.r.t. signal restoration (e.g., Figure 3, 10, 11 in the revised manuscript) has demonstrated that Tucker-FNO can effectively learn the high-frequency components in the signal.
> The high-frequency components in signal often represents the domain with weaker smoothness or even non-smoothness in the real-world data.
> The results w.r.t. Plasticity equation (i.e., Figure 8 in the revised manuscript) have also verified the robust performance of our methods in regions with significant numerical variation.
>
> [3] Li, Z., Kovachki, N., Choy, C., Li, B., Kossaifi, J., Otta, S., Nabian, M., Stadler, M., Hundt, C., Azizzadenesheli, K., Anandkumar, A. (2023) Geometry-Informed Neural Operator for Large-Scale 3D PDEs. NeurIPS 2023.
>
> [4] Angel X Chang, Thomas Funkhouser, Leonidas Guibas, Pat Hanrahan, Qixing Huang, Zimo Li, Silvio Savarese, Manolis Savva, Shuran Song, Hao Su, et al. Shapenet: An information-rich 3d model repository. arXiv preprint arXiv:1512.03012, 2015.
>
> [5] S. Lanthaler, S. Mishra, and G. E. Karniadakis, Error estimates for DeepOnets: A deep learning framework in infinite dimensions, 2021.
>
> [6] Kovachki, N., Lanthaler, S., & Mishra, S. (2021). On universal approximation and error bounds for Fourier neural operators. Journal of Machine Learning Research, 22(290), 1-76.
>
> [7] Luo, Y., Zhao, X., Li, Z., Ng, M. K., & Meng, D. (2023). Low-rank tensor function representation for multi-dimensional data recovery. IEEE transactions on pattern analysis and machine intelligence, 46(5), 3351-3369.

---

> ### Author Response · Authors · 2025-11-20
> **Response to Reviewer APdc Part 4**
>
> >**W5: Additional hyperparameter sensitivity: Performance depends on Tucker rank selection and frequency truncation threshold, requiring careful tuning. It's unclear how the rank should be set for a given problem.**
>
> R5: Thanks! For most matrix and tensor decomposition approaches, the setting of rank is a major challenge in tensor decomposition methods. In fact, early researches [8] have proven that computing the tensor rank is an NP-hard problem.
> Recent researches [9,10] analyzed the selection for decomposition rank in the tensor completion task.
> Tensor-decomposition-based neural network methods still lack of relevant research.
> Following the recent tensor-decomposition-based neural network methods [11], we pre-set the rank to a fixed value.
>
> We have evaluated Tucker-FNO’s latent dimension $d_v$, Tucker rank, and frequency truncation threshold $k$ in Table 6 and 7 in the revised manuscript. We also paste the tables here for your convenience.
> The first table below (i.e., Table 6 in the revised manuscript) demonstrates the influence of the latent dimension $d_v$ and the frequency truncation threshold $k$, where the Tucker rank is the same as $d_v$.
> From the results, performance improves with both $k$ and $d_v$, with $d_v$ having a greater influence.
> The second table below (i.e., Table 7 in the revised manuscript) shows the influence of Tucker rank and latent dimension $d_v$, where we set the decomposition rank and the latent dimension $d_v$ as independent hyperparameters.
> From the results, we can see that our method is relatively more robust w.r.t. decomposition rank than latent dimension $d_v$.
> And, Tucker-FNO demonstrates relatively satisfactory performances for a range of hyperparameter selections of rand and $d_v$.
>
> Consequently, Tucker-FNO’s latent dimension $d_v$, Tucker rank, and frequency truncation threshold $k$ exhibit a positive correlation with the Tucker-FNO’s performance, and the Tucker rank and $k$ demonstrate relatively greater robustness. Meanwhile, increases in these hyperparameters also lead to a larger model size and a heavier computation cost. In summary, the selection of model hyperparameters indicates a trade-off between efficiency and performance, and should be considered based on the specific application scenario.
>
> |$k$ | 8 | 8 | 12 | 12 | 16 | 16
> |-|-|-|-|-|-|-|
> $d_v$ & Rank | NMSE | VCE | NMSE | VCE | NMSE | VCE |
> 16 | 0.0098 | 2.07e-4 | 0.0092 | 2.06e-4 | 0.0092 | 2.06e-5 |
> 32 | 0.0074 | 2.18e-4 | 0.0074 | 2.00e-4 | 0.0073 | 1.98e-4 |
> 64 | 0.0068 | 2.45e-4 | 0.0064 | 2.04e-4 | 0.0062 | 2.00e-4 |
>
>
> |Rank|16|16|32|32|64|64|
> |-|-|-|-|-|-|-|
> |$d_v$|NMSE|VCE|NMSE|VCE|NMSE|VCE|
> |16|0.0092|2.06e-4|0.0087|1.84e-4|0.0086|1.97e-5|
> |32|0.0079|2.03e-4|0.0074|2.00e-4|0.0071|1.78e-4|
> |64|0.0064|1.87e-4|0.0064|2.00e-4|0.0063|2.04e-4|
>
> [8] J. Håstad, “Tensor rank is NP-complete,” in Proc. Int. Colloq. Automata,
> Lang., Program. Berlin, Germany: Springer, 1989, pp. 451–460.
>
> [9] Y. -L. Chen, C. -T. Hsu and H. -Y. M. Liao, "Simultaneous Tensor Decomposition and Completion Using Factor Priors," in IEEE Transactions on Pattern Analysis and Machine Intelligence, vol. 36, no. 3, pp. 577-591, March 2014, doi: 10.1109/TPAMI.2013.164.
>
> [10] J. Yu, G. Zhou, W. Sun and S. Xie, "Robust to Rank Selection: Low-Rank Sparse Tensor-Ring Completion," in IEEE Transactions on Neural Networks and Learning Systems, vol. 34, no. 5, pp. 2451-2465, May 2023, doi: 10.1109/TNNLS.2021.3106654.
>
> [11] Y. Luo, X. Zhao, Z. Li, M. K. Ng and D. Meng, "Low-Rank Tensor Function Representation for Multi-Dimensional Data Recovery," in IEEE Transactions on Pattern Analysis and Machine Intelligence, vol. 46, no. 5, pp. 3351-3369, May 2024, doi: 10.1109/TPAMI.2023.3341688.

---

> ### Author Response · Authors · 2025-11-20
> **Response to Reviewer APdc Part 5**
>
> >**W6: Scalability questions: While efficient for moderate dimensions, scaling behavior for very high-dimensional problems (d > 10) remains unexplored.**
>
> R6: Thanks! The efficiency burden in solving high-dimensional PDEs is indeed a significant problem. We now further demonstrate the efficiency of our Tucker-FNO in a 10-D toy setting. Here, we compare the single-layer FNO and Tucker-FNO, where $d_v=4$ and $k=2$, in a 10-D PDE approximation problem in terms of efficiency. The shape of input is $4\times 4\times\cdots\times 4$. The comparison of parameter number and running time is demonstrated in the table below.
> Although our efficiency advantage diminishes in the 10-D PDE scenario, our approach still holds certain advantages.
>
> Currently, most research on Neural Operators (NO) focuses on 3-D or lower-dimensional PDEs, while studies addressing the solution of very high-dimensional PDEs remain relatively scarce. Methods such as [12] often require specialized designs to handle high-dimensional (>5-D) PDEs. In the future, we will further explore the use of Tucker-FNO to improve the efficiency of solving genuinely high-dimensional PDEs.
>
> |Methods|Params|Time (seconds)|
> |-|-|-|
> |FNO|26M|0.0093|
> |Tucker-FNO|22M|0.0061|
>
> [12] Luo, D., O’Leary-Roseberry, T., Chen, P., & Ghattas, O. (2025). Efficient PDE-constrained optimization under high-dimensional uncertainty using derivative-informed neural operators. SIAM Journal on Scientific Computing, 47(4), C899-C931.
>
> >**W7: I wonder why the authors chose these examples and whether they explored other tensor decompositions. It would also be interested to include such results in the theory section.**
>
> R7: Thanks!
> We have already discussed preliminarily the reasons for choosing the Tucker decomposition  in response for your comment W2.3, and we will further discuss it here.
> The main reasons we chose Tucker-FNO are as follows：
>
> 1. Recent research [11] has proved the effectiveness and efficiency of functional Tucker decomposition for signal restoration in real-world scenarios.
>
> 2. From the results of CP-FNO (shown in the table below, i.e., Table 13 in the revised manuscript), although the efficiency of CP decomposition is higher than that of Tucker decomposition, CP-FNO compromises on the performance. Compared with CP-FNO, Tucker-FNO achieves a better balance between efficiency and effectiveness.
>
> |Methods | NMSE | VCE |
> |-|-|-|
> |FNO|0.0080|1.9514e-4|
> |D-FNO|0.0097|2.2150e-4|
> |Tucker-FNO|0.0073|2.0010e-4|
> |CP-FNO|0.0082|2.0021e-4|
>
> Furthermore, we believe that UAT of CP-FNO can also be proved in a similar manner, as the UAT of the functional CP decomposition has been discussed in [13].
> Thank you for your valuable suggestions!
>
> [11] Y. Luo, X. Zhao, Z. Li, M. K. Ng and D. Meng, "Low-Rank Tensor Function Representation for Multi-Dimensional Data Recovery," in IEEE Transactions on Pattern Analysis and Machine Intelligence, vol. 46, no. 5, pp. 3351-3369, May 2024, doi: 10.1109/TPAMI.2023.3341688.
>
> [13] N. Kargas and N. D. Sidiropoulos, "Supervised Learning and Canonical Decomposition of Multivariate Functions," in IEEE Transactions on Signal Processing, vol. 69, pp. 1097-1107, 2021, doi: 10.1109/TSP.2021.3055000.

---

> ### Author Response · Authors · 2025-11-20
> **Response to Reviewer APdc Part 6**
>
> Should you require any further information, please do not hesitate to let us know. We would greatly appreciate your feedback.
>
> >**Reference**
>
> [1] Kossaifi, J., Kovachki, N. B., Azizzadenesheli, K., & Anandkumar, A. Multi-Grid Tensorized Fourier Neural Operator for High-Resolution PDEs. Transactions on Machine Learning Research.
>
> [2] Kangjie Li and Wenjing Ye. D-fno: A decomposed fourier neural operator for large-scale parametric partial differential equations. Computer Methods in Applied Mechanics and Engineering, 436: 117732, 2025.
>
> [3] Li, Z., Kovachki, N., Choy, C., Li, B., Kossaifi, J., Otta, S., Nabian, M., Stadler, M., Hundt, C., Azizzadenesheli, K., Anandkumar, A. (2023) Geometry-Informed Neural Operator for Large-Scale 3D PDEs. NeurIPS 2023.
>
> [4] Angel X Chang, Thomas Funkhouser, Leonidas Guibas, Pat Hanrahan, Qixing Huang, Zimo Li, Silvio Savarese, Manolis Savva, Shuran Song, Hao Su, et al. Shapenet: An information-rich 3d model repository. arXiv preprint arXiv:1512.03012, 2015.
>
> [5] S. Lanthaler, S. Mishra, and G. E. Karniadakis, Error estimates for DeepOnets: A deep learning framework in infinite dimensions, 2021.
>
> [6] Kovachki, N., Lanthaler, S., & Mishra, S. (2021). On universal approximation and error bounds for Fourier neural operators. Journal of Machine Learning Research, 22(290), 1-76.
>
> [7] Luo, Y., Zhao, X., Li, Z., Ng, M. K., & Meng, D. (2023). Low-rank tensor function representation for multi-dimensional data recovery. IEEE transactions on pattern analysis and machine intelligence, 46(5), 3351-3369.
>
> [8] J. Håstad, “Tensor rank is NP-complete,” in Proc. Int. Colloq. Automata,
> Lang., Program. Berlin, Germany: Springer, 1989, pp. 451–460.
>
> [9] Y. -L. Chen, C. -T. Hsu and H. -Y. M. Liao, "Simultaneous Tensor Decomposition and Completion Using Factor Priors," in IEEE Transactions on Pattern Analysis and Machine Intelligence, vol. 36, no. 3, pp. 577-591, March 2014, doi: 10.1109/TPAMI.2013.164.
>
> [10] J. Yu, G. Zhou, W. Sun and S. Xie, "Robust to Rank Selection: Low-Rank Sparse Tensor-Ring Completion," in IEEE Transactions on Neural Networks and Learning Systems, vol. 34, no. 5, pp. 2451-2465, May 2023, doi: 10.1109/TNNLS.2021.3106654.
>
> [11] Y. Luo, X. Zhao, Z. Li, M. K. Ng and D. Meng, "Low-Rank Tensor Function Representation for Multi-Dimensional Data Recovery," in IEEE Transactions on Pattern Analysis and Machine Intelligence, vol. 46, no. 5, pp. 3351-3369, May 2024, doi: 10.1109/TPAMI.2023.3341688.
>
> [12] Luo, D., O’Leary-Roseberry, T., Chen, P., & Ghattas, O. (2025). Efficient PDE-constrained optimization under high-dimensional uncertainty using derivative-informed neural operators. SIAM Journal on Scientific Computing, 47(4), C899-C931.
>
> [13] N. Kargas and N. D. Sidiropoulos, "Supervised Learning and Canonical Decomposition of Multivariate Functions," in IEEE Transactions on Signal Processing, vol. 69, pp. 1097-1107, 2021, doi: 10.1109/TSP.2021.3055000.

---

> ### Comment · Reviewer_APdc · 2025-11-24
>
> I thank the reviewers for their comments. I misunderstood the main point about how this approach reduces the computational complexity of the operator as opposed to T-FNO.
>
> I thank the authors for their detailed rebuttal and addressing the misconceptions. I think the modified manuscript is clearer and avoid possible confusion between the two approaches which see similar on the surface level. I have updated my score accordingly.

---

> > ### Author Response · Authors · 2025-11-25
> > **Response to Reviewer APdc**
> >
> > We thank the reviewer again for the constructive comments and suggestions!

---

### Official Review · Reviewer_ofCR · 2025-11-01

**Soundness:** 3
**Presentation:** 3
**Contribution:** 3
**Rating:** 6
**Confidence:** 3

**Summary:**

This paper proposes Tucker-FNO, a tensor-decomposed variant of the Fourier Neural Operator (FNO) designed to improve efficiency and scalability in high-dimensional operator learning. The main idea is to employ Tucker decomposition to represent a high-dimensional Fourier operator as a composition of a low-dimensional core tensor and multiple independent 1D FNOs. This formulation substantially reduces computational costs, particularly the cubic scaling of high-dimensional FFTs, while maintaining expressive power. In addition to the algorithmic design, the paper establishes a universal approximation theorem (UAT) for Tucker-FNO using tools from functional analysis in Sobolev spaces, providing a theoretical guarantee for its representational capacity. Empirical results on several PDE benchmarks, including Navier–Stokes, Burgers’, and Plasticity equations, as well as continuous signal learning tasks, show that Tucker-FNO achieves improved efficiency and accuracy over standard FNOs.

**Strengths:**

The paper’s strength lies in its clear and well-motivated attempt to address one of the key limitations of FNOs—their inefficiency in high-dimensional function spaces. By leveraging Tucker decomposition, the proposed model reduces both memory and computational complexity while preserving the operator’s structural expressiveness. This approach is both elegant and practical, as it connects classical tensor factorization with modern neural operator design.
The inclusion of a rigorous universal approximation theorem significantly strengthens the contribution by grounding the empirical observations in a solid theoretical foundation. This theorem not only supports Tucker-FNO’s representational power but also provides a framework for future theoretical extensions.
Experimentally, Tucker-FNO demonstrates consistent improvement in speed and accuracy compared to baseline FNOs, with notably lower FFT costs in three-dimensional settings. The paper also extends Tucker-FNO to continuous signal learning via implicit neural representations (INRs), showing that the proposed decomposition generalizes well to non-PDE tasks. Overall, the method provides both practical computational benefits and theoretical depth, positioning it as a meaningful advance in scalable operator learning.

**Weaknesses:**

While the paper makes a valuable contribution, there are several limitations that should be addressed. First, although the method is advertised as being effective for high-dimensional settings, most experiments are conducted on relatively low-dimensional PDEs (2D or 3D). The current results therefore do not convincingly demonstrate scalability to genuinely high-dimensional problems (e.g., 5D+), which is central to the paper’s motivation.
Second, rank selection in Tucker decomposition is a critical factor for both performance and efficiency. The paper treats Tucker ranks as fixed hyperparameters but does not provide a mechanism to learn or adapt them during training. Over- or under-factorization can degrade performance or efficiency, and the absence of an adaptive rank selection strategy limits the robustness of the approach.
Moreover, while the theoretical result is a strong addition, the practical interpretability of the UAT remains limited — it guarantees existence but does not clarify how rank or decomposition depth affect approximation accuracy in practice. Finally, when Tucker ranks are large, the method may still suffer from bottlenecks due to the size of the core tensor, potentially reducing the claimed computational advantage.

**Questions:**

- Can the Tucker ranks be learned or dynamically adapted during training, rather than being fixed a priori?
- How does Tucker-FNO behave in genuinely high-dimensional PDEs (e.g., 5D or higher)? Are there benchmark results or computational analyses for such settings?
- For large Tucker ranks, does the core tensor introduce new bottlenecks that offset the decomposition’s computational gains?
- In the universal approximation theorem, can the approximation error be explicitly related to Tucker rank or decomposition depth?

---

> ### Author Response · Authors · 2025-11-20
> **Response to Reviewer ofCR Part 1**
>
> We sincerely thank the reviewer for the valuable and constructive comments! We have carefully made more discussions and clarifications according to your comments.
>
> For easy reference, a list of responses to the reviewers' comments has been compiled, and the corresponding changes have been highlighted in **blue** in the main text for easy reference of our revision. Should you need further information, please let us know. We look forward to hearing from you soon.
>
> >**W1: Can the Tucker ranks be learned or dynamically adapted during training, rather than being fixed a priori?**
>
> R1: Thanks! For most matrix and tensor decomposition approaches, the setting of rank is a major challenge in tensor decomposition methods. In fact, early research [1] has proven that computing the tensor rank is an NP-hard problem.
> Recent researches [2,3] analyzed the selection for decomposition rank in the tensor completion task.
> Tensor-decomposition-based neural network methods still lack relevant research.
> Following recent tensor-decomposition-based neural network methods [4], we pre-set the rank to a fixed value.
>
> We now further demonstrate that our Tucker-FNO is not overly sensitive to the selection of rank.
> The table below (i.e., Table 7 in the revised manuscript) shows the results of NMSE with respect to rank and latent dimension $d_v$, where we set the decomposition rank and the latent dimension $d_v$ as independent hyperparameters.
> From the results, we can see that our method is relatively more robust w.r.t. decomposition rank than latent dimension $d_v$.
> And, Tucker-FNO demonstrates relatively satisfactory performances for a range of hyperparameter selections of rank and $d_v$.
>
> |Rank|16|16|32|32|64|64|
> |-|-|-|-|-|-|-|
> |$d_v$|NMSE|VCE|NMSE|VCE|NMSE|VCE|
> |16|0.0092|2.06e-4|0.0087|1.84e-4|0.0086|1.97e-5|
> |32|0.0079|2.03e-4|0.0074|2.00e-4|0.0071|1.78e-4|
> |64|0.0064|1.87e-4|0.0064|2.00e-4|0.0063|2.04e-4|
>
> [1] J. Håstad, “Tensor rank is NP-complete,” in Proc. Int. Colloq. Automata,
> Lang., Program. Berlin, Germany: Springer, 1989, pp. 451–460.
>
> [2] Y. -L. Chen, C. -T. Hsu and H. -Y. M. Liao, "Simultaneous Tensor Decomposition and Completion Using Factor Priors," in IEEE Transactions on Pattern Analysis and Machine Intelligence, vol. 36, no. 3, pp. 577-591, March 2014, doi: 10.1109/TPAMI.2013.164.
>
> [3] J. Yu, G. Zhou, W. Sun and S. Xie, "Robust to Rank Selection: Low-Rank Sparse Tensor-Ring Completion," in IEEE Transactions on Neural Networks and Learning Systems, vol. 34, no. 5, pp. 2451-2465, May 2023, doi: 10.1109/TNNLS.2021.3106654.
>
> [4] Y. Luo, X. Zhao, Z. Li, M. K. Ng and D. Meng, "Low-Rank Tensor Function Representation for Multi-Dimensional Data Recovery," in IEEE Transactions on Pattern Analysis and Machine Intelligence, vol. 46, no. 5, pp. 3351-3369, May 2024, doi: 10.1109/TPAMI.2023.3341688.

---

> ### Author Response · Authors · 2025-11-20
> **Response to Reviewer ofCR Part 2**
>
> >**W2: How does Tucker-FNO behave in genuinely high-dimensional PDEs (e.g., 5D or higher)? Are there benchmark results or computational analyses for such settings?**
>
> R2: Thanks! The efficiency burden in solving high-dimensional PDEs is indeed a significant problem. We now further demonstrate the efficiency of our Tucker-FNO in a $10$-D toy setting. Here, we compare the single-layer FNO and Tucker-FNO, where $d_v=4$ and $k=2$, in a $10$-D PDE approximation problem in terms of efficiency. The shape of input is $4\times 4\times\cdots\times 4$. The comparison of parameter number and running time is demonstrated in the table below.
> Although our efficiency advantage diminishes in the $10$-D PDE scenario, our approach still holds certain advantages.
>
> Currently, most research on Neural Operators (NO) focuses on 3-D or lower-dimensional PDEs, while studies addressing the solution of very high-dimensional PDEs remain relatively scarce. Methods such as [5] often require specialized designs to handle high-dimensional (>5-D) PDEs. In the future, we aim to explore the use of Tucker-FNO to improve the efficiency of solving genuinely high-dimensional PDEs.
>
> |Methods|Params|Time (seconds)|
> |-|-|-|
> |FNO|26M|0.0093|
> |Tucker-FNO|22M|0.0061|
>
> [5] Luo, D., O’Leary-Roseberry, T., Chen, P., & Ghattas, O. (2025). Efficient PDE-constrained optimization under high-dimensional uncertainty using derivative-informed neural operators. SIAM Journal on Scientific Computing, 47(4), C899-C931.

---

> ### Author Response · Authors · 2025-11-20
> **Response to Reviewer ofCR Part 3**
>
> >**W3: For large Tucker ranks, does the core tensor introduce new bottlenecks that offset the decomposition’s computational gains?**
>
> R3: Thanks! The decomposition's computational complexity is $O( \hat{r} d_vn^d)$, where $\hat{r}=\max\\{r_1,\cdots,r_d\\}$ is the rank, $d_v$ is the latent dimension and $n$ is the size of input. This cost is indeed related to the computation involving the core tensor.
> Given this computation cost, an efficiency bottleneck only occurs when the rank is set to a sufficiently large value.
> In the table below (i.e., Table 13 in the revised manuscript), we demonstrate the running time of the lifting, FNO processing, decomposition, and post-projection in FNO and Tucker-FNO, where $d_v=16$ and $k=12$. We set a $2$-D PDE problem where the size of the input is $128\times 128$ with batch size $8$.
> To highlight the differences, we test the running time on the CPU.
> The results demonstrate that the bottleneck of the decomposition module occurs when the rank is extremely large (>1024), which far exceeds the rank settings (~32) of our method.
> Consequently, for Tucker-FNO, the primary efficiency burden still stems from FNO module, particularly the FFT in it.
>
> |Methods|Lifting|FNO Processing|Decomposition|Projection|Total Time|Param|
> |-|-|-|-|-|-|-|
> |FNO|0.0010|0.4057|-|0.0935|0.5002|593281|
> |Tucker-FNO (rank=16)|0.0008|0.1761|0.0005|0.0423|0.2197|70593|
> |Tucker-FNO (rank=64)|0.0011|0.2753|0.0011|0.0597|0.3372|170529|
> |Tucker-FNO (rank=256)|0.0009|0.2736|0.0088|0.0691|0.3551|1307553|
> |Tucker-FNO (rank=1024)|0.0010|0.3114|0.1412|0.0648|0.5184|17652129|
>
> The more detailed analysis is demonstrated in Section 3.6 in the revised manuscript.

---

> ### Author Response · Authors · 2025-11-20
> **Response to Reviewer ofCR Part 4**
>
> >**W4: In the universal approximation theorem, can the approximation error be explicitly related to Tucker rank or decomposition depth?**
>
> R4: Thanks! Theoretically, the approximation error can be estimated by the Tucker rank.
> From the proof of Theorem 3, the error consists of two components: (1) errors from the FNO in Lemma 2; (2) errors from the operator Tucker decomposition in Theorem 2. Both of them can be estimated:
>
> 1. For FNO, explicit error bounds are derived to show that the size of the FNO, approximating operators associated with a Darcy type elliptic PDE and with the incompressible Navier-Stokes equations of fluid dynamics, only increases sub (log)-linearly in terms of the reciprocal of the error [6].
>
> 2. For operator Tucker decomposition, the relationship between approximation error and the Tucker rank can be established based on Theorem 1, which presents the universal approximation theorem (UAT) for functional Tucker decomposition. In Equation (50), we construct an approximatio of functional Tucker tensor decomposition using the Stone–Weierstrass Theorem. The associated error under this theorem has been well-studied in classical approximation theory [7]. Leveraging the error bounds provided by the Stone–Weierstrass Theorem, we can derive the error estimates presented in Theorem 1, Lemma 10, and Theorem 2.
>
> Therefore, by the error bounds in Lemma 2 and Theorem 2, we can derive the approximation error, which is explicitly related to the Tucker rank.
> Since further theoretical derivation is relatively complex, we will refine the theory in our future work.
>
> [6] Kovachki, N., Lanthaler, S., & Mishra, S. (2021). On universal approximation and error bounds for Fourier neural operators. Journal of Machine Learning Research, 22(290), 1-76.
>
> [7] Anastassiou, G.A. (2010). ON BEST APPROXIMATION AND JACKSON-TYPE ESTIMATES BY GENERALIZED FUZZY POLYNOMIALS. In: Fuzzy Mathematics: Approximation Theory. Studies in Fuzziness and Soft Computing, vol 251. Springer, Berlin, Heidelberg.

---

> ### Author Response · Authors · 2025-11-20
> **Response to Reviewer ofCR Part 5**
>
> Should you require any further information, please do not hesitate to let us know. We would greatly appreciate your feedback.
>
> >**Reference**
>
> [1] J. Håstad, “Tensor rank is NP-complete,” in Proc. Int. Colloq. Automata,
> Lang., Program. Berlin, Germany: Springer, 1989, pp. 451–460.
>
> [2] Y. -L. Chen, C. -T. Hsu and H. -Y. M. Liao, "Simultaneous Tensor Decomposition and Completion Using Factor Priors," in IEEE Transactions on Pattern Analysis and Machine Intelligence, vol. 36, no. 3, pp. 577-591, March 2014, doi: 10.1109/TPAMI.2013.164.
>
> [3] J. Yu, G. Zhou, W. Sun and S. Xie, "Robust to Rank Selection: Low-Rank Sparse Tensor-Ring Completion," in IEEE Transactions on Neural Networks and Learning Systems, vol. 34, no. 5, pp. 2451-2465, May 2023, doi: 10.1109/TNNLS.2021.3106654.
>
> [4] Y. Luo, X. Zhao, Z. Li, M. K. Ng and D. Meng, "Low-Rank Tensor Function Representation for Multi-Dimensional Data Recovery," in IEEE Transactions on Pattern Analysis and Machine Intelligence, vol. 46, no. 5, pp. 3351-3369, May 2024, doi: 10.1109/TPAMI.2023.3341688.
>
> [5] Luo, D., O’Leary-Roseberry, T., Chen, P., & Ghattas, O. (2025). Efficient PDE-constrained optimization under high-dimensional uncertainty using derivative-informed neural operators. SIAM Journal on Scientific Computing, 47(4), C899-C931.
>
> [6] Kovachki, N., Lanthaler, S., & Mishra, S. (2021). On universal approximation and error bounds for Fourier neural operators. Journal of Machine Learning Research, 22(290), 1-76.
>
> [7] Anastassiou, G.A. (2010). ON BEST APPROXIMATION AND JACKSON-TYPE ESTIMATES BY GENERALIZED FUZZY POLYNOMIALS. In: Fuzzy Mathematics: Approximation Theory. Studies in Fuzziness and Soft Computing, vol 251. Springer, Berlin, Heidelberg.

---

### Author Response · Authors · 2025-11-29
**Brief Summary of the Discussion Phase**

Dear AC and Reviewers,

We sincerely appreciate the time and effort you have dedicated to reviewing our manuscript. We fully understand the additional workload and challenges brought about by the current situation, and we truly appreciate your continued efforts.

In the hope of facilitating your meta-review and decision-making process, we would like to provide a brief summary of the discussion phase:

**Reviewer ofCR** raised questions regarding the experimental setup (Part 1), computational complexity related to Tucker rank (Part 2), extension to high-dimensional PDEs (Part 3), and the theoretical approximation error with respect to rank (Part 4). After our replies, this reviewer maintained a positive score of **6** prior to the reverting process.

**Reviewer APdc** raised questions regarding the novelty of the decomposition compared to the tensorized FNO (Part 1), the inclusion of additional baselines and complex examples (Part 2,3), the underlying theoretical assumptions (Part 3), hyperparameter sensitivity (Part 4), and scalability (Part 5). After our replies, the reviewer explicitly acknowledged: *“I misunderstood the main point about how this approach reduces the computational complexity of the operator as opposed to T-FNO”*, and noted that *“the modified manuscript is clearer and avoid possible confusion between the two approaches which see similar on the surface level.”* The score was raised from 4 to **8** prior to the reverting process.

**Reviewer rkTN** raised questions regarding the convergence behavior (Part 1), scalability (Part 2, 6), rank selection (Part 3), and additional baselines (Part 4). After our replies, the reviewer explicitly confirmed that all concerns had been resolved and updated the score from 6 to **8** prior to the reverting process.

**Reviewer EBp7** raised questions regarding the novelty of the decomposition compared to the tensorized FNO (Part 2, 6), validation in 3D (or general high-D) PDEs (Part 3, 6, 8-11), code implementation (Part 4), and further discussion for computation complexity (Part 5). After our replies, the reviewer explicitly acknowledged: *“The results from Tables 9 and 10 demonstrate the potential to scale to problems at quite large scales”*, and expressly confirmed that the remaining concerns were adequately addressed. The score was raised from 4 to **8** prior to the reverting process.

As a result, the final scores before the reverting process were **6 / 8 / 8 / 8**. All revisions and additional clarifications provided during the discussion have been carefully incorporated into the updated manuscript, with changes highlighted in **blue** for your convenience. We hope this summary could assist your meta-review and decision-making process.

Additionally, we would like to affirm that the discussion of our paper was conducted in a fully legitimate, scientific, and anonymous manner. The authors did not engage in any non-scientific activities, deanonymization, or collusion throughout the discussion process.

---

### Meta-Review · Area_Chair_PcUN · 2026-01-06

**Summary:**

* Strength: Computational complexity of the Fourier transform is reduced from exponential to linear with respect to dimensions.
* Strength: The paper provides proof for the universal approximation capability of tensor-decomposed operators.
* Strength: Empirical results on 3D car pressure data show higher accuracy and speed than standard benchmarks.
* Strength: The method integrates with existing graph-based frameworks for non-grid geometries.
* Weakness: The architecture relies on manually selected ranks, potentially difficult to tune for new problems.

Tucker-FNO introduces a way to scale Fourier Neural Operators to high-dimensional problems. It factorizes the expensive multidimensional spectral convolution into parallel 1-dimensional streams. This change allows the model to process large data grids with significantly less memory and time (authors also include a proof showing this factorization does not lose the ability to represent complex functions)

Initial concerns about novelty were addressed during the discussion. The authors demonstrated that their approach reduces the cost of the operations themselves, which is different from prior work that only reduced model parameters. New tests on complex vehicle shapes prove the model works for real-world engineering tasks. Main limitation is the lack of an automated way to choose the optimal decomposition rank.

**Reviewer Concerns:**

**Addressed by rebuttal**
* [Core] Distinction from weight-decomposed operators: Rebuttal proved that Tucker-FNO reduces FFT costs while prior methods only reduced parameters.
* [Core] Lack of 3D geometry validation: Authors added experiments on the Car-CFD dataset showing improved speed and accuracy.
* [Non-core] Code completeness and reproducibility:  Authors provided full code for signal and PDE tasks in the supplement.
* [Core] Convergence in signal restoration: Authors showed that results improve with longer training and proper regularization.

**Still outstanding**
* [Core] Adaptive rank selection: The model requires the user to pick the Tucker rank manually before training.
* [Core] Error bounds related to rank: The exact relationship between the chosen rank and the approximation error is not yet fully quantified.

The discussion phase was productive. Aauthors provided new evidence regarding high-dimensional scaling + geometry. They clarified their contributions between their operator decomposition when compared to previous weight decomposition strategies.

**Reviewer Scores:**

* Reviewer ofCR
* Original score: 6
* Estimated score shift: unchanged
* The reviewer recognized the value of the theory and efficiency but correctly identified that hyperparameter tuning remains a challenge.

* Reviewer APdc
* Original score: 4
* Estimated score shift: increase
* The reviewer acknowledged they initially misunderstood how the method reduces complexity and was satisfied with the new geometry tests.

* Reviewer rkTN
* Original score: 6
* Estimated score shift: increase
* The reviewer confirmed that their concerns about overfitting and convergence in video tasks were addressed by the new experiments.

* Reviewer EBp7
* Original score: 4
* Estimated score shift: increase
* The reviewer found the extensive 3D fluid dynamics results and code updates sufficient to resolve all initial technical doubts.

The estimated score shifts show that the authors' managed to clarify the novelty of their proposed methods + providing new benchmarks. All reviewers who initially gave low scores indicated that their technical concerns were addressed.

---

### Decision · Program_Chairs · 2026-01-26

Accept (Poster)